# NeoRL: A Near Real-World Benchmark for Offline Reinforcement Learning

**Rongjun Qin**[1,2,*] **Xingyuan Zhang**[2,*], **Songyi Gao**[2,*], **Zhen Xu**[2], **Shengkai Huang**[2]
**Zewen Li**[2], **Weinan Zhang**[3], **Yang Yu**[1,2,◇]
[1]Nanjing University [2]Polixir Technologies [3]Shanghai Jiao Tong University

## Abstract

Offline reinforcement learning (RL) aims at learning a good policy from a batch of collected data, without extra interactions with the environment during training. However, current offline RL benchmarks commonly have a large *reality gap*, because they involve large datasets collected by highly exploratory policies, and the trained policy is directly evaluated in the environment. In real-world situations, running an overly exploratory policy is prohibited to ensure system safety, the data is commonly very limited, and a trained policy should be carefully evaluated before deployment. In this paper, we present a **N**ear r**e**al-world **o**ffline **RL** benchmark, named NeoRL, which contains datasets from various domains with controlled sizes, and extra test datasets for offline policy evaluation. We evaluate recent SOTA offline RL algorithms on NeoRL, through both online evaluation and purely offline evaluation. The empirical results demonstrate that the tested offline RL algorithms become less competitive to BC on many datasets, and the current offline policy evaluation methods can hardly select truly effective policies. We hope this work will shed some light on future research and draw more attention when deploying RL in real-world systems.

## 1 Introduction

Recent years have witnessed the great success of machine learning, especially deep learning systems, in computer vision, and natural language processing tasks. These tasks are usually based on a large dataset and are divided into training and test phases. The deep learning algorithm updates its model and tunes its hyper-parameters on the training dataset. In general, the trained model will be evaluated on the *unseen* test dataset before deployment. On the contrary, reinforcement learning (RL) agents interact with the environment and collect trajectory data online to maximize the expected return. Combined with deep learning, RL shows impressive ability in simulated environments even without human knowledge [1, 2]. However, beyond the scope of cheap simulated environments, current RL algorithms are hard to leverage in real-world applications, because the lack of a simulator makes it unrealistic to train an RL agent in critical applications. Fortunately, the running systems will produce data, which come from expert demonstrations, human-designed rules, learned prediction models, etc. A recent trend to alleviate the online trial-and-error cost is offline RL (batch RL) [3], which aims to learn an optimal policy from these static data, without extra online interactions. Thus, it is a promising approach to scale RL to more real-world applications, such as industrial control and quantitative trading, where online training may incur safety, and ethical problems.

**Data limitation in reality.** The literature of offline RL usually assumes a large batch of data at hand [4, 5]. However, the requirements of a large dataset limit the use of offline RL, because collecting enough data will be both time-consuming and costly for some real-world systems, e.g., the numbers

---

*These authors contribute equally. ◇ Correspondence to Yang Yu <yuy@polixir.ai>.

Submitted to the 35th Conference on Neural Information Processing Systems (NeurIPS 2021) Track on Datasets and Benchmarks. Do not distribute.

of trajectories are often less than 100 in the traditional industry. Therefore, the out-of-data problem is more challenging in the low-data regime for offline RL, and it is crucial for an RL policy to apply. Current offline RL methods are often pessimistic about the out-of-data distribution, by constraining the RL agent to be close to the offline data [6–8], or reconstructing an environment to learn from and only trusting it when the uncertainty about the generated data is low [9, 10]. This constraint obscures the distinction of naive behavioral cloning (BC). It is widely believed that the naive BC approach can hardly outperform the behavior policy that produced the offline data, and because the behavior policy is sub-optimal in general, BC is seldom applied in practice. An intuitive solution to the out-of-data problem is trying to cover the decision space (state-action space), e.g., collecting data from random policy or using replay buffer data [6, 4, 5]. The reality is that the real-world system commonly allows a working policy only to guarantee the system performance, thus the collected data are conservative, rather than exploratory.

**Evaluation protocol can be unpractical.** Another critical issue is about evaluating the trained policy and selecting the best of them before deployment. Figure 1 summarizes the pipeline of training and deploying offline RL. Analogous to a supervised learning task, it is necessary to validate the trained RL agent and finish the policy selection before deployment (we call it evaluation in this work), rather than directly running it in the real environment. In current literature, online policy evaluation is the mainstream approach, which refers to directly running the trained policy in the original environment, thus the validation phase has not been taken seriously. On the other hand, online evaluation is overly optimistic towards the trained policy since it allows perfect evaluation beforehand, thus is unrealistic to apply in the real world. Furthermore, online policy selection, which corresponds to utilizing the test

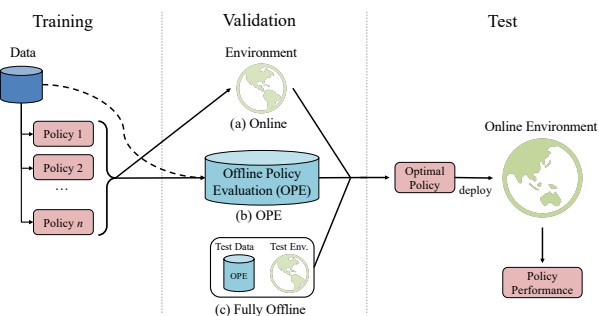

Figure 1: The pipeline of training and deploying offline RL, including training, validation (offline test before deployment), and test (deploying) phases. In the validation phase, (a) uses the online environment to validate the trained policy. (b) uses offline policy evaluation models on training data. (c) uses offline policy evaluation method on an extra offline test data or uses a test environment, where the test environment can be learned from test data or uses other cheap simulators instead. After validating, an optimal policy is obtained and deployed in the online environment.

dataset to select a model in supervised learning, will raise the ideal performance of an algorithm and result in misleading conclusions. Offline evaluation uses the dataset to assess a policy, without running in the environment [11–13]. Current benchmarks may use OPE methods on the training data [13], as in Figure 1(b) or perform online selection [14]. It will be more compelling to conduct OPE on an unseen test dataset or an unseen cheap test environment.

To tackle the above issues (we name them *reality gap*), we propose NeoRL, a suite of **n**ear r**e**al-world benchmarks for **o**ffline RL. The datasets include robotics, industrial control, finance trading and city management tasks with real-world properties. We provide three-level sizes of datasets, three-level quality of data collected from corresponding simulators, and benchmark recent model-free and model-based offline RL methods as a reference. The online and offline evaluations are both performed for policy selection based on each training algorithm. Moreover, the running system commonly involves a deterministic working policy and we slightly perturbed this policy to collect data from simulators, thus the performance of the perturbed behavior policy, i.e., the reward on the dataset decreases. So offline RL methods are also compared with the deterministic behavior policy, and it appears competitive to recent offline RL methods. The comparison results suggest that many of the current offline RL methods do not exceed this deterministic behavior policy significantly. Although offline evaluation before deployment is crucial, using current OPE methods can be hard to select a training algorithm or a trained policy that matches the online performance. We hope these findings will facilitate the design of offline RL algorithms for real-world applications.

## 2    Offline Reinforcement Learning

Traditional RL algorithms need to interact with the environment to collect trajectories with the current policy and update it, where the environment is treated as a black-box function. The RL agent needs to explore in the environment and then learn to get a high episode return.

In the offline RL setting, the environment is not provided during training, and only a batch of static data is accessible, thus the agent is unable to explore in the environment. Real-world tasks also involve issues such as action delays and non-stationarities [15]. The data can be gathered by sub-optimal expert policies with noise. For simplicity, we denote the policy that collected the data as the behavior policy $\pi_b$. Although off-policy algorithms can be readily applied to a static replay buffer, running an off-policy RL algorithm on a static buffer can sometimes diverge, due to issues like the distribution shift [16]. To learn a robust policy, recent offline RL algorithms explicitly or implicitly prevent the training policy from being too disjoint with $\pi_b$ [6, 17, 7, 8]. Besides, the absence of a cheap environment also makes it untamed to evaluate a training policy. Offline policy evaluation (OPE) is subtly different from off-policy policy evaluation [18], where the latter may have access to the behavior policy, thus novel techniques should be proposed to tackle the issue of offline evaluation.

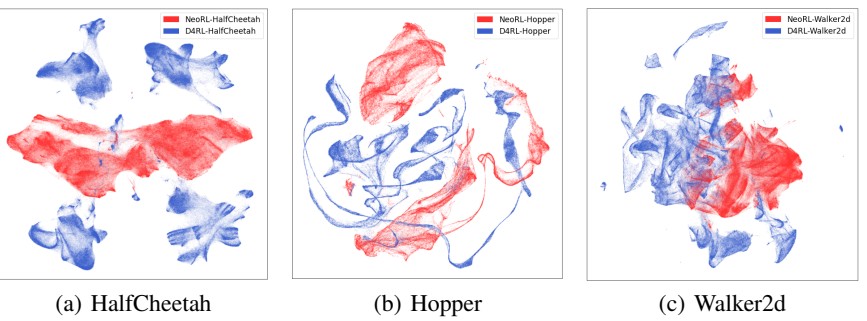

| (a) HalfCheetah | (b) Hopper | (c) Walker2d |

Figure 2: The distribution of state-action pairs. UMAP is the projection method.

## 3    Previous Benchmarks

Recently, some offline RL benchmarks have been proposed to facilitate the research and evaluation of offline RL algorithms. These benchmarks include multiple aspects of offline tasks and datasets, and also the performance of prior offline algorithms on these tasks [4, 5, 19]. Previously, the celebrated Atari 57 games and Gym-MuJoCo tasks (or DeepMind Control Suite [20]) have been widely used to benchmark online and offline RL methods. Besides these two domains, D4RL [5] also releases offline datasets of maze environments, FrankaKitchen [21], and offline CARLA [22], etc. These datasets in D4RL are designed to cover a range of challenging properties in real-world scenarios, including narrow and biased data distributions, multi-task data, sparse rewards, sub-optimal data. RL Unplugged [4] includes datasets from Atari and DM control suite, where the properties of these tasks range from different action spaces, observation spaces, partial observability, the difficulty of exploration, and real-world challenges [15]. Despite the properties of tasks are well covered, the properties of the real-world dataset are underexplored. To guarantee the system stability and performance, datasets from real-world running systems cannot be too exploratory. Recent works utilize the training data to assess RL algorithms [14, 6, 16] or sample from the training data to collect datasets [4, 5]. Intuitively, a wider data distribution weakens the exploration challenge, thus may overestimate the offline RL algorithms.

D4RL and RL Unplugged both noticed online policy selection is not allowed in a strict offline setting and proposed an evaluation protocol where they used a similar domain for policy selection and then trained with the optimal hyper-parameters from that similar domain. This protocol blurs the boundary of offline RL and transfer learning (or meta-learning), since we can learn from that domain and adapt to the online environment [23, 24]. DOPE benchmark [13] is designed to measure the performance of OPE methods and tested on RL Unplugged and D4RL. Offline training and OPE are conducted separately, yet have not been combined to select optimal policy before deployment.

# 4 The Reality Gap

Very few production environments are paired with a simulator in practice, and building a high-fidelity simulator comes at high expenses, e.g., it takes domain experts years of work to build a simulator in complex industrial tasks, while the devices may have aged and updated during this period, so that the simulator need to rebuild from scratch. The production environment is often risk-sensitive and the candidate policies must be evaluated before deployment. Besides, the data are directly logged from the production environment, so the data are often conservative and limited. Thus, the *reality gap* exists in the following forms:

**Offline evaluation before deployment**: In supervised learning, the trained models are evaluated on an unseen test set before deployment to assess the possible performance. Current offline RL algorithms are directly evaluating and selecting policy in an online manner [6, 16, 7, 9], which may cause unaffordable costs in real-world systems. Recent benchmarks have proposed a protocol to conduct evaluation through a different simulated environment that has similar dynamics [5, 4]. However, this evaluation approach somewhat contradicts the offline setting. If we had access to a cheap simulator that has similar dynamics, we could benefit more from this simulator, e.g., to pre-train a policy, and the offline RL problem reduces to transfer learning. Besides, it is unlikely to conduct such validation providing only the production environment is available. Nevertheless, offline policy selection and evaluation is compulsory for RL to apply in real-world domains.

**Conservative data**: Because of the cost and potential risks of random exploration, the human operators or designed rules in the production environment usually take conservative actions that stick to domain knowledge passed from generation to generation. This will result in a less diverse dataset than current benchmarks. These datasets can have different quality.

**Limited available data**: Although previous works assume that a large amount of logged data are easily obtained, it only holds for large-scale or streaming applications, such as recommendation systems. Datasets containing dozens of trajectories are common in traditional industry.

**Non-stationary environments**: The real-world systems appear to be non-stationary (highly stochastic, evolving through time, etc.). They may constantly evolve themselves and contain confounders that are not controllable.

Although previous benchmarks provide diverse datasets and useful tools for evaluating the performance of offline RL algorithms, the reality gap hinders the selection of the appropriate algorithm to train or the best policy to deploy in real-world systems. Considering the above gaps, we provide various datasets and tasks to fill these gaps and explore what can achieve with current offline RL algorithms under such limitations.

Table 1: An overview of existing benchmarks with respect to real-world properties. The principal differences are listed below, while some common features such as high state and action spaces are omitted. *SP* and *All* mean the benchmark provided this property for a *small portion* or all of their tasks and domains respectively.

| | Data properties | | | Domain property | Policy selection |
|---|---|---|---|---|---|
| Benchmark | Limited data | Conservative data | Contain overly exploratory data | Non-stationarity | Offline policy selection |
| RL Unplugged | SP | SP | ✓ | ✓ | × |
| D4RL | SP | SP | ✓ | × | × |
| NeoRL (Ours) | All | All | × | ✓ | ✓ |

# 5 Near Real-World Benchmarks

To address the above issues, we construct datasets with near real-world properties. In real-world systems, the working policies can be various and unknown, no matter whether they are trained, designed rules, or human demonstrations. We only assume that the working policies are sub-optimal and conservative, which are often common in realistic applications but are not well embodied in

previous benchmarks. Therefore, we produce policies to have these two properties. Most importantly, we follow the complete training and validation pipeline, conducting OPE for policy selection. The schematic comparisons with two existing benchmarks are listed in Table 1.

## 5.1 Near Real-World Environments

Compared to existing environments such as Gym-MuJoCo, in real-world environments, the state and action space can be relatively large and the transition functions are complex, with stronger stochasticity. Hence, we select tasks that are both high dimensional and with high stochasticity. i.e., industrial controlling, financial marketing, and city energy management scenarios. In real scenarios, the rewards may be calculated based on predefined quantifiable goals, e.g., a function of two successive states. Therefore, we encapsulate the reward function for each environment and provide an interface to use it, while for benchmarking, our default datasets contain the original environment rewards. By using tasks that capture the nature of real-world environments, it could help offline RL step further towards the real world.

## 5.2 Multi-Level Policy and Dataset Sizes

The historical interaction data collected from the real world are often produced by expert policies, rather than from a random policy or replay buffer. Note that these policies may not be optimal, and we have no knowledge of how sub-optimal they are. To simulate the real-world data collection scenarios, for each environment, we use SAC [25] to train on the environment until convergence and record a policy at every epoch. We denote the policy with the highest episode return during the whole training as the expert policy. Another three levels of policies with around $25\%, 50\%, 75\%$ expert performance are stored to simulate multi-level sub-optimal policies, denoted by low, medium, and high respectively. For each level, 4 policies with similar returns are selected, among which three policies are randomly selected to collect the training data used for offline RL policy training, and the left one produces the test data. The size of the test data is $1/10$ of the training data for each task. The extra test dataset can be used to design the offline evaluation method for the model selection during training and hyper-parameter selection. Because of human manipulation or sensory errors, demonstrations are noisy in general, to reproduce this phenomenon, with probability $20\%$, we sample from the trained Gaussian policies to execute, otherwise, use the mean of Gaussian to execute. Previous work [5] collects the data by sampling from the policy output distribution, which collects more explorative data. Besides the limited data setting, to help verify the impact of different amounts of data, for each task, we provide training data with a maximum of $10^4$ trajectories and three-level sizes of $10^2, 10^3$, and $10^4$ trajectories by default. An interface is available to slice and shuffle the data set arbitrarily to meet specific demands. It should be noted that the samples in domains with terminal functions may be less than #Trajectories × Max_Timesteps. See Appendix A for detailed sample sizes.

We use UMAP [26] to project the $(s, a)$ tuple onto a 2D plane for the seemingly closest datasets in the data collection process from D4RL and NeoRL, i.e., the 3 Gym-MuJoCo medium tasks on D4RL and the corresponding 3 medium tasks on NeoRL. The samples of the D4RL HalfCheetah-medium task and the NeoRL HalfCheetah-medium-1000 task are the same, so they can directly be used with UMAP. For Hopper and Walker2d, we use the first 387,466 and 768,249 samples from D4RL to make the size of samples the same. Figure 2 visualizes the data distribution of D4RL medium tasks and the NeoRL medium task, which demonstrates D4RL presents a wider data distribution, especially on HalfCheetah and Walker2d.

## 5.3 Benchmarks with Online and Offline Policy Selection

We benchmark some recent offline RL algorithms on the proposed datasets, with both online and offline policy selection. The online selection is contained because the performance via online selection can reflect the upper bound of an algorithm, and would help once OPE or other approaches can select the optimal policy without interacting with the environment. We also follow the fully offline training pipeline and benchmark these algorithms, where the policy model is selected by offline policy evaluation (OPE) methods. Especially, since data are collected with a perturbed $\pi_b$, which can degrade the dataset reward, we provide comparisons with the deterministic version of $\pi_b$.

## 6 Tasks and Datasets

Despite the tasks vary a lot, we provide a unified API on our datasets. Each item of a dataset consists of $(s_t, a_t, r_t, s_{t+1})$ tuples, and a unified interface for calling the reward calculation function and the terminal function for each task. Besides the provided reward for benchmarking, users can define their reward function for their purpose.

**Gym-MuJoCo** The Gym-MuJoCo is based on MuJoCo [27] engine, and its continuous control tasks are the standard testbeds for online RL algorithms. We select three environments and construct the offline RL tasks, i.e., HalfCheetah-v3, Walker2d-v3, and Hopper-v3. The subtle difference is that we include the first dimension of the position. Because part of the reward function of these three environments is the distance moved forward, so adding the location information simplifies the reward calculation for the current step. The 3 selected tasks are widely used in existing benchmarks, so we introduce the conservative and limited data properties into these tasks to investigate the impact on previous benchmarking results.

**IB** The industrial benchmark (IB) [28] is an RL benchmark environment motivated to simulate the characteristics presented in various industrial control tasks, such as wind or gas turbines, chemical reactors, etc. It includes problems commonly encountered in real-world industrial environments, such as high-dimensional continuous state spaces, delayed rewards, complex noise patterns, and high stochasticity of multiple reactive targets. Since the IB environment is high-dimensional and highly stochastic, we use the mean of Gaussian policy when collecting data, rather than sample from it.

**FinRL** The FinRL environment [29] provides a way to build a trading simulator that replicates the real stock market and supports backtesting with important market frictions such as transaction costs, market liquidity, investor risk aversion, and so on. In FinRL, per trading day can trade once for the stocks in the pool (30 stocks). The reward function is the difference in the total asset value between the end of the day and the day before. The environment may evolve itself as time elapsed. Because the dataset of $10^4$ trajectories is too large, we only provide $10^2$ and $10^3$ trajectories for FinRL.

**CityLearn** The CityLearn (CL) environment [30] reshapes the aggregation curve of electricity demand by controlling energy storage in different types of buildings. The objective is to coordinate the control of domestic hot water and chilled water storage by the electricity consumers (i.e., buildings) to reshape the overall curve of electricity demand. This environment is highly stochastic and with high-dimensional space.

For each domain, NeoRL contains 9 tasks (3 kinds of behavior policy performances and 3 kinds of sizes) except for FinRL environment. So currently, NeoRL contains 6 domains with 51 tasks in total. Detailed features of IB, FinRL, and CityLearn environment can be found in the Appendix A.

## 7 Experiments

To make fair comparisons for all the offline RL algorithms, a copy of codes with good quality (reproducibility, running time, resource demands, etc.) is the first to consider. However, publicly available codes are usually implemented with specific frameworks, and these algorithms are heavily coupled with specific frameworks. To focus on the algorithms and be easy to call them by a unified interface, we re-implement several algorithms (codes can be found in supplementary materials). The re-implementation has been verified on Gym-MuJoCo-medium tasks from D4RL dataset and matches the result (see Table 6). We roughly divide these algorithms into two categories: model-based and model-free. Since offline RL algorithms are sensitive to the choice of hyper-parameters, we conduct a grid search on hyper-parameter space to choose the best policy. Details of the hyper-parameters settings are in Appendix D.

### 7.1 Comparing Methods

#### 7.1.1 Baselines

**Expert** We run SAC until convergence in each environment to choose the policy with the highest returns and call it *expert*. Expert is used as a reference of a good policy. However, it does not imply that the expert is optimal.

**Deterministic Policy** Commonly, the running system involves a working deterministic policy. We take the deterministic behavior policy as the deterministic policy in our experiments.

**Behavior Policy** The behavior policy is used to collect the data. If the offline data collection process has no randomness injected, the behavior policy equals the deterministic policy. However, in many situations, we randomize the deterministic policy to mimic the stochasticity by systematical error.

### 7.1.2 Model-Free Methods

Most algorithms in current offline RL favor a model-free fashion, especially, by extending from off-policy algorithms. Since offline RL is learning from a fixed static dataset, directly utilizing off-policy algorithms will suffer from distribution shift [31] or extrapolation error [6], where the training policies try to reach out-of-data states and actions. For this reason, model-free algorithms usually explicitly or implicitly constrain the learned policy to be close to the offline data [6, 17, 7].

**BC** Behavioral cloning trains a policy to imitate the behavior policy from the data. We treat BC as a baseline of learning methods.

**BCQ** [6] learns a state-conditioned generative model $G_\omega(s)$, i.e., VAE, to mimic the behavior policy on the dataset, and a perturbation network $\xi_\phi(s, a, \Phi)$ to generate actions $\{a_i = a_i + \xi_\phi(s', a_i, \Phi)\}_{i=1}^n$, where $\{a_i \sim G_\omega(s')\}_{i=1}^n$ and the perturbation $\xi_\phi(s, a, \Phi)$ lies in the range $[-\Phi, \Phi]$. Controlling the perturbation amount by a hyper-parameter $\Phi$, the learned policy is constrained near the original data.

**PLAS** [17] is an extension of BCQ. Instead of learning a perturbation model on the action space, PLAS learns a deterministic policy on the latent space of VAE and assumes that the latent action space implicitly defines a constraint over the action output, thus the policy selects actions within the support of the dataset during training. In PLAS architecture, actions are decoded from latent actions. An optional perturbation layer can be applied in the PLAS architecture to improve the out-of-data generalization, akin to the perturbation model in BCQ.

**CQL** [7] penalizes the value function for states and actions not supported by the data to prevent overestimation of the training policy. By introducing an extra term under the offline data distribution ($\mathbf{E}_{s \sim \mathbf{D}, a \sim \hat{\pi}_b(s, a)}[Q(s, a)]$), CQL learns a *conservative* Q function. The authors have also proved this additional term helps achieve a tighter lower bound on the expected Q-value of the training policy $\pi$.

**CRR** [8] can be viewed as weighted BC which uses critic function $f$ to weight $\log \pi(a|s)$ to discourage $\pi$ from taking actions that are outside the offline data. Similar approaches include BAIL [32] and ABM [33]. We choose CRR as the representative due to its good performance and robustness to OPE-based offline selection [12].

### 7.1.3 Model-Based Methods

Although model-free methods perform well in offline RL algorithms and are easy to use, an overly con-strained policy can hinder stronger results, especially when the data is collected by low-performance behavior policies. On the other hand, model-based methods learn the transition function of the environment, which depends less on the quality of the behavior policy $\pi_b$. The transition model takes $(s, a)$ pair as input and outputs next state $s'$, thus online RL algorithms can use these models to perform rollout or plan. However, a learned imperfect model without any safeguards against model inaccuracy can result in *model exploitation* [34, 35].

**BREMEN** [36] uses BC to initialize the policy and uses TRPO [37] to update the policy with ensemble models. The authors proved the total variation of the learned policy and BC initialization grows linearly in terms of TRPO iteration, thus the policy search on a controllable space. Although BREMEN is not tailored towards purely offline, it reduces to purely offline by setting deployment times equal to 1. In this case, it is a straightforward model-based approach.

**MOPO** [9] constructs a pessimistic MDP from the transition models. MOPO uses the ensemble of models to estimate the uncertainty of model predictions. When generating rollouts from the transition models, the reward is penalized by the uncertainty term to encourage the policy to explore states that the transition models are certain about. The similar spirit appears in MOReL [10] which truncates the trajectory when the uncertainty becomes high.

Table 2: Average ranks over 51 tasks of online, FQE, WIS policy selection results.

| Name | Det. policy | Behavior policy | Random | BC | BCQ | PLAS | CQL | CRR | BREMEN | MOPO |
|---|---|---|---|---|---|---|---|---|---|---|
| Online | 4.80 | 5.67 | 8.92 | 4.94 | 6.15 | 5.33 | 2.17 | 3.98 | 5.25 | 7.76 |
| FQE | 3.29 | 3.92 | 8.61 | 3.22 | 6.20 | 6.61 | 4.43 | 4.53 | 6.20 | 8.00 |
| WIS | 3.71 | 4.43 | 8.61 | 3.65 | 5.90 | 5.69 | 4.51 | 4.43 | 6.24 | 7.84 |

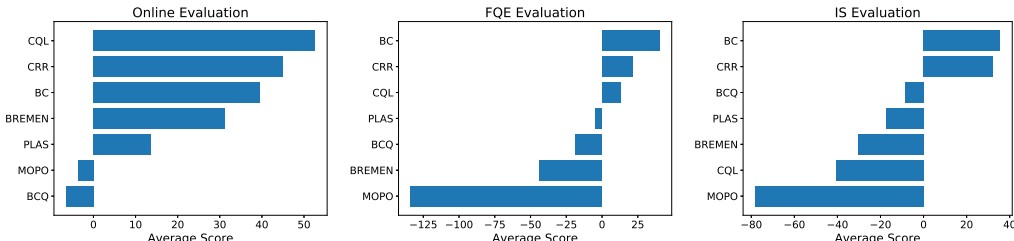

Figure 3: Average normalized score of each algorithm on 51 tasks by online evaluation and OPE.

## 7.2 Evaluation Protocol

**Online Evaluation.** Although not practical, the online selection score is important because it is indicative of performance given perfect offline selection methods while it favors algorithms with more hyper-parameters (also noted in [4]). We keep the policy at the last epoch for each hyper-parameter configuration and seed, except for BC (see Appendix. D). Each trained policy interacts with the environment for 1,000 episodes to get a score. The final performance is reported for the best hyper-parameter with the highest average score over 3 seeds.

**Offline Policy Selection.** A not evaluated policy is strongly forbidden to run in real-world systems, so offline evaluation is crucial for real-world applications, to know about the candidate policy in advance and to select the best policy for deployment. In our settings, we use off-policy evaluation (OPE) on the extra test dataset to select the best policy among policies trained by different hyper-parameters and seeds, then we report their online performance. To select the best model, an effective OPE method only needs to tell the relative performance between policies, rather than approximating the ground-truth performance to some extent.

In general, we only have one or two chances to deploy trained policies in real-world systems, even though the trained policies only differ in random seeds, they will be treated as different policies. Thus, we stored the policy from each hyper-parameter and each random seed to form the candidate policy set. Specifically, we choose two OPE methods: fitted Q evaluation (FQE) [38] and weighted importance sampling (WIS) [39]. FQE takes a policy as input and performs policy evaluation on the fixed dataset by Bellman backup. After learning the Q function of the policy, the performance is measured by the mean Q values on the initial states from the dataset and actions by the policy. WIS is a canonical variant of important sampling (IS). IS only uses the ratio between target policy and behavioral policy to weight the episodic reward in the dataset, while WIS can further reduce the IS variance. Both methods are run with 3 seeds on the candidate policy set. The three non-learning baselines do not need to go through OPE process.

## 7.3 Results

We calculate an average rank and average normalized scores respectively. The rank of an algorithm or baseline is determined by the score on each task, and the final average rank is computed over the 51 tasks. The average rank of each algorithm is shown in Table 2, and the average normalized scores are shown in Figure 3, for online and offline evaluation respectively. Detailed raw scores and normalized scores of each task are deferred to Appendix. F due to the page limitation. The normalization $100 \times \frac{\text{raw score} - \text{random score}}{\text{expert score} - \text{random score}}$ is also adopted in our evaluation.

In online evaluation, CQL achieves the highest rank of 2.17, which greatly outperforms other algorithms. BC matches the performance of the deterministic policy, which indicates that BC recovered the deterministic behavior policy from the datasets. Interestingly, results of BC form

Table 3: The difference of the normalized scores between each algorithms and the behavior policy on Gym-MuJoCo medium tasks.

| Task Name | BCQ | PLAS | CQL | MOPO |
|---|---|---|---|---|
| HalfCheetah-D4RL | 6.6 | 8.1 | 10.3 | 6.1 |
| HalfCheetah-NeoRL | 4.6 | 4.8 | 8.6 | 16.3 |
| Hopper-D4RL | 22.5 | 4.9 | 54.6 | $-5.5$ |
| Hopper-NeoRL | 5.7 | 19.2 | 22.5 | $-41.0$ |
| Walker2d-D4RL | 42.3 | 56.1 | 63.7 | 3.2 |
| Walker2d-NeoRL | 18.7 | $-8.4$ | 14.3 | $-3.1$ |

very strong baselines: the other six offline RL algorithms fail to outperform BC in 152 out of 306 comparisons (note that we have set the quality of datasets to three levels where BC is believed to perform poorly in the low-quality dataset). Using the Nemenyi test [40], the critical difference of 10 comparing methods over 51 tasks with confidence level 95% is 1.8970. Therefore, if we take BC as the reference, only CQL is significantly better than BC, while Random and MOPO are significantly worse. The result is the same if we take the deterministic policy as the reference. The winning rates against behavior policy, the deterministic policy, and BC for each compared baselines can be found in Table 21.

For model-based approaches, the overall performance is worse than model-free methods, but they can bring remarkable improvements in some domains. For instance, on HalfCheetah-Low and HalfCheetah-Medium tasks, BREMEN and MOPO can outperform other algorithms and baselines by a large margin, which reveals the potential of model-based offline RL approaches. However, the dataset can be less diverse as the quality improves, which may incur bias in environment learning and lead to poorer performance on high-quality datasets. To investigate how the conservative data affect

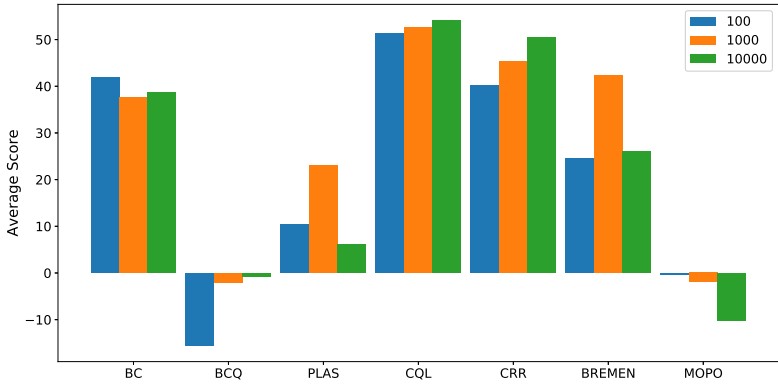

Figure 4: Average normalized score of each algorithms with respect to the number of trajectories. BCQ performed badly and even worse than random on IB domains, thus its average score is low.

the evaluation of offline RL, we calculate the difference of normalized score between the comparing algorithm (online performance) and the dataset reward (behavior policy performance) in Table 3. The performance on D4RL is directly adopted from the D4RL results or the original paper. It can be observed from Table 3, 10 out of 12 results are overestimated when compared with the behavior policy. We also evaluate the performance of each algorithm with respect to the number of trajectories used in training. As shown in Figure 4, for 5 out of 7 algorithms, the performance grows as the training number increases from 100 to 1000. However, only performances of BCQ, CQL, CRR increase when the training trajectories further increased to 10000. We notice that the BCQ performs badly on IB, which degrades its overall average score. The reason may lie in the highly stochastic nature of IB so that BCQ needs more carefully hyper-parameter tuning to achieve a decent score.

However, the result of offline evaluation favors BC. From Table 2 and Figure 3, for both OPE methods, the average rank and average normalized score of BC become the best. That means if we follow a

strict offline setting and fully offline training pipeline, current offline RL algorithms are no better than the naive BC and the deterministic policy. Except CQL and CRR, other learning algorithms significantly fall behind BC (see Table 22 and 23 for winning rates). From the normalized scores over three evaluations, on over a half of tasks, online evaluation, and two OPE could not reach an agreement on the best algorithms and policies (see Appendix F for detailed score). We conjecture this disagreement of online and offline evaluation is due to the performance of candidate policies; if the candidate set contains many extremely low-performance policies, FQE and WIS cannot distinguish them (see correlation figure in Appendix E, FQE and WIS can give both extremely high or low evaluation to a policy with very low online performance). Empirically, we may benefit from OPE if we can preclude these poor policies with little effort, e.g., preclude a policy when the value function loss explodes.

# 8    Conclusion

**NeoRL.** In this paper we present NeoRL, a near real-world benchmark for offline RL. Since real-world datasets are usually very limited and collected with conservative policies to ensure system safety. For real-world considerations, NeoRL focuses on conservative actions, limited data, non-stationary dynamics, and especially offline policy evaluation before deployment, which are ubiquitous and crucial in real-world decision-making scenarios. So far, NeoRL has included Gym-MuJoCo tasks, industrial control, financial trading, and city management tasks, where the training and test datasets are collected from these domains with different sizes.

**Findings.** We benchmark some state-of-the-art offline RL algorithms on NeoRL tasks, including model-free and model-based algorithms, in both online and offline policy evaluation manner. Surprisingly, the experimental results demonstrate that these compared offline RL algorithms fail to outperform neither the simplest behavior cloning method nor the deterministic behavior policy on NeoRL, only except CQL. With constraints to be close to the data or a pessimistic MDP, their performance may be extremely bounded by the data.

Our experiment results further show that model-based offline RL approaches are overall worse than model-free approaches. However, model-based approaches may have better potential to achieve the out-of-data generalization ability. Meanwhile, we have noticed that better model-learning approaches based on adversarial learning [41–43] could help. We will test these approaches in the future.

**Lessons learned.** For real-world applications, the trained policy must be evaluated before deployment. We recommend using offline policy evaluation methods on an unseen test dataset (or using a cheap learned simulator) to evaluate the trained policy. Despite the importance of offline evaluation in real-world scenarios, it can be inferred from the experiments that current offline policy evaluation methods (FQE and WIS in the experiments) may hardly help improve the policy selection and favor algorithms that are not sensitive to different hyper-parameters. We argue that offline RL algorithms should pay more attention to real-world restrictions and offline evaluation, and recommend using extra test datasets to conduct offline policy evaluation, which leads to a great challenge for existing offline RL methods.

**Future work.** In the future, we will step further towards real-world scenarios and investigate more real-world offline RL challenges, by constantly providing new near real-world datasets and tasks. We also hope the NeoRL benchmark will shed some light on future research and draw more attention to real-world RL applications.

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

## A  Task Description

Table 4: Configuration of environments.

| Environment | Observation Shape | Action Shape | Have Done | Max Timesteps |
|---|---|---|---|---|
| HalfCheetah-v3 | 18 | 6 | False | 1000 |
| Hopper-v3 | 12 | 3 | True | 1000 |
| Walker2d-v3 | 18 | 6 | True | 1000 |
| IB | 180 | 3 | False | 1000 |
| FinRL | 181 | 30 | False | 2516 |
| CL | 74 | 14 | False | 1000 |

Table 5: Number of samples contained in Hopper and Walker2d datasets.

| Tasks | Training Set | Test Set |
|---|---|---|
| Hopper-v3-Low-$10^2$ | 19259 | 1979 |
| Hopper-v3-Low-$10^3$ | 192346 | 19790 |
| Hopper-v3-Low-$10^4$ | 1918370 | 198188 |
| Hopper-v3-Medium-$10^2$ | 39219 | 2843 |
| Hopper-v3-Medium-$10^3$ | 387466 | 33435 |
| Hopper-v3-Medium-$10^4$ | 3885950 | 315728 |
| Hopper-v3-High-$10^2$ | 42142 | 4086 |
| Hopper-v3-High-$10^3$ | 413793 | 46981 |
| Hopper-v3-High-$10^4$ | 4168323 | 471693 |
| Walker2d-v3-Low-$10^2$ | 55353 | 5521 |
| Walker2d-v3-Low-$10^3$ | 543557 | 49426 |
| Walker2d-v3-Low-$10^4$ | 5455589 | 502659 |
| Walker2d-v3-Medium-$10^2$ | 77738 | 8605 |
| Walker2d-v3-Medium-$10^3$ | 768249 | 86776 |
| Walker2d-v3-Medium-$10^4$ | 7688849 | 867596 |
| Walker2d-v3-High-$10^2$ | 80880 | 7767 |
| Walker2d-v3-High-$10^3$ | 806876 | 83334 |
| Walker2d-v3-High-$10^4$ | 7963782 | 837832 |

**Gym-MuJoCo** We set EXCLUDE_CURRENT_POSITIONS_FROM_OBSERVATION to false to include the first dimension of the position in HalfCheetah-v3, Walker2d-v3, and Hopper-v3. We use Gym-MuJoCo: `https://gym.openai.com/envs/#mujoco`.

**IB** IB [28] simulates the characteristics presented in various industrial control tasks, such as wind or gas turbines, chemical reactors, etc. The raw system output for each time step is a 6-dimensional vector including velocity, gain, shift, setpoint, consumption, and fatigue. To enhance the Markov property, the authors stitch the system outputs of the last $K$ timesteps as observations ($K = 30$ by default). The action space is three-dimensional. Each action can be interpreted as three proposed changes to the three observable state variables called current steerings. Original codes can be found at `https://github.com/siemens/industrialbenchmark`.

**FinRL** FinRL [29] contains 30 stocks in the pool and the trading histories over the past 10 years. Each stock is represented as a 6-dimensional feature vector, where one dimension is the number of stocks currently owned, another five dimensions are the factor information of that stock. The observation has one dimension of information representing the current account cash balance. The dimension of the action space is 30, corresponding to the transactions of each of the thirty stocks. Original codes can be found at `https://github.com/AI4Finance-LLC/FinRL-Library`. For

**CityLearn** The CityLearn (CL) environment [30] reshapes the aggregation curve of electricity demand by controlling energy storage in different types of buildings. Domestic hot water (DHW) and solar power demands are modeled in the CL environment. High electricity demand raises the price of

electricity and the overall cost of the distribution network. Flattening, smoothing, and narrowing the electricity demand curve help to reduce the operating and capital costs of generation, transmission, and distribution. The observation encodes the states of buildings, including time, outdoor temperature, indoor temperature, humidity, solar radiation, power consumption, charging status of the cooling and heating storage units, etc. The action is to control each building to increase or decrease the amount of energy stored in its own heat storage and cooling equipment. Original codes can be found at `https://github.com/intelligent-environments-lab/CityLearn`.

The state and action spaces of all environments are summarized in Table 4. `Have Done` means the respective environment provides a terminal function that will finish the episode before reaching the maximum timesteps. For tasks without the terminal function, the number of samples in the dataset is Traj_Numbers * Max_Timesteps. On the other hand, for tasks with a terminal function, i.e. Hopper-v3 and Walker2d-v3, the samples can be less. The accurate sample numbers of these two tasks are summarized in Table 5.For domains that provide terminal function, the sample sizes may be less than $\#\texttt{Trajectories} \times \texttt{Max\_Timesteps}$, so we list the detailed number of samples for these domains in Table 5.

## B    The Verification of Re-implementation

The reproducibility issue is critical in offline RL. Even if using codes from the original authors, we may have difficulty reproducing the results for some algorithms on previous benchmarks. Random seeds and which model to keep seem to matter a lot. Since we aim to use the same training workflow, we re-implement compared baselines and have verified our re-implementations on D4RL MuJoCo-medium tasks. The hyper-parameters are set to the recommended values in the original papers. The results are shown in Table.6. Note that, in order to make a fair comparison between BREMEN and MOPO, we use the same implementation of stochastic ensemble models. However, we do notice that the original implementation of BREMEN adopted deterministic models, which may cause a discrepancy in the results.

Table 6: Normalized scores of the re-implementations on D4RL. Values in the brackets state the reported score in the original papers (except for CRR whose scores on D4RL are not available). The difference between two scores greater than 10 are in bold.

| Task Name | CQL | PLAS | BCQ | CRR | BREMEN | MOPO |
|---|---|---|---|---|---|---|
| Walker2d-medium | **78.5** (58.0) | 70.9 (66.9) | **69.0** (53.1) | 30.2 | **29.8** (59.6) | **27.6** (14.0) |
| Hopper-medium | 78.3 (79.2) | 34.2 (36.9) | **32.0** (54.5) | 53.3 | **29.7** (69.3) | 21.9 (26.5) |
| HalfCheetah-medium | 41.5 (44.4) | 40.9 (42.2) | 43.2 (40.7) | 39.8 | 50.2 (55.0) | 39.3 (40.2) |

## C    Computation Resources

We run all the experiments on the local clusters with multiple NVIDIA Telsa V100 GPUs (10 times CPU cores). By rough calculation, training all the offline policies require 21,420 GPU hours, and evaluating them with OPEs requires 15,300 GPU hours.

## D    Choice of Hyper-parameters

To make a fair comparison, all the policies and value functions are implemented by the same network structure, i.e., an MLP with 2 hidden layers and 256 units per layer. Because network architecture search (NAS) consumes large computation resources, especially in offline RL, since it takes a long time to train a policy and the ground-truth performance replies on online interactions. Thus, we directly use the same network architecture as the behavior policy that produced the datasets, and they do learn something in the online training process. We hope future work will enrich the property network architecture for offline RL. The output of the policies is transformed by `tanh` function to ensure the actions are within the range. For model-based approaches, the transition model is represented by an ensemble of Gaussian models, i.e., for each model, $s_{t+1} \sim \mathcal{N}(s_t + \Delta_\theta(s_t, a_t), \sigma_\theta(s_t, a_t))$, where $\Delta_\theta$ and $\sigma_\theta$ are implemented by an MLP with 4 hidden layers

and two heads. For Gym-MoJuCo tasks, we use 256 units in each hidden layer, for other tasks with higher input dimensions, we use 1024 units. Each transition model is trained by Adam optimizer via maximum likelihood until the MSE plateaus on the test dataset.

For BC, the policies are trained by Adam optimizer with a learning rate of 1e-3 for $100K$ steps with a batch size of 256, and it is early stopped with the lowest MSE on the test dataset to prevent overfitting. Although the best policy may get from the middle of the training process, except for BC, there does not exist a decent criterion to early stop. Thus, we only consider the finally trained policy for evaluation.

For BREMEN, we follow the original settings to treat 25 TRPO steps as an epoch and train for 250 epochs. For other methods, we treat 1000 learning steps as an epoch and then train BCQ, PLAS, CRR, MOPO for 200 epochs and train CQL for 300 epochs (The original CQL used 3000 epochs, but it spends too much time and the best performance can occur before 300 epochs).

Except for BC, offline RL algorithms can be very sensitive to the choice of hyper-parameters. To evaluate the performance of these algorithms, we conduct grid searches for the important hyper-parameters noted by the original papers. The search space of these algorithms is summarized in Table 7 and the hyper-parameters used in the reported results are summarized in Table 8. For parameters not mentioned, their values are the same as the original papers.

Table 7: The search space of hyper-parameters.

| Algorithms | Search Space |
|---|---|
| BCQ | $\Phi \in \{0.05, 0.1, 0.2, 0.5\}$ |
| PLAS | $\Phi \in \{0, 0.05, 0.1, 0.2, 0.5\}$ |
| CQL | variant $\in \{\mathcal{H}, \rho\}$
$\alpha \in \{5, 10\}$
$\tau \in \{-1, 2, 5, 10\}$ |
| CRR | advantage mode $\in \{$max, mean$\}$
weight mode $\in \{$exp, binary$\}$ |
| BREMEN | $h \in \{250, 1000\}$
exploration mode $\in \{$sample, static$\}$ |
| MOPO | uncertainty type $\in \{$aleatoric, disagreement$\}$
$h \in \{1, 5\}$
$\lambda \in \{0.5, 1, 2, 5\}$ |

For BCQ, the action is decoded from VAE plus a perturbation, i.e., $a = \hat{a} + \Phi \tanh(\xi_\phi(s, \hat{a}))$. Here, $\Phi$ controls the maximum deviation allowed for the learned policy from the behavior policy. We search for $\Phi \in \{0.05, 0.1, 0.2, 0.5\}$.

For PLAS, the default setting is to learn a deterministic policy in the latent space of VAE. The authors mentioned that a similar perturbation layer as BCQ can be applied to the output action to improve its generalization out of the dataset. Thus, we search for the value of $\Phi \in \{0, 0.05, 0.1, 0.2, 0.5\}$, where $\Phi = 0$ stands for the perturbation is not applied.

For CQL, we mainly consider three parameters mentioned in the original paper:

- Variant: The paper proposed two variants of CQL algorithms, i.e., CQL($\mathcal{H}$) and CQL($\rho$). The former uses entropy as the regularizer, whereas the latter one uses KL-divergence.

- Q-values penalty parameter $\alpha$: In the formulation of CQL, $\alpha$ stands for how large penalty will be enforced on the Q function. As suggested in the paper, we search for $\alpha \in \{5, 10\}$.

- $\tau$: Since $\alpha$ can be hard to tune, the authors also introduce an auto-tuning trick via dual gradient-descent. The trick introduces a threshold $\tau > 0$. When the difference between Q-values is greater than $\tau$, $\alpha$ will be auto-tuned to a greater value to make the penalty more aggressive. As suggested by the paper, we search $\tau \in \{-1, 2, 5, 10\}$. $\tau = -1$ indicates removing this trick.

Table 8: Hyper-parameters for reported results.

| Task Name | BCQ $\Phi$ | PLAS $\Phi$ | CQL Variant | CQL $\alpha$ | CQL $\tau$ | CRR Advantage Mode | CRR Weight Mode | BREMEN $h$ | BREMEN Exploration Mode | MOPO Uncertainty Type | MOPO $h$ | MOPO $\lambda$ |
|---|---|---|---|---|---|---|---|---|---|---|---|---|
| HalfCheetah-L-$10^2$ | 0.05 | 0.05 | $\mathcal{H}$ | 5 | 2 | mean | exp | 250 | sample | aleatoric | 5 | 1.0 |
| HalfCheetah-L-$10^3$ | 0.2 | 0.05 | $\mathcal{H}$ | 10 | 10 | mean | exp | 250 | sample | aleatoric | 5 | 1.0 |
| HalfCheetah-L-$10^4$ | 0.5 | 0.05 | $\mathcal{H}$ | 5 | 10 | max | binary | 250 | sample | disagreement | 1 | 1.0 |
| HalfCheetah-M-$10^2$ | 0.05 | 0.0 | $\rho$ | 10 | -1 | mean | binary | 1000 | sample | aleatoric | 5 | 1.0 |
| HalfCheetah-M-$10^3$ | 0.05 | 0.0 | $\rho$ | 5 | -1 | mean | binary | 250 | sample | aleatoric | 5 | 2.0 |
| HalfCheetah-M-$10^4$ | 0.05 | 0.0 | $\mathcal{H}$ | 10 | 5 | mean | binary | 250 | sample | disagreement | 1 | 5.0 |
| HalfCheetah-H-$10^2$ | 0.05 | 0.0 | $\rho$ | 5 | 10 | max | exp | 1000 | sample | aleatoric | 5 | 5.0 |
| HalfCheetah-H-$10^3$ | 0.05 | 0.0 | $\rho$ | 5 | 10 | mean | binary | 1000 | sample | aleatoric | 5 | 2.0 |
| HalfCheetah-H-$10^4$ | 0.05 | 0.0 | $\rho$ | 10 | -1 | mean | binary | 1000 | static | aleatoric | 1 | 1.0 |
| Hopper-L-$10^2$ | 0.1 | 0.1 | $\mathcal{H}$ | 5 | 10 | max | binary | 250 | static | aleatoric | 1 | 1.0 |
| Hopper-L-$10^3$ | 0.1 | 0.5 | $\mathcal{H}$ | 5 | 10 | mean | exp | 250 | static | disagreement | 5 | 5.0 |
| Hopper-L-$10^4$ | 0.2 | 0.2 | $\mathcal{H}$ | 5 | 10 | max | exp | 250 | static | disagreement | 1 | 0.5 |
| Hopper-M-$10^2$ | 0.1 | 0.0 | $\rho$ | 10 | 10 | mean | binary | 1000 | static | aleatoric | 1 | 5.0 |
| Hopper-M-$10^3$ | 0.05 | 0.1 | $\mathcal{H}$ | 10 | -1 | max | exp | 250 | static | disagreement | 5 | 5.0 |
| Hopper-M-$10^4$ | 0.05 | 0.05 | $\mathcal{H}$ | 5 | 10 | mean | exp | 250 | static | aleatoric | 5 | 1.0 |
| Hopper-H-$10^2$ | 0.05 | 0.0 | $\rho$ | 5 | 10 | mean | exp | 250 | static | aleatoric | 1 | 0.5 |
| Hopper-H-$10^3$ | 0.2 | 0.0 | $\rho$ | 10 | -1 | mean | binary | 250 | static | aleatoric | 1 | 1.0 |
| Hopper-H-$10^4$ | 0.05 | 0.0 | $\mathcal{H}$ | 5 | -1 | mean | binary | 1000 | static | disagreement | 1 | 0.5 |
| Walker2d-L-$10^2$ | 0.05 | 0.0 | $\rho$ | 10 | 2 | mean | exp | 1000 | static | disagreement | 1 | 0.5 |
| Walker2d-L-$10^3$ | 0.2 | 0.0 | $\mathcal{H}$ | 5 | 10 | mean | binary | 1000 | static | aleatoric | 1 | 5.0 |
| Walker2d-L-$10^4$ | 0.05 | 0.0 | $\mathcal{H}$ | 10 | 5 | max | exp | 1000 | static | aleatoric | 1 | 0.5 |
| Walker2d-M-$10^2$ | 0.1 | 0.0 | $\mathcal{H}$ | 5 | -1 | max | binary | 1000 | static | aleatoric | 5 | 5.0 |
| Walker2d-M-$10^3$ | 0.2 | 0.0 | $\mathcal{H}$ | 10 | 2 | mean | binary | 1000 | static | aleatoric | 5 | 5.0 |
| Walker2d-M-$10^4$ | 0.05 | 0.0 | $\rho$ | 5 | -1 | mean | binary | 1000 | static | aleatoric | 5 | 2.0 |
| Walker2d-H-$10^2$ | 0.05 | 0.0 | $\rho$ | 5 | -1 | mean | exp | 1000 | static | disagreement | 1 | 2.0 |
| Walker2d-H-$10^3$ | 0.2 | 0.0 | $\rho$ | 5 | -1 | mean | binary | 1000 | static | disagreement | 1 | 2.0 |
| Walker2d-H-$10^4$ | 0.1 | 0.0 | $\rho$ | 10 | -1 | mean | binary | 250 | static | disagreement | 5 | 1.0 |
| IB-L-$10^2$ | 0.5 | 0.05 | $\rho$ | 10 | 10 | mean | exp | 1000 | sample | aleatoric | 5 | 5.0 |
| IB-L-$10^3$ | 0.5 | 0.2 | $\rho$ | 5 | 5 | mean | exp | 250 | sample | disagreement | 5 | 5.0 |
| IB-L-$10^4$ | 0.5 | 0.05 | $\rho$ | 10 | -1 | mean | binary | 250 | static | aleatoric | 5 | 2.0 |
| IB-M-$10^2$ | 0.5 | 0.5 | $\mathcal{H}$ | 10 | 2 | mean | exp | 250 | static | aleatoric | 1 | 2.0 |
| IB-M-$10^3$ | 0.2 | 0.0 | $\mathcal{H}$ | 5 | 5 | max | exp | 1000 | static | aleatoric | 1 | 0.5 |
| IB-M-$10^4$ | 0.5 | 0.0 | $\mathcal{H}$ | 5 | 2 | max | binary | 250 | static | disagreement | 1 | 1.0 |
| IB-H-$10^2$ | 0.5 | 0.2 | $\rho$ | 10 | 5 | mean | exp | 250 | static | disagreement | 5 | 2.0 |
| IB-H-$10^3$ | 0.05 | 0.5 | $\rho$ | 5 | 2 | mean | exp | 250 | static | aleatoric | 1 | 1.0 |
| IB-H-$10^4$ | 0.1 | 0.05 | $\rho$ | 10 | 5 | mean | exp | 250 | static | aleatoric | 5 | 2.0 |
| FinRL-L-$10^2$ | 0.5 | 0.5 | $\mathcal{H}$ | 5 | 2 | mean | binary | 250 | static | aleatoric | 1 | 0.5 |
| FinRL-L-$10^3$ | 0.5 | 0.2 | $\mathcal{H}$ | 10 | -1 | max | exp | 250 | sample | aleatoric | 1 | 0.5 |
| FinRL-M-$10^2$ | 0.1 | 0.5 | $\rho$ | 10 | 2 | mean | binary | 250 | static | aleatoric | 1 | 0.5 |
| FinRL-M-$10^3$ | 0.5 | 0.0 | $\rho$ | 10 | 10 | max | exp | 1000 | sample | aleatoric | 5 | 0.5 |
| FinRL-H-$10^2$ | 0.5 | 0.0 | $\mathcal{H}$ | 5 | 10 | max | exp | 250 | sample | aleatoric | 5 | 0.5 |
| FinRL-H-$10^3$ | 0.5 | 0.2 | $\rho$ | 10 | -1 | mean | exp | 250 | sample | aleatoric | 5 | 0.5 |
| CL-L-$10^2$ | 0.05 | 0.0 | $\mathcal{H}$ | 10 | 10 | mean | binary | 1000 | static | disagreement | 1 | 5.0 |
| CL-L-$10^3$ | 0.2 | 0.05 | $\mathcal{H}$ | 10 | -1 | mean | binary | 250 | static | disagreement | 1 | 2.0 |
| CL-L-$10^4$ | 0.1 | 0.1 | $\mathcal{H}$ | 10 | -1 | mean | exp | 1000 | sample | aleatoric | 5 | 1.0 |
| CL-M-$10^2$ | 0.2 | 0.05 | $\rho$ | 10 | 10 | mean | exp | 250 | static | disagreement | 5 | 0.5 |
| CL-M-$10^3$ | 0.2 | 0.0 | $\mathcal{H}$ | 10 | 2 | max | binary | 1000 | sample | aleatoric | 1 | 0.5 |
| CL-M-$10^4$ | 0.05 | 0.1 | $\mathcal{H}$ | 10 | 10 | max | exp | 250 | static | aleatoric | 1 | 5.0 |
| CL-H-$10^2$ | 0.05 | 0.0 | $\rho$ | 10 | 2 | mean | exp | 250 | static | disagreement | 5 | 0.5 |
| CL-H-$10^3$ | 0.1 | 0.0 | $\mathcal{H}$ | 10 | 10 | mean | exp | 250 | static | aleatoric | 5 | 1.0 |
| CL-H-$10^4$ | 0.05 | 0.0 | $\mathcal{H}$ | 10 | 2 | mean | exp | 250 | static | aleatoric | 5 | 5.0 |

Note that, there is an approximate-max backup trick mentioned in the original paper. By default, the bellman backup is computed with double Q, i.e., $y = r + \min_{i=1,2} Q_i(s', a')$, where $a' \sim \pi(s')$. In addition, the authors propose a approximate-max backup, which use 10 samples to approximate the max Q-values, where the backup is computed by $y = r + \min_{i=1,2} \max_{a'_1 \dots a'_{10} \sim \pi(s')} Q_i(s', a')$. In the former experiments, we found this trick impairs the performance. Thus, we keep the double-Q target to reduce the search space.

In CRR, the policy is learned via $\arg\max_\pi \mathbb{E}_{(s,a) \sim D} \left[ f(Q_\theta, \pi, s, a) \log \pi(a|s) \right]$, where $f$ is the weight function that is non-negative and monotonous in Q value. The authors mainly use the advantage function to compute $f$. There are mainly two design choices that effect $f$:

- Advantage mode: The original paper gives two methods to estimate the advantage function, i.e., $\hat{A}_{\text{mean}}(s,a) = Q_\theta(s,a) - \frac{1}{m} \sum_{i=1}^{m} Q_\theta(s,a_i)$ and $\hat{A}_{\max}(s,a) = Q_\theta(s,a) - \max_{i=1\dots m} Q_\theta(s,a_i)$, where $a_i \sim \pi(a|s)$. The former one is termed as *mean* while the later one is termed *max*.

- Weight mode: The original paper gives two ways to compute weight given advantage, i.e., $f := \mathbb{1}\left[\hat{A}(s,a) > 0\right]$ and $f := \exp(A(s,a)/\beta)$. The former one is termed as *binary* while the later one is termed *exp*. For the *exp* method, the $\beta$ is set to 1 to be align with the original paper.

For BREMEN, we consider two parameters mentioned in the original paper:

- Rollout horizon $h$: BREMEN uses the transition models to generate imaginary rollouts whose length is controlled by parameter $h$. As suggested in the original paper, we search for $h \in \{250, 1000\}$.

- Exploration Mode: In the original paper, the authors conducted an ablation study on the exploration strategy when generating rollouts. They found using a stationary Gaussian noise with $\sigma = 0.1$ other than sampling from the policy can significantly boost the performance. However, in our experiment, we observe that using stationary noise does not always help. Thus, we perform a search on this strategy. The term *sample* is referred to directly sample from the policy, while *static* is referred to the stationary noise suggested by the authors.

For MOPO, we consider three parameters mentioned in the original paper:

- Uncertainty type: In the default setting, MOPO uses the maximum $L_2$-norm of the output standard deviation among ensemble transition models, i.e., $\max_{i=1...N} \|\sigma_\theta^i(s,a)\|_2^2$, as the uncertainty measure. Since the learned variance can theoretically recover the true aleatoric uncertainty [44, 9], we denote this type of uncertainty as aleatoric. Another variant that uses the disagreement between ensemble transition models is also included, i.e., $\max_{i=1...N} \|\Delta_\theta^i(s,a) - \frac{1}{N}\sum_i \Delta_\theta^i(s,a)\|_2^2$. We refer to this variant as disagreement.

- Rollout horizon $h$: MOPO uses a branch rollout trick that rollouts from states in the dataset with a small length. $h$ determines the length of the rollout. As suggested in the paper, we search for $h \in \{1, 5\}$.

- Uncertainty penalty weight $\lambda$: The main idea of MOPO is to penalize the reward function with the uncertainty term, i.e., $\hat{r} = r - \lambda u(s,a)$. Here, $\lambda$ control the amplitude of the penalty. As suggested in the original paper, we search for $\lambda \in \{0.5, 1, 2, 5\}$.

# E Details of Offline Policy Evaluation

This section describes implementation details and hyper-parameters for offline evaluation and provides additional results. Corresponding to supervised learning, all the OPE methods are conducted on the holdout test dataset with a discount factor $\gamma = 0.99$.

For FQE, we follow the hyper-parameters in [12]. The critic network is implemented with an MLP of 4 layers with 1024 units per layer and is trained for $250K$ steps by Adam optimizer with a batch size of 256. In the experiment, we observe that FQE is inclined to explode to extremely large values. Therefore, we use a value clipping trick on the target of bellman backups. The max and min values are computed by the rewards from the dataset with $40\%$ enlargement of the interval. That is, $v_{\max} = (1.2r_{\max} - 0.2r_{\min})/(1 - \gamma)$ and $v_{\min} = (1.2r_{\min} - 0.2r_{\max})/(1 - \gamma)$.

IS based methods rely on the probability density function of policies to compute the important ratio $\rho = \frac{\pi(a|s)}{\pi_b(a|s)}$. However, the behavior policy $\pi_b(a|s)$ is unknown in the offline setting, and the target policy $\pi(a|s)$, i.e., the one trained by offline RL algorithms, can also be deterministic or stochastic with implicit distribution, as in BCQ and PLAS. Thus, we adopt BC to estimate the density function of the respective policy. For the behavior policy, BC is directly applied to the raw dataset. For the target policy, we first relabel the dataset by the output of the target policy, then apply BC on the relabeled dataset. We follow [45] to implement the WIS. The policy is implemented as a `TanhGaussian` distribution in BC with an MLP of 2 layers and 256 units per layer.

In addition to directly select the best policy according to the OPE estimations, we also consider other two metrics to evaluate the OPE methods as in [12, 13]:

**Rank Correlation Score (RC Score)**: RC score indicates how the OPE produces the same rank as the ground-truth in the online evaluation. It is computed as Spearman correlation coefficient between

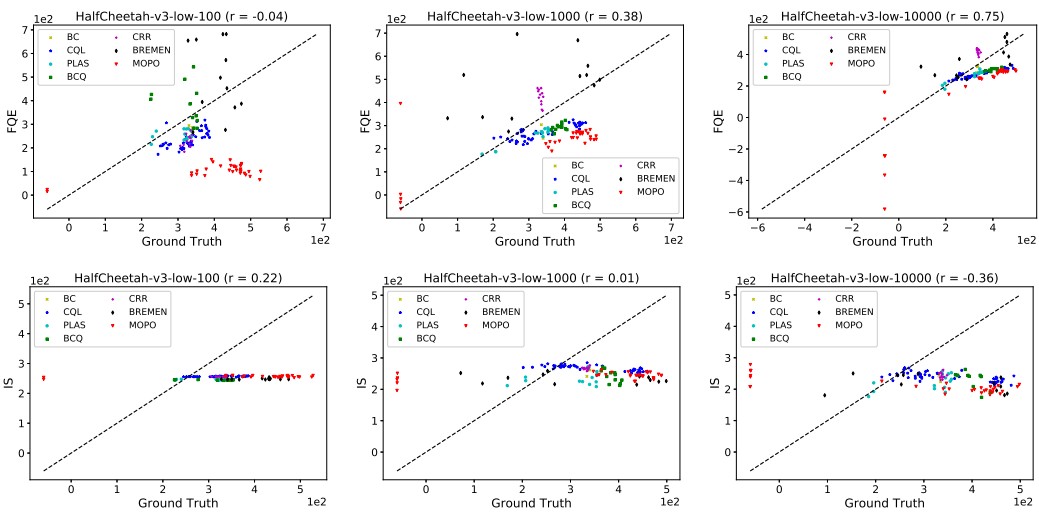

Figure 5: Scatter plot of OPE results for HalfCheetah-Low tasks. $r$ stands for the correlation coefficient.

the two rankings produced by OPE and online evaluation respectively. RC score lies in $[-1, 1]$, and if the rank is uniformly random, the score will be $0$.

**Top-$K$ Score**: Top-$K$ score represents the relative performance of the chosen $K$ policies via OPE. To compute this score, the real online performance of each policy is first normalized to a score within $[0, 1]$ by the min and max values over the whole candidate policy set of all the algorithms. Let $\pi_{\text{off}}^k$ denote the $k$-th ranked policy by the offline evaluation, then we use $\frac{1}{K}\sum_{k=1}^{K} \pi_{\text{off}}^k$ and $\max_k\{\pi_{\text{off}}^k\}$ as the mean and max top-$K$ score respectively. We report the scores with $K \in \{1, 3, 5\}$.

In addition, we report the average performance of the candidate policies as Policy Mean Score. Note that, it also represents the expectation of the top-1 score for a random selection method. All the metrics are shown from Table 9 to 20 for each domain and corresponding OPE method.

We also show additional correlation figures of each task on whole candidate policies below. The scatter plots compare the estimated values from OPEs against the ground truth values for every policy. The ground truth is estimated by the online performance, i.e., $v_{\text{gt}} = \frac{R_{\text{online}}}{(1-\gamma)h_{\text{max}}}$, where $h_{\text{max}}$ denotes the maximum horizon of the environment. Dots on the dashed line indicates the OPE methods perfectly predict the online performance. We found the FQE and WIS estimation can be far from the real online performance in most tasks. Especially, we can identify a vertical line on the left in most of the scatter plots of FQE, which indicates FQE fails to evaluate policies with very bad performance.

Table 9: FQE performance on the policies from HalfCheetah tasks. L, M, H stands for low, medium and high quality of dataset.

| Task | RC Score | Top-1 Mean Score | Top-3 Mean Score | Top-5 Mean Score | Top-1 Max Score | Top-3 Max Score | Top-5 Max Score | Policy Mean Score |
|---|---|---|---|---|---|---|---|---|
| HalfCheetah-L-$10^2$ | $-.122 \pm .007$ | $.834 \pm .007$ | $.787 \pm .000$ | $.771 \pm .000$ | $.834 \pm .007$ | $.839 \pm .000$ | $.839 \pm .000$ | $0.701$ |
| HalfCheetah-L-$10^3$ | $.306 \pm .036$ | $.586 \pm .000$ | $.804 \pm .001$ | $.785 \pm .065$ | $.586 \pm .000$ | $.936 \pm .003$ | $.980 \pm .028$ | $0.724$ |
| HalfCheetah-L-$10^4$ | $.631 \pm .052$ | $.621 \pm .439$ | $.697 \pm .275$ | $.730 \pm .124$ | $.621 \pm .439$ | $.932 \pm .000$ | $.932 \pm .000$ | $0.700$ |
| HalfCheetah-M-$10^2$ | $-.636 \pm .009$ | $.730 \pm .000$ | $.741 \pm .041$ | $.724 \pm .085$ | $.730 \pm .000$ | $.884 \pm .021$ | $.899 \pm .000$ | $0.649$ |
| HalfCheetah-M-$10^3$ | $.024 \pm .030$ | $.640 \pm .195$ | $.620 \pm .105$ | $.581 \pm .043$ | $.640 \pm .195$ | $.807 \pm .134$ | $.807 \pm .134$ | $0.683$ |
| HalfCheetah-M-$10^4$ | $.382 \pm .016$ | $.449 \pm .007$ | $.481 \pm .030$ | $.499 \pm .017$ | $.449 \pm .007$ | $.537 \pm .083$ | $.622 \pm .046$ | $0.634$ |
| HalfCheetah-H-$10^2$ | $-.295 \pm .021$ | $.518 \pm .190$ | $.418 \pm .079$ | $.459 \pm .065$ | $.518 \pm .190$ | $.653 \pm .001$ | $.738 \pm .059$ | $0.468$ |
| HalfCheetah-H-$10^3$ | $-.207 \pm .028$ | $.760 \pm .103$ | $.441 \pm .146$ | $.429 \pm .145$ | $.760 \pm .103$ | $.795 \pm .089$ | $.795 \pm .089$ | $0.533$ |
| HalfCheetah-H-$10^4$ | $.204 \pm .005$ | $.363 \pm .000$ | $.333 \pm .015$ | $.316 \pm .008$ | $.363 \pm .000$ | $.363 \pm .000$ | $.363 \pm .000$ | $0.467$ |
| Average | $.032 \pm .369$ | $.611 \pm .226$ | $.591 \pm .202$ | $.588 \pm .179$ | $.611 \pm .226$ | $.750 \pm .193$ | $.775 \pm .187$ | $.618 \pm .096$ |

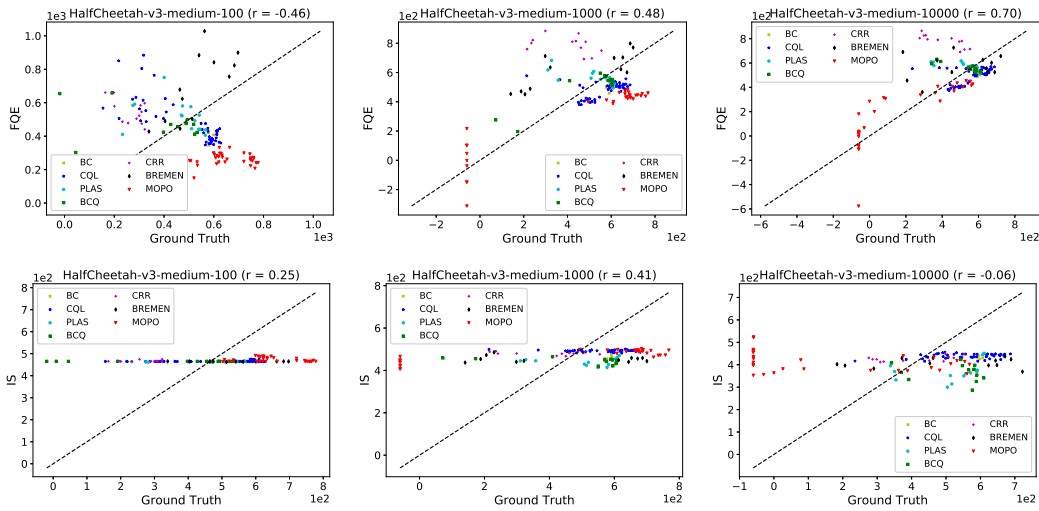

Figure 6: Scatter plot of OPE results for HalfCheetah-Medium tasks. $r$ stands for the correlation coefficient.

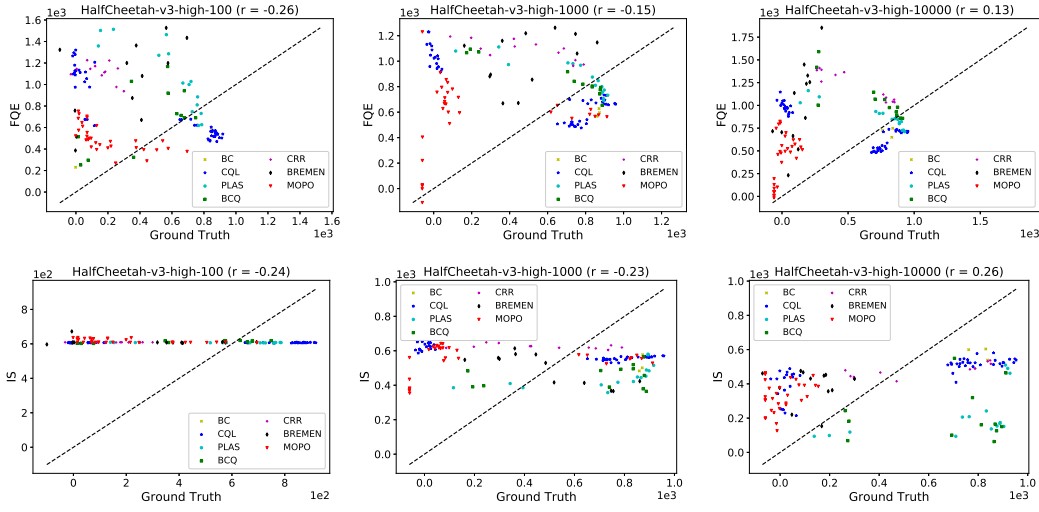

Figure 7: Scatter plot of OPE results for HalfCheetah-High tasks. $r$ stands for the correlation coefficient.

Table 10: IS performance on the policies from HalfCheetah tasks. L, M, H stands for low, medium and high quality of dataset.

| Task | RC Score | Top-1 Mean Score | Top-3 Mean Score | Top-5 Mean Score | Top-1 Max Score | Top-3 Max Score | Top-5 Max Score | Policy Mean Score |
|---|---|---|---|---|---|---|---|---|
| HalfCheetah-L-$10^2$ | $.039 \pm .242$ | $.689 \pm .044$ | $.729 \pm .053$ | $.732 \pm .058$ | $.689 \pm .044$ | $.855 \pm .133$ | $.915 \pm .062$ | $0.701$ |
| HalfCheetah-L-$10^3$ | $-.309 \pm .034$ | $.658 \pm .069$ | $.649 \pm .052$ | $.642 \pm .006$ | $.658 \pm .069$ | $.718 \pm .017$ | $.742 \pm .000$ | $0.724$ |
| HalfCheetah-L-$10^4$ | $-.446 \pm .015$ | $.457 \pm .333$ | $.418 \pm .045$ | $.511 \pm .026$ | $.457 \pm .333$ | $.654 \pm .094$ | $.746 \pm .067$ | $0.700$ |
| HalfCheetah-M-$10^2$ | $.215 \pm .068$ | $.764 \pm .100$ | $.653 \pm .116$ | $.664 \pm .104$ | $.764 \pm .100$ | $.789 \pm .083$ | $.802 \pm .080$ | $0.649$ |
| HalfCheetah-M-$10^3$ | $.218 \pm .099$ | $.540 \pm .254$ | $.633 \pm .126$ | $.642 \pm .057$ | $.540 \pm .254$ | $.829 \pm .050$ | $.829 \pm .050$ | $0.683$ |
| HalfCheetah-M-$10^4$ | $.108 \pm .017$ | $.001 \pm .000$ | $.001 \pm .000$ | $.072 \pm .100$ | $.001 \pm .000$ | $.001 \pm .000$ | $.184 \pm .258$ | $0.634$ |
| HalfCheetah-H-$10^2$ | $.061 \pm .184$ | $.147 \pm .064$ | $.207 \pm .087$ | $.205 \pm .060$ | $.147 \pm .064$ | $.351 \pm .231$ | $.417 \pm .176$ | $0.468$ |
| HalfCheetah-H-$10^3$ | $-.192 \pm .103$ | $.105 \pm .059$ | $.100 \pm .020$ | $.125 \pm .028$ | $.105 \pm .059$ | $.191 \pm .089$ | $.321 \pm .118$ | $0.533$ |
| HalfCheetah-H-$10^4$ | $.346 \pm .029$ | $.880 \pm .000$ | $.870 \pm .008$ | $.870 \pm .023$ | $.880 \pm .000$ | $.916 \pm .025$ | $.948 \pm .035$ | $0.467$ |
| Average | $.004 \pm .275$ | $.471 \pm .333$ | $.473 \pm .297$ | $.496 \pm .279$ | $.471 \pm .333$ | $.589 \pm .326$ | $.656 \pm .287$ | $.618 \pm .096$ |

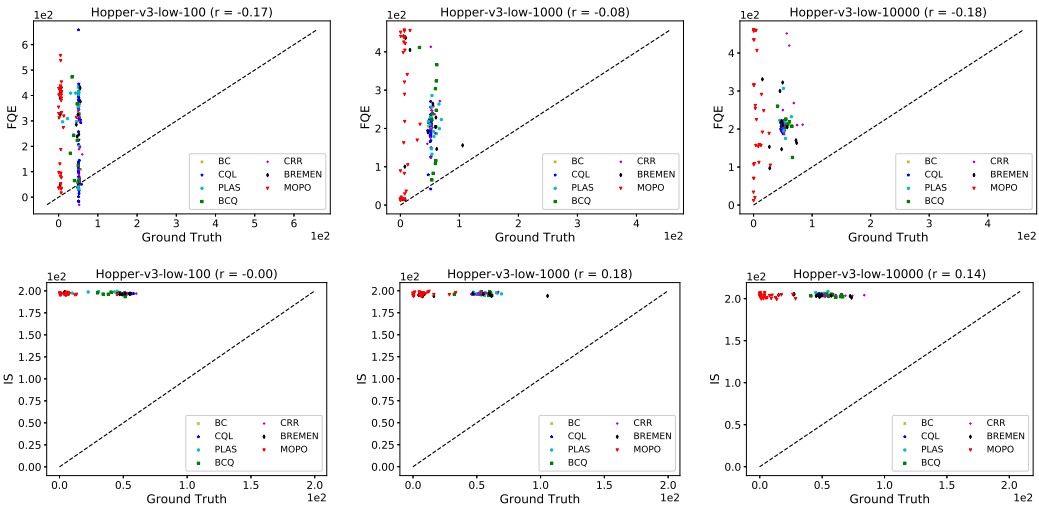

Figure 8: Scatter plot of OPE results for Hopper-Low tasks. $r$ stands for the correlation coefficient.

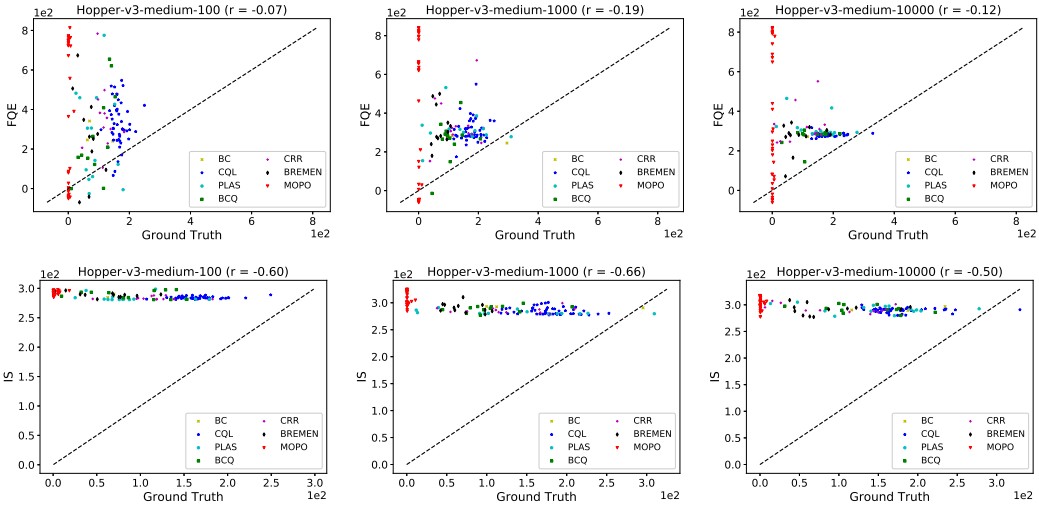

Figure 9: Scatter plot of OPE results for Hopper-Medium tasks. $r$ stands for the correlation coefficient.

Table 11: FQE performance on the policies from Hopper tasks. L, M, H stands for low, medium and high quality of dataset.

| Task | RC Score | Top-1 Mean Score | Top-3 Mean Score | Top-5 Mean Score | Top-1 Max Score | Top-3 Max Score | Top-5 Max Score | Policy Mean Score |
|---|---|---|---|---|---|---|---|---|
| Hopper-L-$10^2$ | $-.101 \pm .059$ | $.586 \pm .359$ | $.416 \pm .141$ | $.377 \pm .029$ | $.586 \pm .359$ | $.830 \pm .015$ | $.844 \pm .005$ | 0.619 |
| Hopper-L-$10^3$ | $.085 \pm .071$ | $.022 \pm .029$ | $.057 \pm .029$ | $.053 \pm .011$ | $.022 \pm .029$ | $.104 \pm .038$ | $.112 \pm .031$ | 0.386 |
| Hopper-L-$10^4$ | $.223 \pm .152$ | $.260 \pm .331$ | $.267 \pm .211$ | $.189 \pm .130$ | $.260 \pm .331$ | $.551 \pm .361$ | $.551 \pm .361$ | 0.491 |
| Hopper-M-$10^2$ | $-.086 \pm .065$ | $.104 \pm .107$ | $.215 \pm .054$ | $.131 \pm .032$ | $.104 \pm .107$ | $.404 \pm .093$ | $.404 \pm .093$ | 0.383 |
| Hopper-M-$10^3$ | $-.005 \pm .177$ | $.001 \pm .001$ | $.002 \pm .000$ | $.002 \pm .000$ | $.001 \pm .001$ | $.002 \pm .001$ | $.002 \pm .000$ | 0.359 |
| Hopper-M-$10^4$ | $-.112 \pm .113$ | $.001 \pm .000$ | $.002 \pm .000$ | $.002 \pm .000$ | $.001 \pm .000$ | $.002 \pm .000$ | $.002 \pm .000$ | 0.344 |
| Hopper-H-$10^2$ | $-.246 \pm .060$ | $.054 \pm .074$ | $.020 \pm .024$ | $.012 \pm .015$ | $.054 \pm .074$ | $.055 \pm .073$ | $.055 \pm .073$ | 0.402 |
| Hopper-H-$10^3$ | $-.437 \pm .028$ | $.002 \pm .000$ | $.001 \pm .000$ | $.003 \pm .002$ | $.002 \pm .000$ | $.002 \pm .000$ | $.005 \pm .005$ | 0.387 |
| Hopper-H-$10^4$ | $-.201 \pm .063$ | $.001 \pm .001$ | $.008 \pm .009$ | $.005 \pm .006$ | $.001 \pm .001$ | $.021 \pm .027$ | $.021 \pm .027$ | 0.409 |
| Average | $-.098 \pm .206$ | $.115 \pm .250$ | $.110 \pm .168$ | $.086 \pm .129$ | $.115 \pm .250$ | $.219 \pm .314$ | $.222 \pm .316$ | $.420 \pm .080$ |

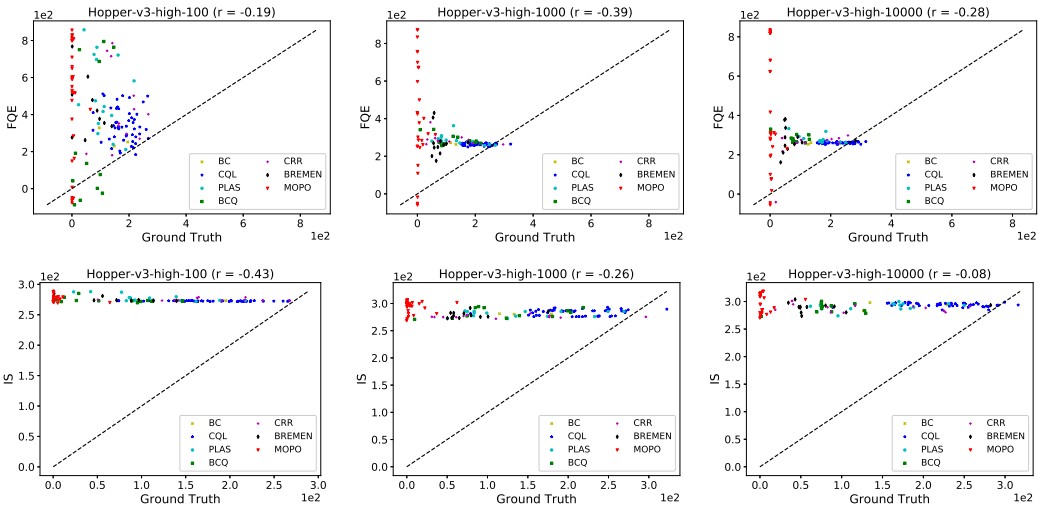

Figure 10: Scatter plot of OPE results for Hopper-High tasks. $r$ stands for the correlation coefficient.

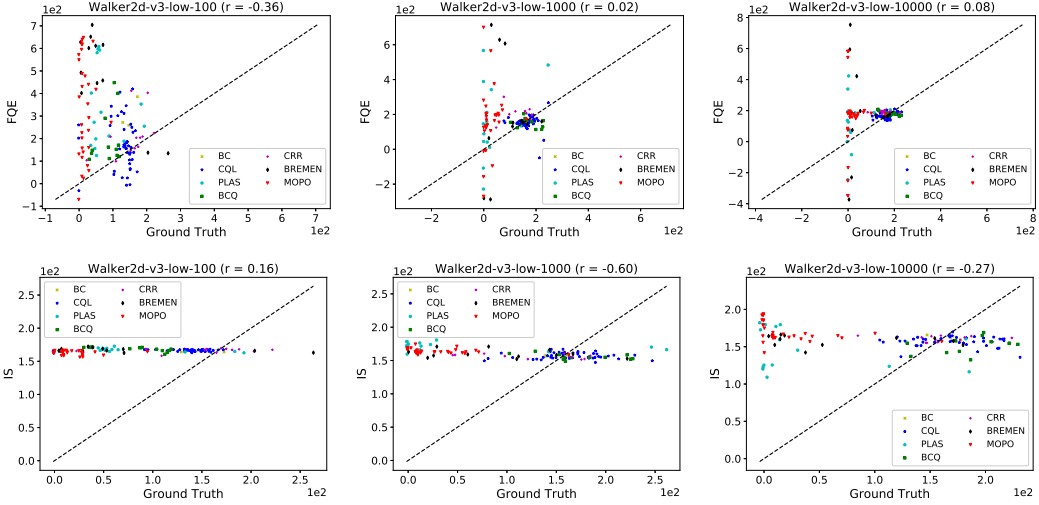

Figure 11: Scatter plot of OPE results for Walker2d-Low tasks. $r$ stands for the correlation coefficient.

Table 12: IS performance on the policies from Hopper tasks. L, M, H stands for low, medium and high quality of dataset.

| Task | RC Score | Top-1 Mean Score | Top-3 Mean Score | Top-5 Mean Score | Top-1 Max Score | Top-3 Max Score | Top-5 Max Score | Policy Mean Score |
|---|---|---|---|---|---|---|---|---|
| Hopper-L-$10^2$ | $.098 \pm .091$ | $.378 \pm .304$ | $.375 \pm .208$ | $.323 \pm .142$ | $.378 \pm .304$ | $.545 \pm .157$ | $.626 \pm .179$ | $0.619$ |
| Hopper-L-$10^3$ | $.161 \pm .037$ | $.287 \pm .236$ | $.406 \pm .024$ | $.338 \pm .037$ | $.287 \pm .236$ | $.587 \pm .023$ | $.609 \pm .027$ | $0.386$ |
| Hopper-L-$10^4$ | $.138 \pm .113$ | $.653 \pm .000$ | $.558 \pm .106$ | $.417 \pm .120$ | $.653 \pm .000$ | $.700 \pm .066$ | $.700 \pm .066$ | $0.491$ |
| Hopper-M-$10^2$ | $-.430 \pm .158$ | $.338 \pm .187$ | $.273 \pm .084$ | $.263 \pm .107$ | $.338 \pm .187$ | $.436 \pm .122$ | $.468 \pm .138$ | $0.383$ |
| Hopper-M-$10^3$ | $-.620 \pm .045$ | $.002 \pm .000$ | $.001 \pm .000$ | $.001 \pm .000$ | $.002 \pm .000$ | $.002 \pm .000$ | $.002 \pm .000$ | $0.359$ |
| Hopper-M-$10^4$ | $-.442 \pm .030$ | $.000 \pm .001$ | $.001 \pm .000$ | $.005 \pm .003$ | $.000 \pm .001$ | $.002 \pm .000$ | $.023 \pm .016$ | $0.344$ |
| Hopper-H-$10^2$ | $-.439 \pm .134$ | $.037 \pm .050$ | $.036 \pm .024$ | $.072 \pm .017$ | $.037 \pm .050$ | $.090 \pm .065$ | $.219 \pm .054$ | $0.402$ |
| Hopper-H-$10^3$ | $-.209 \pm .051$ | $.002 \pm .000$ | $.007 \pm .006$ | $.008 \pm .005$ | $.002 \pm .000$ | $.010 \pm .008$ | $.029 \pm .023$ | $0.387$ |
| Hopper-H-$10^4$ | $-.016 \pm .052$ | $.013 \pm .000$ | $.013 \pm .000$ | $.030 \pm .031$ | $.013 \pm .000$ | $.013 \pm .000$ | $.074 \pm .086$ | $0.409$ |
| Average | $-.195 \pm .296$ | $.190 \pm .264$ | $.185 \pm .222$ | $.162 \pm .177$ | $.190 \pm .264$ | $.265 \pm .288$ | $.305 \pm .289$ | $.420 \pm .080$ |

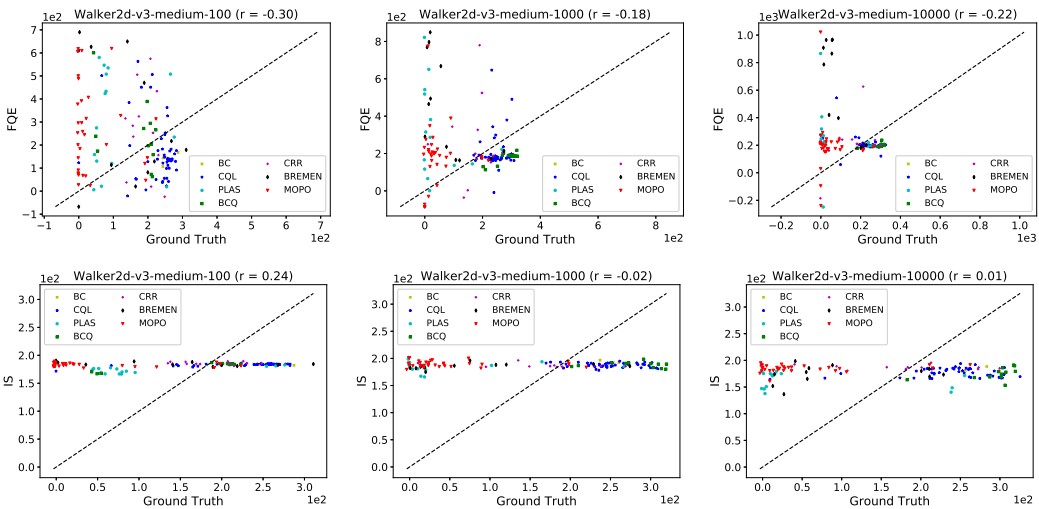

Figure 12: Scatter plot of OPE results for Walker2d-Medium tasks. $r$ stands for the correlation coefficient.

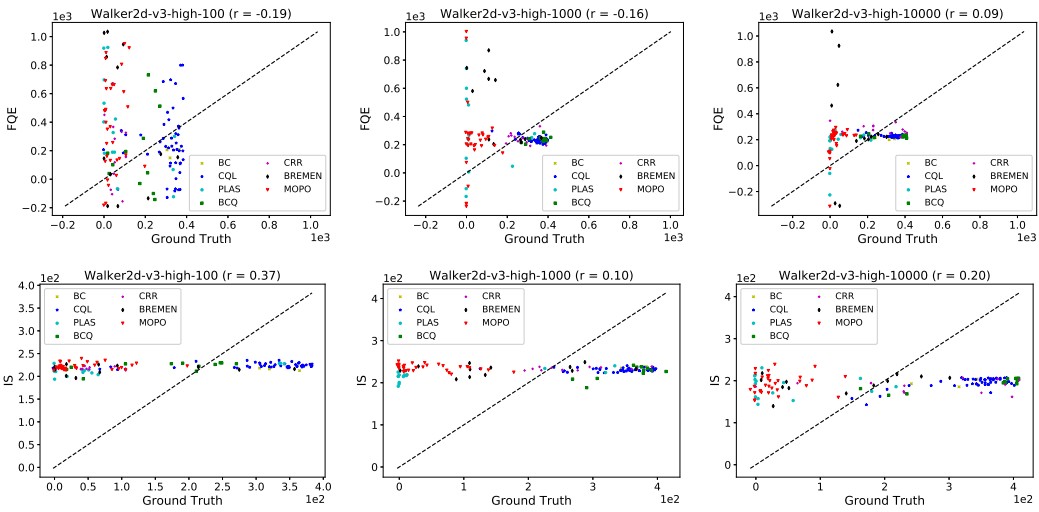

Figure 13: Scatter plot of OPE results for Walker2d-High tasks. $r$ stands for the correlation coefficient.

Table 13: FQE performance on the policies from Walker2d tasks. L, M, H stands for low, medium and high quality of dataset.

| Task | RC Score | Top-1 Mean Score | Top-3 Mean Score | Top-5 Mean Score | Top-1 Max Score | Top-3 Max Score | Top-5 Max Score | Policy Mean Score |
|---|---|---|---|---|---|---|---|---|
| Walker2d-L-$10^2$ | $-.287 \pm .020$ | $.072 \pm .057$ | $.127 \pm .013$ | $.126 \pm .013$ | $.072 \pm .057$ | $.182 \pm .023$ | $.212 \pm .045$ | 0.345 |
| Walker2d-L-$10^3$ | $.025 \pm .045$ | $.161 \pm .113$ | $.102 \pm .035$ | $.156 \pm .102$ | $.161 \pm .113$ | $.218 \pm .087$ | $.454 \pm .356$ | 0.418 |
| Walker2d-L-$10^4$ | $.267 \pm .136$ | $.035 \pm .018$ | $.036 \pm .008$ | $.040 \pm .007$ | $.035 \pm .018$ | $.063 \pm .014$ | $.073 \pm .011$ | 0.487 |
| Walker2d-M-$10^2$ | $-.220 \pm .037$ | $.262 \pm .183$ | $.239 \pm .115$ | $.245 \pm .088$ | $.262 \pm .183$ | $.461 \pm .124$ | $.535 \pm .063$ | 0.497 |
| Walker2d-M-$10^3$ | $-.036 \pm .044$ | $.044 \pm .027$ | $.133 \pm .140$ | $.215 \pm .146$ | $.044 \pm .027$ | $.292 \pm .325$ | $.562 \pm .213$ | 0.497 |
| Walker2d-M-$10^4$ | $-.101 \pm .130$ | $.107 \pm .073$ | $.155 \pm .043$ | $.143 \pm .030$ | $.107 \pm .073$ | $.249 \pm .043$ | $.249 \pm .043$ | 0.496 |
| Walker2d-H-$10^2$ | $-.306 \pm .124$ | $.051 \pm .000$ | $.093 \pm .054$ | $.129 \pm .039$ | $.051 \pm .000$ | $.188 \pm .097$ | $.275 \pm .035$ | 0.435 |
| Walker2d-H-$10^3$ | $-.171 \pm .052$ | $.031 \pm .034$ | $.052 \pm .049$ | $.106 \pm .035$ | $.031 \pm .034$ | $.145 \pm .147$ | $.322 \pm .037$ | 0.534 |
| Walker2d-H-$10^4$ | $.150 \pm .093$ | $.077 \pm .047$ | $.087 \pm .017$ | $.069 \pm .002$ | $.077 \pm .047$ | $.137 \pm .004$ | $.137 \pm .004$ | 0.516 |
| Average | $-.075 \pm .205$ | $.093 \pm .108$ | $.114 \pm .089$ | $.136 \pm .092$ | $.093 \pm .108$ | $.215 \pm .172$ | $.313 \pm .215$ | $.469 \pm .056$ |

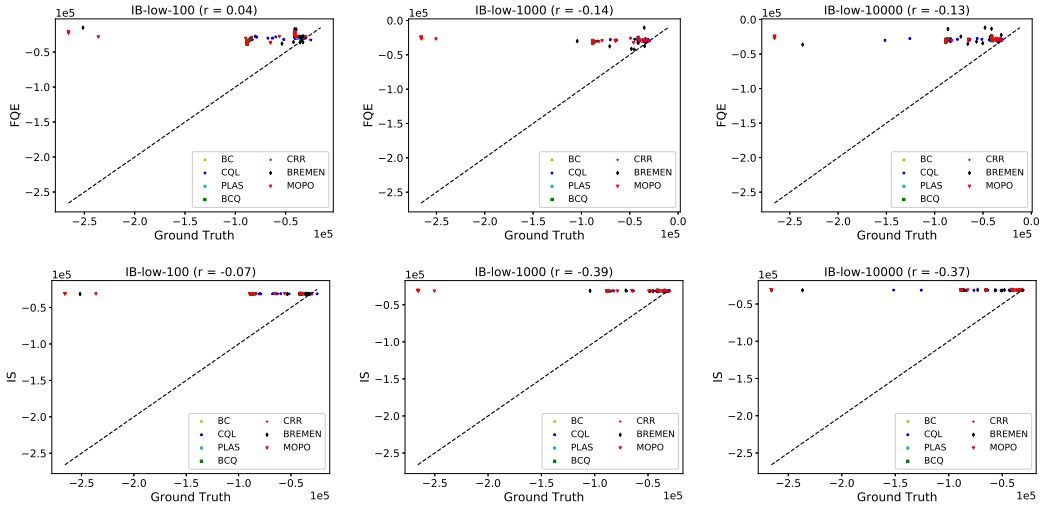

Figure 14: Scatter plot of OPE results for IB-Low tasks. $r$ stands for the correlation coefficient.

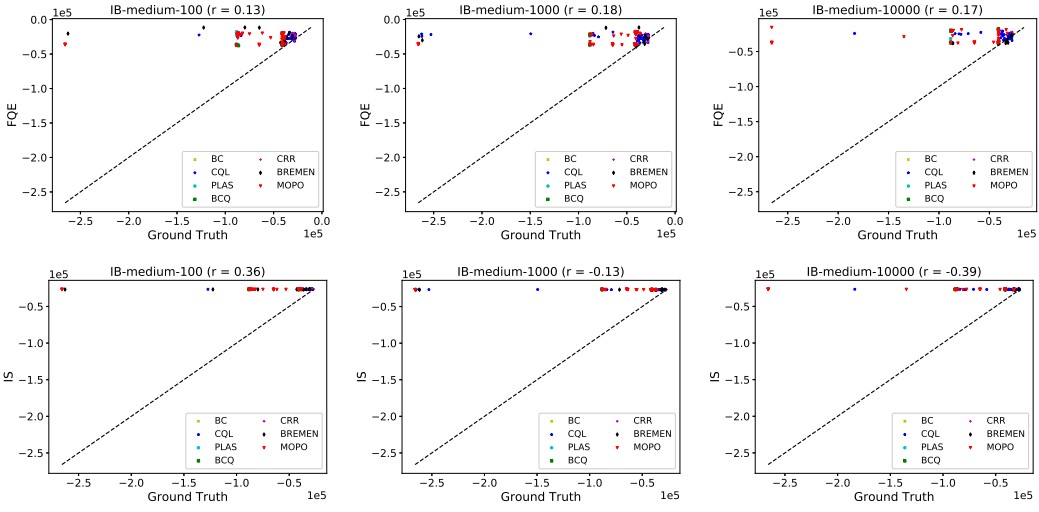

Figure 15: Scatter plot of OPE results for IB-Medium tasks. $r$ stands for the correlation coefficient.

Table 14: IS performance on the policies from Walker2d tasks. L, M, H stands for low, medium and high quality of dataset.

| Task | RC Score | Top-1 Mean Score | Top-3 Mean Score | Top-5 Mean Score | Top-1 Max Score | Top-3 Max Score | Top-5 Max Score | Policy Mean Score |
|---|---|---|---|---|---|---|---|---|
| Walker2d-L-$10^2$ | $.064 \pm .094$ | $.167 \pm .051$ | $.164 \pm .013$ | $.165 \pm .006$ | $.167 \pm .051$ | $.220 \pm .021$ | $.246 \pm .016$ | 0.345 |
| Walker2d-L-$10^3$ | $-.515 \pm .051$ | $.094 \pm .030$ | $.060 \pm .019$ | $.041 \pm .009$ | $.094 \pm .030$ | $.115 \pm .000$ | $.115 \pm .000$ | 0.418 |
| Walker2d-L-$10^4$ | $-.326 \pm .027$ | $.011 \pm .002$ | $.018 \pm .007$ | $.016 \pm .004$ | $.011 \pm .002$ | $.030 \pm .023$ | $.030 \pm .023$ | 0.487 |
| Walker2d-M-$10^2$ | $.161 \pm .166$ | $.020 \pm .018$ | $.256 \pm .111$ | $.361 \pm .092$ | $.020 \pm .018$ | $.571 \pm .186$ | $.734 \pm .078$ | 0.497 |
| Walker2d-M-$10^3$ | $-.021 \pm .038$ | $.009 \pm .001$ | $.009 \pm .000$ | $.072 \pm .048$ | $.009 \pm .001$ | $.010 \pm .000$ | $.306 \pm .245$ | 0.497 |
| Walker2d-M-$10^4$ | $-.036 \pm .036$ | $.298 \pm .229$ | $.450 \pm .250$ | $.334 \pm .134$ | $.298 \pm .229$ | $.752 \pm .161$ | $.790 \pm .147$ | 0.496 |
| Walker2d-H-$10^2$ | $.441 \pm .055$ | $.364 \pm .297$ | $.519 \pm .088$ | $.528 \pm .099$ | $.364 \pm .297$ | $.858 \pm .058$ | $.878 \pm .033$ | 0.435 |
| Walker2d-H-$10^3$ | $-.044 \pm .065$ | $.093 \pm .124$ | $.160 \pm .078$ | $.221 \pm .072$ | $.093 \pm .124$ | $.436 \pm .188$ | $.649 \pm .235$ | 0.534 |
| Walker2d-H-$10^4$ | $.215 \pm .070$ | $.117 \pm .092$ | $.108 \pm .041$ | $.101 \pm .006$ | $.117 \pm .092$ | $.191 \pm .070$ | $.241 \pm .000$ | 0.516 |
| Average | $-.007 \pm .279$ | $.130 \pm .182$ | $.194 \pm .199$ | $.204 \pm .177$ | $.130 \pm .182$ | $.354 \pm .316$ | $.443 \pm .326$ | $.469 \pm .056$ |

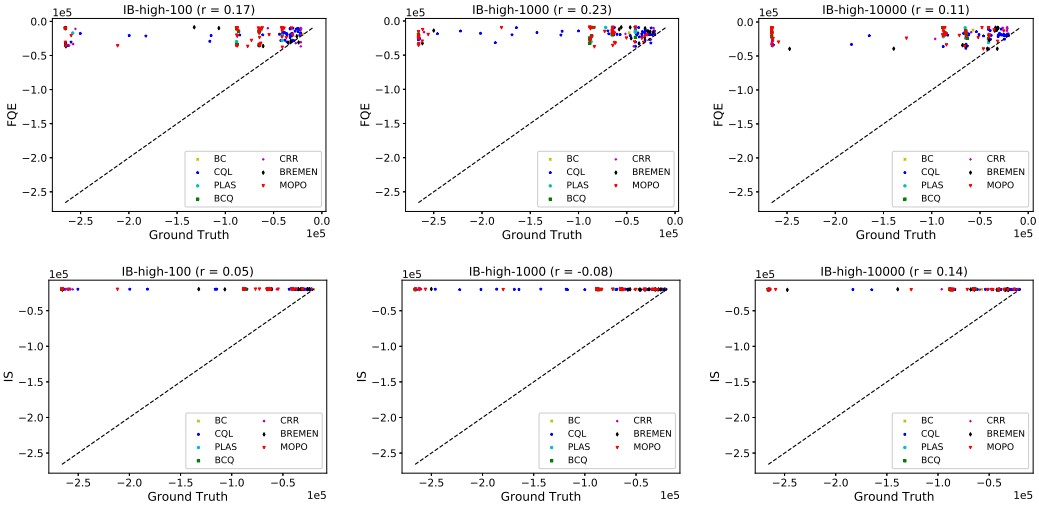

Figure 16: Scatter plot of OPE results for IB-High tasks. $r$ stands for the correlation coefficient.

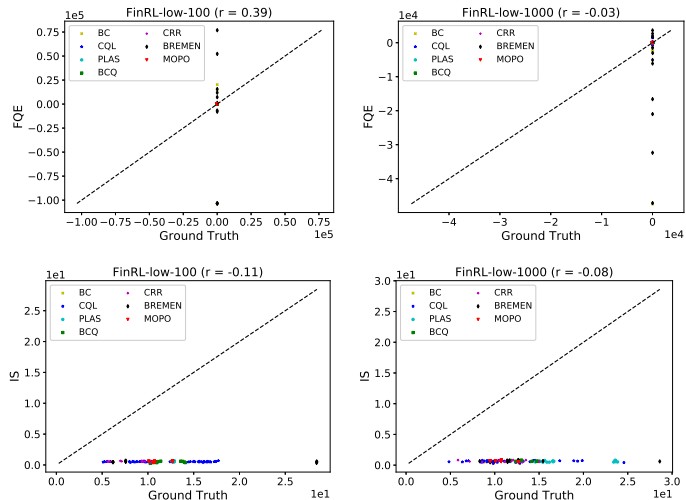

Figure 17: Scatter plot of OPE results for FinRL-Low tasks. $r$ stands for the correlation coefficient.

Table 15: FQE performance on the policies from IB tasks. L, M, H stands for low, medium and high quality of dataset.

| Task | RC Score | Top-1 Mean Score | Top-3 Mean Score | Top-5 Mean Score | Top-1 Max Score | Top-3 Max Score | Top-5 Max Score | Policy Mean Score |
|------|----------|------------------|------------------|------------------|-----------------|-----------------|-----------------|-------------------|
| IB-L-$10^2$ | $.282 \pm .027$ | $.060 \pm .000$ | $.645 \pm .000$ | $.511 \pm .088$ | $.060 \pm .000$ | $.940 \pm .000$ | $.940 \pm .000$ | 0.847 |
| IB-L-$10^3$ | $-.013 \pm .121$ | $.967 \pm .012$ | $.322 \pm .004$ | $.320 \pm .088$ | $.967 \pm .012$ | $.967 \pm .012$ | $.967 \pm .012$ | 0.862 |
| IB-L-$10^4$ | $-.136 \pm .091$ | $.935 \pm .014$ | $.895 \pm .020$ | $.802 \pm .176$ | $.935 \pm .014$ | $.968 \pm .020$ | $.982 \pm .020$ | 0.850 |
| IB-M-$10^2$ | $.170 \pm .047$ | $.781 \pm .000$ | $.834 \pm .067$ | $.828 \pm .038$ | $.781 \pm .000$ | $.922 \pm .057$ | $.966 \pm .012$ | 0.873 |
| IB-M-$10^3$ | $.182 \pm .009$ | $.863 \pm .067$ | $.902 \pm .002$ | $.919 \pm .001$ | $.863 \pm .067$ | $.948 \pm .007$ | $.953 \pm .007$ | 0.842 |
| IB-M-$10^4$ | $.243 \pm .015$ | $.000 \pm .000$ | $.632 \pm .005$ | $.756 \pm .003$ | $.000 \pm .000$ | $.953 \pm .015$ | $.953 \pm .015$ | 0.881 |
| IB-H-$10^2$ | $.098 \pm .015$ | $.452 \pm .303$ | $.708 \pm .106$ | $.640 \pm .037$ | $.452 \pm .303$ | $.914 \pm .000$ | $.926 \pm .017$ | 0.715 |
| IB-H-$10^3$ | $.102 \pm .034$ | $.879 \pm .058$ | $.871 \pm .031$ | $.855 \pm .052$ | $.879 \pm .058$ | $.911 \pm .041$ | $.911 \pm .042$ | 0.732 |
| IB-H-$10^4$ | $.007 \pm .025$ | $.889 \pm .119$ | $.928 \pm .034$ | $.858 \pm .034$ | $.889 \pm .119$ | $.982 \pm .009$ | $.982 \pm .009$ | 0.694 |
| Average | $.104 \pm .138$ | $.647 \pm .377$ | $.749 \pm .190$ | $.721 \pm .200$ | $.647 \pm .377$ | $.945 \pm .035$ | $.953 \pm .029$ | $.811 \pm .070$ |

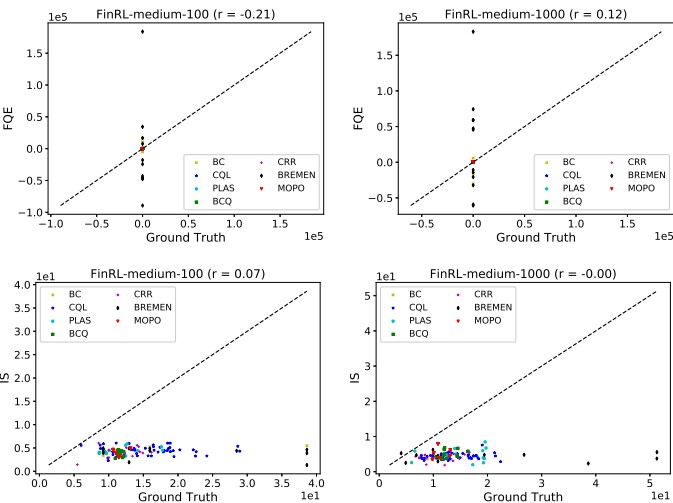

Figure 18: Scatter plot of OPE results for FinRL-Medium tasks. $r$ stands for the correlation coefficient.

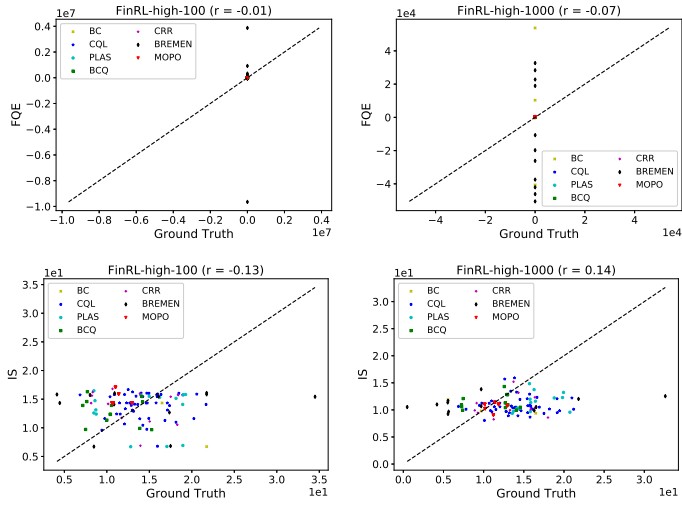

Figure 19: Scatter plot of OPE results for FinRL-High tasks. $r$ stands for the correlation coefficient.

Table 16: IS performance on the policies from IB tasks. L, M, H stands for low, medium and high quality of dataset.

| Task | RC Score | Top-1 Mean Score | Top-3 Mean Score | Top-5 Mean Score | Top-1 Max Score | Top-3 Max Score | Top-5 Max Score | Policy Mean Score |
|---|---|---|---|---|---|---|---|---|
| IB-L-$10^2$ | $-.375 \pm .129$ | $.867 \pm .091$ | $.837 \pm .045$ | $.843 \pm .048$ | $.867 \pm .091$ | $.932 \pm .002$ | $.934 \pm .005$ | 0.847 |
| IB-L-$10^3$ | $-.519 \pm .120$ | $.317 \pm .447$ | $.317 \pm .447$ | $.308 \pm .436$ | $.317 \pm .447$ | $.317 \pm .447$ | $.317 \pm .447$ | 0.862 |
| IB-L-$10^4$ | $-.375 \pm .016$ | $.980 \pm .018$ | $.861 \pm .150$ | $.771 \pm .270$ | $.980 \pm .018$ | $.993 \pm .000$ | $.993 \pm .000$ | 0.850 |
| IB-M-$10^2$ | $.195 \pm .294$ | $.962 \pm .025$ | $.963 \pm .015$ | $.908 \pm .103$ | $.962 \pm .025$ | $.980 \pm .026$ | $.995 \pm .005$ | 0.873 |
| IB-M-$10^3$ | $-.250 \pm .045$ | $.877 \pm .094$ | $.888 \pm .041$ | $.885 \pm .043$ | $.877 \pm .094$ | $.944 \pm .001$ | $.944 \pm .000$ | 0.842 |
| IB-M-$10^4$ | $-.341 \pm .036$ | $.251 \pm .355$ | $.571 \pm .005$ | $.722 \pm .002$ | $.251 \pm .355$ | $.960 \pm .014$ | $.972 \pm .008$ | 0.881 |
| IB-H-$10^2$ | $.053 \pm .099$ | $.820 \pm .004$ | $.789 \pm .122$ | $.861 \pm .067$ | $.820 \pm .004$ | $.993 \pm .000$ | $.993 \pm .000$ | 0.715 |
| IB-H-$10^3$ | $-.170 \pm .018$ | $.822 \pm .002$ | $.810 \pm .014$ | $.814 \pm .008$ | $.822 \pm .002$ | $.822 \pm .002$ | $.822 \pm .002$ | 0.732 |
| IB-H-$10^4$ | $.097 \pm .006$ | $.819 \pm .000$ | $.819 \pm .000$ | $.819 \pm .000$ | $.819 \pm .000$ | $.819 \pm .000$ | $.819 \pm .000$ | 0.694 |
| Average | $-.187 \pm .263$ | $.746 \pm .320$ | $.762 \pm .248$ | $.770 \pm .247$ | $.746 \pm .320$ | $.862 \pm .252$ | $.866 \pm .253$ | $.811 \pm .070$ |

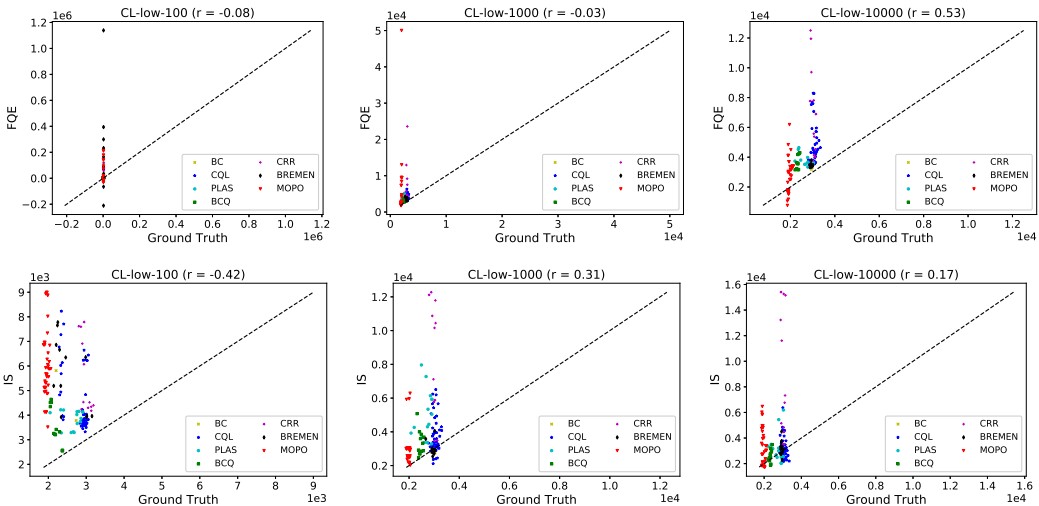

Figure 20: Scatter plot of OPE results for CL-Low tasks. $r$ stands for the correlation coefficient.

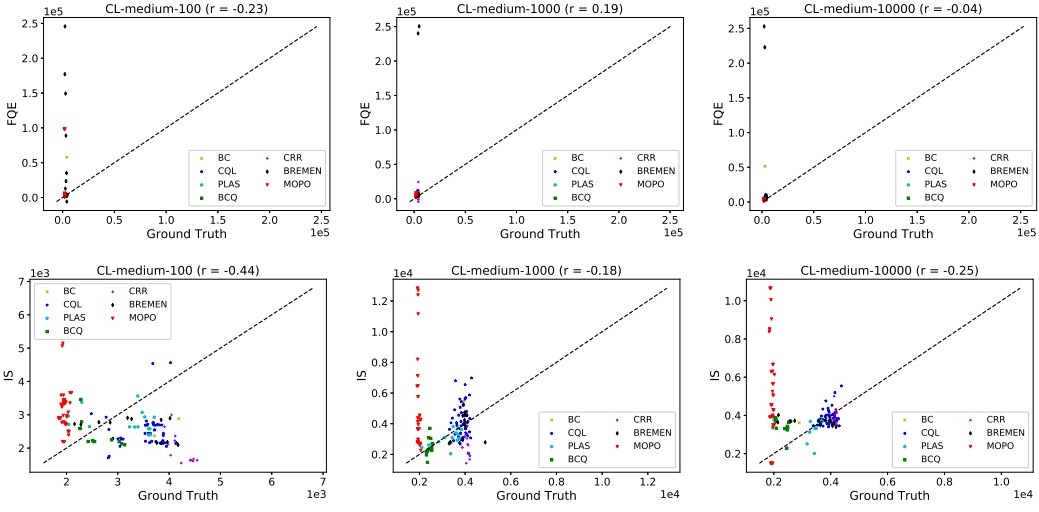

Figure 21: Scatter plot of OPE results for CL-Medium tasks. $r$ stands for the correlation coefficient.

Table 17: FQE performance on the policies from FinRL tasks. L, M, H stands for low, medium and high quality of dataset.

| Task | RC Score | Top-1 Mean Score | Top-3 Mean Score | Top-5 Mean Score | Top-1 Max Score | Top-3 Max Score | Top-5 Max Score | Policy Mean Score |
|------|----------|------------------|------------------|------------------|-----------------|-----------------|-----------------|-------------------|
| FinRL-L-$10^2$ | $-.012 \pm .080$ | $.701 \pm .421$ | $.800 \pm .140$ | $.510 \pm .093$ | $.701 \pm .421$ | $.999 \pm .001$ | $.999 \pm .001$ | 0.285 |
| FinRL-L-$10^3$ | $-.015 \pm .030$ | $.209 \pm .010$ | $.266 \pm .040$ | $.353 \pm .063$ | $.209 \pm .010$ | $.349 \pm .071$ | $.685 \pm .232$ | 0.313 |
| FinRL-M-$10^2$ | $-.042 \pm .058$ | $.112 \pm .000$ | $.257 \pm .084$ | $.416 \pm .044$ | $.112 \pm .000$ | $.442 \pm .178$ | $1.000 \pm .000$ | 0.248 |
| FinRL-M-$10^3$ | $-.005 \pm .068$ | $.400 \pm .246$ | $.385 \pm .043$ | $.377 \pm .121$ | $.400 \pm .246$ | $.821 \pm .126$ | $.911 \pm .126$ | 0.195 |
| FinRL-H-$10^2$ | $-.056 \pm .121$ | $.165 \pm .042$ | $.108 \pm .012$ | $.192 \pm .029$ | $.165 \pm .042$ | $.196 \pm .037$ | $.484 \pm .068$ | 0.291 |
| FinRL-H-$10^3$ | $-.042 \pm .149$ | $.385 \pm .160$ | $.337 \pm .168$ | $.387 \pm .126$ | $.385 \pm .160$ | $.440 \pm .210$ | $.721 \pm .208$ | 0.384 |
| Average | $-.029 \pm .095$ | $.329 \pm .289$ | $.359 \pm .237$ | $.372 \pm .129$ | $.329 \pm .289$ | $.541 \pm .306$ | $.800 \pm .234$ | $.286 \pm .058$ |

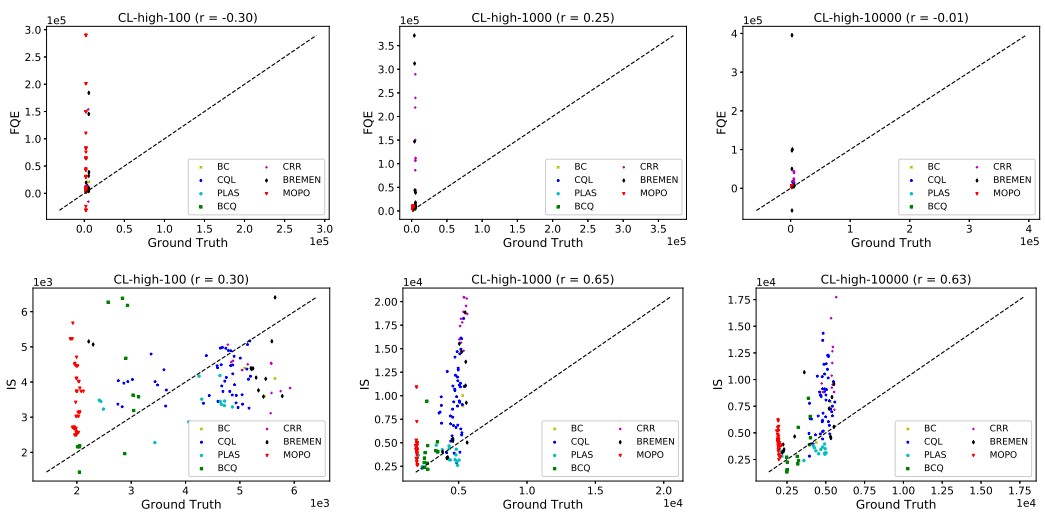

Figure 22: Scatter plot of OPE results for CL-High tasks. $r$ stands for the correlation coefficient.

Table 18: IS performance on the policies from FinRL tasks. L, M, H stands for low, medium and high quality of dataset.

| Task | RC Score | Top-1 Mean Score | Top-3 Mean Score | Top-5 Mean Score | Top-1 Max Score | Top-3 Max Score | Top-5 Max Score | Policy Mean Score |
|---|---|---|---|---|---|---|---|---|
| FinRL-L-$10^2$ | $.066 \pm .109$ | $.267 \pm .147$ | $.264 \pm .022$ | $.247 \pm .027$ | $.267 \pm .147$ | $.489 \pm .035$ | $.489 \pm .035$ | 0.285 |
| FinRL-L-$10^3$ | $-.041 \pm .032$ | $.333 \pm .047$ | $.361 \pm .035$ | $.301 \pm .018$ | $.333 \pm .047$ | $.406 \pm .029$ | $.406 \pm .029$ | 0.313 |
| FinRL-M-$10^2$ | $.103 \pm .112$ | $.162 \pm .051$ | $.263 \pm .089$ | $.279 \pm .090$ | $.162 \pm .051$ | $.426 \pm .210$ | $.428 \pm .212$ | 0.248 |
| FinRL-M-$10^3$ | $.056 \pm .077$ | $.537 \pm .328$ | $.297 \pm .097$ | $.289 \pm .080$ | $.537 \pm .328$ | $.548 \pm .319$ | $.687 \pm .276$ | 0.195 |
| FinRL-H-$10^2$ | $-.046 \pm .065$ | $.283 \pm .042$ | $.286 \pm .067$ | $.304 \pm .077$ | $.283 \pm .042$ | $.348 \pm .108$ | $.411 \pm .144$ | 0.291 |
| FinRL-H-$10^3$ | $.111 \pm .027$ | $.458 \pm .080$ | $.398 \pm .007$ | $.404 \pm .011$ | $.458 \pm .080$ | $.458 \pm .080$ | $.477 \pm .074$ | 0.384 |
| Average | $.042 \pm .100$ | $.340 \pm .198$ | $.312 \pm .081$ | $.304 \pm .077$ | $.340 \pm .198$ | $.446 \pm .178$ | $.483 \pm .185$ | $.286 \pm .058$ |

Table 19: FQE performance on the policies from CL tasks. L, M, H stands for low, medium and high quality of dataset.

| Task | RC Score | Top-1 Mean Score | Top-3 Mean Score | Top-5 Mean Score | Top-1 Max Score | Top-3 Max Score | Top-5 Max Score | Policy Mean Score |
|---|---|---|---|---|---|---|---|---|
| CL-L-$10^2$ | $.144 \pm .056$ | $.270 \pm .012$ | $.296 \pm .028$ | $.294 \pm .024$ | $.270 \pm .012$ | $.358 \pm .065$ | $.411 \pm .047$ | 0.443 |
| CL-L-$10^3$ | $.479 \pm .013$ | $.511 \pm .303$ | $.533 \pm .163$ | $.476 \pm .091$ | $.511 \pm .303$ | $.807 \pm .014$ | $.807 \pm .014$ | 0.504 |
| CL-L-$10^4$ | $.641 \pm .048$ | $.710 \pm .005$ | $.738 \pm .019$ | $.750 \pm .009$ | $.710 \pm .005$ | $.798 \pm .056$ | $.847 \pm .027$ | 0.494 |
| CL-M-$10^2$ | $.288 \pm .250$ | $.231 \pm .100$ | $.242 \pm .058$ | $.183 \pm .088$ | $.231 \pm .100$ | $.396 \pm .098$ | $.396 \pm .098$ | 0.414 |
| CL-M-$10^3$ | $.429 \pm .038$ | $.780 \pm .155$ | $.812 \pm .004$ | $.725 \pm .018$ | $.780 \pm .155$ | $1.000 \pm .000$ | $1.000 \pm .000$ | 0.405 |
| CL-M-$10^4$ | $.638 \pm .031$ | $.220 \pm .138$ | $.297 \pm .002$ | $.440 \pm .068$ | $.220 \pm .138$ | $.414 \pm .000$ | $.798 \pm .076$ | 0.486 |
| CL-H-$10^2$ | $-.116 \pm .145$ | $.626 \pm .422$ | $.508 \pm .347$ | $.486 \pm .345$ | $.626 \pm .422$ | $.627 \pm .420$ | $.645 \pm .428$ | 0.423 |
| CL-H-$10^3$ | $.584 \pm .029$ | $.621 \pm .082$ | $.697 \pm .065$ | $.771 \pm .044$ | $.621 \pm .082$ | $.843 \pm .088$ | $.907 \pm .027$ | 0.487 |
| CL-H-$10^4$ | $.618 \pm .044$ | $.115 \pm .006$ | $.221 \pm .116$ | $.405 \pm .065$ | $.115 \pm .006$ | $.450 \pm .352$ | $.979 \pm .030$ | 0.483 |
| Average | $.412 \pm .267$ | $.454 \pm .301$ | $.483 \pm .256$ | $.503 \pm .233$ | $.454 \pm .301$ | $.633 \pm .293$ | $.754 \pm .260$ | $.460 \pm .036$ |

Table 20: IS performance on the policies from CL tasks. L, M, H stands for low, medium and high quality of dataset.

| Task | RC Score | Top-1 Mean Score | Top-3 Mean Score | Top-5 Mean Score | Top-1 Max Score | Top-3 Max Score | Top-5 Max Score | Policy Mean Score |
|---|---|---|---|---|---|---|---|---|
| CL-L-$10^2$ | $-.341 \pm .128$ | $.079 \pm .015$ | $.068 \pm .011$ | $.072 \pm .007$ | $.079 \pm .015$ | $.083 \pm .009$ | $.092 \pm .006$ | 0.443 |
| CL-L-$10^3$ | $.292 \pm .081$ | $.563 \pm .113$ | $.575 \pm .158$ | $.653 \pm .097$ | $.563 \pm .113$ | $.705 \pm .110$ | $.815 \pm .013$ | 0.504 |
| CL-L-$10^4$ | $.301 \pm .108$ | $.757 \pm .077$ | $.775 \pm .010$ | $.767 \pm .011$ | $.757 \pm .077$ | $.866 \pm .000$ | $.866 \pm .000$ | 0.494 |
| CL-M-$10^2$ | $-.284 \pm .225$ | $.269 \pm .172$ | $.199 \pm .066$ | $.275 \pm .073$ | $.269 \pm .172$ | $.344 \pm .075$ | $.503 \pm .227$ | 0.414 |
| CL-M-$10^3$ | $.079 \pm .091$ | $.014 \pm .004$ | $.014 \pm .002$ | $.014 \pm .001$ | $.014 \pm .004$ | $.017 \pm .000$ | $.017 \pm .000$ | 0.405 |
| CL-M-$10^4$ | $.108 \pm .316$ | $.260 \pm .341$ | $.289 \pm .386$ | $.380 \pm .326$ | $.260 \pm .341$ | $.337 \pm .448$ | $.592 \pm .405$ | 0.486 |
| CL-H-$10^2$ | $.217 \pm .078$ | $.414 \pm .366$ | $.331 \pm .180$ | $.301 \pm .153$ | $.414 \pm .366$ | $.595 \pm .316$ | $.679 \pm .298$ | 0.423 |
| CL-H-$10^3$ | $.615 \pm .066$ | $.883 \pm .035$ | $.914 \pm .042$ | $.912 \pm .037$ | $.883 \pm .035$ | $.969 \pm .029$ | $.969 \pm .029$ | 0.487 |
| CL-H-$10^4$ | $.678 \pm .096$ | $.943 \pm .055$ | $.872 \pm .020$ | $.863 \pm .019$ | $.943 \pm .055$ | $.958 \pm .035$ | $.958 \pm .035$ | 0.483 |
| Average | $.185 \pm .362$ | $.465 \pm .371$ | $.448 \pm .360$ | $.471 \pm .343$ | $.465 \pm .371$ | $.542 \pm .391$ | $.610 \pm .380$ | $.460 \pm .036$ |

# F   Additional Tables

In this section, we provide the winning rates table, raw and normalized scores that are not fitted in the main paper.

Table 21: Ratio of winning the 3 baselines over the 51 tasks by online evaluation.

| Baseline | BCQ | PLAS | CQL | CRR | BREMEN | MOPO |
|---|---|---|---|---|---|---|
| Det. Policy | 35.3% | 43.1% | 86.3% | 64.7% | 41.2% | 13.7% |
| Behavior Policy | 41.2% | 52.9% | 92.2% | 74.5% | 49.0% | 15.7% |
| BC | 41.2% | 45.1% | 88.2% | 70.6% | 39.2% | 15.7% |

Table 22: Ratio of winning the 3 baselines over the 51 tasks by FQE evaluation.

| Baseline | BCQ | PLAS | CQL | CRR | BREMEN | MOPO |
|---|---|---|---|---|---|---|
| Det. Policy | 15.7% | 11.8% | 31.4% | 39.2% | 23.5% | 11.8% |
| Behavior Policy | 21.6% | 13.7% | 43.1% | 41.2% | 21.6% | 7.8% |
| BC | 11.8% | 15.7% | 41.2% | 39.2% | 15.7% | 7.8% |

Table 23: Ratio of winning the 3 baselines over the 51 tasks by IS evaluation.

| Baseline | BCQ | PLAS | CQL | CRR | BREMEN | MOPO |
|---|---|---|---|---|---|---|
| Det. Policy | 23.5% | 29.4% | 39.2% | 49.0% | 19.6% | 15.7% |
| Behavior Policy | 27.5% | 31.4% | 56.9% | 52.9% | 17.6% | 13.7% |
| BC | 25.5% | 27.5% | 39.2% | 47.1% | 19.6% | 13.7% |

Table 24: Normalized score for HalfCheetah tasks. For each task, three lines indicate the results of online evaluation, FQE evaluation, WIS evaluation respectively. Bold numbers indicate the best result for each task, while numbers marked by * indicate results worse than BC. The task name is composed of the specific task, the quality of dataset, and the number of trajectories. L, M, and H stands for low, medium, high respectively. Det. is abbreviation of deterministic.

| Task Name | Expert Policy | Det. Policy | Behavior Policy | Random | BC | BCQ | PLAS | CQL | CRR | BREMEN | MOPO |
|---|---|---|---|---|---|---|---|---|---|---|---|
| HalfCheetah-L-$10^2$ | 100 | 27 | 25 | 0 | 29.1 +- 0.3 | 30.2 +- 0.3 | 28.8 +- 0.4* | 32.6 +- 0.3 | 29.0 +- 0.2* | 37.6 +- 1.8 | **42.0 +- 1.8** |
| | | | | | 28.9 +- 0.3 | 29.6 +- 0.0 | 25.5 +- 3.5* | 31.5 +- 0.8 | 29.0 +- 0.0 | **36.5 +- 0.3** | 36.3 +- 1.9 |
| | | | | | 29.4 +- 0.0 | 26.6 +- 1.6* | 28.0 +- 0.2* | 30.2 +- 1.6 | 28.0 +- 0.7* | 34.3 +- 3.4 | **35.3 +- 4.3** |
| HalfCheetah-L-$10^3$ | 100 | 27 | 25 | 0 | 29.1 +- 0.2 | 34.1 +- 0.4 | 30.6 +- 0.0 | 38.2 +- 0.5 | 29.2 +- 0.2 | 39.6 +- 1.8 | **40.1 +- 0.9** |
| | | | | | 29.0 +- 0.2 | 34.4 +- 0.0 | 30.5 +- 0.0 | **36.6 +- 0.7** | 28.6 +- 0.6* | 23.6 +- 0.0* | 24.9 +- 19.3* |
| | | | | | 29.1 +- 0.2 | 31.7 +- 0.2 | 29.3 +- 0.0 | 27.3 +- 3.5* | 29.0 +- 0.3* | 22.2 +- 0.3* | **33.5 +- 2.0** |
| HalfCheetah-L-$10^4$ | 100 | 27 | 25 | 0 | 28.9 +- 0.1 | 36.7 +- 0.7 | 30.6 +- 0.2 | **39.8 +- 1.4** | 29.3 +- 0.5 | 39.1 +- 0.3 | 37.7 +- 0.3 |
| | | | | | 29.0 +- 0.0 | 35.7 +- 1.1 | 30.1 +- 0.5 | **39.0 +- 1.2** | 28.9 +- 0.2* | 38.9 +- 0.0 | 24.0 +- 18.7* |
| | | | | | 28.8 +- 0.1 | **32.1 +- 0.7** | 29.8 +- 0.7 | 26.3 +- 4.3* | 29.2 +- 0.3 | 19.4 +- 3.4* | -2.4 +- 0.0* |
| HalfCheetah-M-$10^2$ | 100 | 50 | 46 | 0 | 48.9 +- 0.8 | 43.2 +- 1.5* | 46.7 +- 1.0* | 51.6 +- 0.4 | 27.2 +- 0.6* | 52.3 +- 5.0 | **63.1 +- 0.5** |
| | | | | | 48.3 +- 0.0 | 12.0 +- 7.9* | 34.2 +- 0.0* | 24.8 +- 3.6* | 17.2 +- 1.3* | 47.1 +- 0.0* | **52.1 +- 0.4** |
| | | | | | 49.5 +- 0.7 | 42.9 +- 1.2* | 45.8 +- 2.2* | 40.2 +- 11.4* | 24.7 +- 4.7* | 43.8 +- 11.2* | **51.9 +- 1.3** |
| HalfCheetah-M-$10^3$ | 100 | 50 | 46 | 0 | 49.0 +- 0.6 | 50.6 +- 0.1 | 50.8 +- 0.4 | 54.6 +- 0.3 | 43.2 +- 2.6* | 55.4 +- 3.0 | **62.3 +- 1.1** |
| | | | | | 49.5 +- 0.0 | 42.4 +- 5.3* | 28.4 +- 0.0* | 19.4 +- 0.0* | 27.2 +- 6.7* | **57.0 +- 0.0** | 36.9 +- 27.8* |
| | | | | | 48.9 +- 0.5 | 45.1 +- 7.2* | 50.9 +- 0.4 | 48.7 +- 3.6* | 21.2 +- 0.2* | 34.8 +- 15.1* | **55.4 +- 1.7** |
| HalfCheetah-M-$10^4$ | 100 | 50 | 46 | 0 | 50.0 +- 0.4 | 49.6 +- 0.9* | 50.8 +- 0.2 | **55.8 +- 0.9** | 44.0 +- 1.7* | 55.8 +- 3.2 | 43.7 +- 0.9* |
| | | | | | 49.9 +- 0.0 | 31.0 +- 1.1* | 33.7 +- 6.2* | **55.2 +- 1.6** | 25.5 +- 0.4* | 46.0 +- 9.7* | 45.2 +- 0.8* |
| | | | | | 49.6 +- 0.0 | 41.4 +- 8.6* | 50.9 +- 0.0 | 44.1 +- 3.2* | 42.0 +- 0.0* | **55.7 +- 0.0** | -2.3 +- 0.0* |
| HalfCheetah-H-$10^2$ | 100 | 74 | 64 | 0 | 47.2 +- 31.8 | 57.6 +- 3.1 | 64.2 +- 0.7 | **74.0 +- 1.5** | 24.0 +- 1.6* | 29.0 +- 22.7* | 47.8 +- 8.2 |
| | | | | | **69.6 +- 0.0** | 47.9 +- 0.0* | 16.4 +- 3.3* | 1.6 +- 0.4* | 8.9 +- 7.8* | 47.2 +- 0.0* | 4.2 +- 0.8* |
| | | | | | **69.7 +- 0.1** | 28.6 +- 20.0* | 43.9 +- 21.5* | 21.9 +- 27.9* | 16.4 +- 5.6* | 26.5 +- 18.7* | 23.9 +- 18.4* |
| HalfCheetah-H-$10^3$ | 100 | 74 | 64 | 0 | 71.3 +- 0.5 | 72.4 +- 0.3 | 74.1 +- 0.8 | **77.4 +- 1.3** | 62.5 +- 1.9* | 54.8 +- 17.1* | 65.9 +- 10.3* |
| | | | | | **71.7 +- 0.1** | 17.7 +- 0.0* | 31.0 +- 1.9* | 0.2 +- 0.1* | 9.3 +- 0.0* | 59.0 +- 8.3* | 3.5 +- 4.2* |
| | | | | | 71.7 +- 0.0 | 67.2 +- 3.4* | **74.6 +- 0.8** | 2.3 +- 1.0* | | 29.4 +- 2.5* | 11.0 +- 2.4* |
| HalfCheetah-H-$10^4$ | 100 | 74 | 64 | 0 | 66.7 +- 2.7 | 73.3 +- 1.4 | 75.4 +- 0.6 | **77.2 +- 0.9** | 69.6 +- 0.4 | 15.7 +- 2.8* | 7.6 +- 6.3* |
| | | | | | **69.0 +- 0.0** | 24.5 +- 0.0* | 18.8 +- 4.7* | 1.3 +- 0.0* | 25.4 +- 0.5* | 26.3 +- 0.0* | 1.2 +- 0.0* |
| | | | | | 68.4 +- 0.0 | 52.2 +- 21.5* | **74.7 +- 0.0** | 70.1 +- 3.5 | 69.5 +- 0.4 | 11.7 +- 3.3* | -2.4 +- 0.0* |

Table 25: Normalized score for Hopper tasks. For each task, three lines indicate the results of online evaluation, FQE evaluation, WIS evaluation respectively. Bold numbers indicate the best result for each task, while numbers marked by * indicate results worse than BC. The task name is composed of the specific task, the quality of dataset, and the number of trajectories. L, M, and H stands for low, medium, high respectively. Det. is abbreviation of deterministic.

| Task Name | Expert Policy | Det. Policy | Behavior Policy | Random | BC | BCQ | PLAS | CQL | CRR | BREMEN | MOPO |
|---|---|---|---|---|---|---|---|---|---|---|---|
| Hopper-L-$10^2$ | 100 | 15 | 15 | 0 | 16.1 +- 0.6 | 15.3 +- 0.3* | 15.6 +- 0.3* | **16.5 +- 0.5** | 16.4 +- 1.3 | 15.4 +- 0.9* | 5.0 +- 6.1* |
| | | | | | **16.1 +- 0.6** | 11.1 +- 2.2* | 7.0 +- 5.8* | 15.0 +- 0.3* | 15.7 +- 0.0* | 10.7 +- 6.3* | 1.4 +- 0.1* |
| | | | | | **16.1 +- 0.6** | 12.1 +- 0.0* | 9.8 +- 2.9* | 15.9 +- 0.8* | 16.0 +- 0.2* | 1.1 +- 0.0* | 1.0 +- 0.7* |
| Hopper-L-$10^3$ | 100 | 15 | 15 | 0 | 15.1 +- 0.7 | 18.1 +- 0.2 | 19.3 +- 1.6 | 16.0 +- 0.1 | 16.8 +- 0.6 | **21.4 +- 7.6** | 6.2 +- 3.1* |
| | | | | | 15.1 +- 0.7 | 14.9 +- 3.8* | **18.2 +- 1.6** | 15.4 +- 0.2 | 17.1 +- 2.2 | 2.1 +- 0.4* | 0.5 +- 0.9* |
| | | | | | 14.6 +- 0.6 | 17.4 +- 1.0 | 16.1 +- 2.6 | 15.6 +- 0.4 | **17.5 +- 1.9** | 1.3 +- 0.2* | 3.7 +- 0.0* |
| Hopper-L-$10^4$ | 100 | 15 | 15 | 0 | 15.5 +- 0.3 | 18.7 +- 1.4 | 17.4 +- 1.5 | **20.9 +- 4.3** | 15.7 +- 0.0 | 15.3 +- 1.5* | 7.4 +- 2.3* |
| | | | | | 15.6 +- 0.4 | 14.4 +- 2.9* | 15.0 +- 1.1* | 15.1 +- 0.7* | **17.5 +- 0.6** | 10.4 +- 8.4* | 0.3 +- 0.5* |
| | | | | | 15.7 +- 0.3 | **17.3 +- 1.9** | 16.5 +- 0.0 | 14.2 +- 0.1* | 16.2 +- 1.5 | 8.2 +- 0.0* | 0.6 +- 0.6* |
| Hopper-M-$10^2$ | 100 | 46 | 42 | 0 | 28.0 +- 11.4 | 40.9 +- 1.5 | 50.0 +- 3.4 | **63.2 +- 9.4** | 41.5 +- 9.8 | 28.5 +- 6.3 | 1.8 +- 2.6* |
| | | | | | 28.7 +- 10.9 | 21.0 +- 15.6* | 30.6 +- 7.0 | **43.0 +- 8.4** | 29.8 +- 1.1 | 6.3 +- 4.2* | 1.0 +- 0.7* |
| | | | | | 36.4 +- 10.9 | 29.1 +- 14.2* | 30.6 +- 7.0* | **69.8 +- 8.2** | 36.4 +- 2.0 | 5.9 +- 2.4* | 2.3 +- 2.3* |
| Hopper-M-$10^3$ | 100 | 46 | 42 | 0 | 51.3 +- 27.2 | 47.7 +- 11.1* | 61.2 +- 25.8 | **64.5 +- 7.0** | 42.7 +- 5.5* | 24.7 +- 5.5* | 1.0 +- 1.5* |
| | | | | | **71.1 +- 26.2** | 33.0 +- 13.8* | 32.3 +- 6.8* | 57.3 +- 1.4* | 38.8 +- 15.1* | 21.3 +- 0.0* | -0.1 +- 0.1* |
| | | | | | 30.2 +- 0.0 | 33.3 +- 7.7 | 28.0 +- 29.0* | **53.5 +- 0.2** | 42.1 +- 11.9 | 21.3 +- 0.0* | -0.0 +- 0.0* |
| Hopper-M-$10^4$ | 100 | 46 | 42 | 0 | 54.4 +- 14.8 | 56.6 +- 7.8 | 62.9 +- 17.0 | **81.6 +- 13.1** | 49.1 +- 2.2* | 46.1 +- 14.1* | 1.1 +- 0.9* |
| | | | | | **56.8 +- 0.0** | 29.8 +- 3.2* | 14.3 +- 0.0* | 43.7 +- 6.5* | 35.1 +- 19.5* | 15.0 +- 2.8* | -0.1 +- 0.0* |
| | | | | | **61.6 +- 6.8** | 30.9 +- 0.9* | 7.3 +- 4.9* | 40.8 +- 2.5* | 4.6 +- 0.0* | 16.6 +- 3.9* | -0.1 +- 0.1* |
| Hopper-H-$10^2$ | 100 | 69 | 47 | 0 | 44.4 +- 12.4 | 35.7 +- 6.5* | 57.4 +- 6.9 | **69.7 +- 8.6** | 65.6 +- 12.8 | 28.5 +- 11.6* | 7.6 +- 8.4* |
| | | | | | 39.0 +- 14.3 | 13.3 +- 14.5* | 10.8 +- 2.8* | **44.1 +- 15.1** | 42.4 +- 0.6 | 0.0 +- 0.0* | -0.0 +- 0.1* |
| | | | | | 29.0 +- 0.0 | 8.6 +- 0.0* | 14.3 +- 6.8* | **46.0 +- 11.6** | 38.4 +- 16.3 | 0.0 +- 0.0* | 0.5 +- 0.8* |
| Hopper-H-$10^3$ | 100 | 69 | 47 | 0 | 43.1 +- 8.3 | 51.3 +- 10.2 | 76.0 +- 4.5 | **76.6 +- 1.3** | 55.0 +- 2.0 | 32.8 +- 14.5* | 11.5 +- 5.8* |
| | | | | | 43.1 +- 8.3 | 24.8 +- 21.9* | 26.1 +- 9.3* | **51.9 +- 23.2** | 13.8 +- 3.7* | 17.1 +- 0.3* | 0.0 +- 0.0* |
| | | | | | 41.3 +- 9.3 | 26.5 +- 0.6* | 24.1 +- 1.6* | **69.2 +- 4.5** | 26.4 +- 15.0* | 31.5 +- 15.5* | 0.0 +- 0.0* |
| Hopper-H-$10^4$ | 100 | 69 | 47 | 0 | 49.5 +- 14.1 | 28.1 +- 5.3* | 66.1 +- 10.0 | **81.6 +- 7.3** | 62.4 +- 5.0 | 47.3 +- 27.3* | 5.7 +- 7.8* |
| | | | | | 50.3 +- 13.5 | 13.2 +- 18.1* | 27.1 +- 6.2* | **74.3 +- 17.3** | 29.7 +- 34.1* | 15.2 +- 0.0* | -0.0 +- 0.1* |
| | | | | | 50.3 +- 13.5 | 22.8 +- 0.0* | 40.6 +- 15.5* | **87.6 +- 4.6** | 38.3 +- 28.6* | 12.9 +- 0.0* | 1.0 +- 0.0* |

Table 26: Normalized score for Walker2d tasks. For each task, three lines indicate the results of online evaluation, FQE evaluation, WIS evaluation respectively. Bold numbers indicate the best result for each task, while numbers marked by * indicate results worse than BC. The task name is composed of the specific task, the quality of dataset, and the number of trajectories. L, M, and H stands for low, medium, high respectively. Det. is abbreviation of deterministic.

| Task Name | Expert Policy | Det. Policy | Behavior Policy | Random | BC | BCQ | PLAS | CQL | CRR | BREMEN | MOPO |
|---|---|---|---|---|---|---|---|---|---|---|---|
| Walker2d-L-$10^2$ | 100 | 30 | 24 | 0 | 29.1 +- 3.5 | 22.2 +- 0.3* | 33.0 +- 5.1 | 30.3 +- 1.0 | **36.4 +- 4.8** | 21.8 +- 20.8* | 9.7 +- 9.1* |
|  |  |  |  |  | **29.1 +- 3.5** | 20.6 +- 0.5* | 10.7 +- 0.5* | 16.3 +- 12.8* | 27.1 +- 4.2* | 3.4 +- 2.9* | 4.5 +- 3.7* |
|  |  |  |  |  | 28.6 +- 0.0 | 7.4 +- 0.2* | 10.5 +- 2.5* | 8.4 +- 12.2* | 28.3 +- 7.3* | 7.2 +- 1.7* | 3.8 +- 1.5* |
| Walker2d-L-$10^3$ | 100 | 30 | 24 | 0 | 28.5 +- 1.9 | 38.0 +- 4.5 | 42.1 +- 10.3 | **44.7 +- 2.7** | 34.1 +- 1.8 | 32.4 +- 8.7 | 11.6 +- 14.1* |
|  |  |  |  |  | 27.1 +- 0.1 | 26.8 +- 4.9* | 16.6 +- 22.1* | **45.8 +- 1.6** | 6.3 +- 6.1* | 8.8 +- 5.0* | 0.7 +- 1.2* |
|  |  |  |  |  | 29.9 +- 1.9 | 29.3 +- 9.9* | 4.5 +- 1.6* | **31.6 +- 1.3** | 6.3 +- 6.1* | 12.4 +- 4.8* | 0.9 +- 0.7* |
| Walker2d-L-$10^4$ | 100 | 30 | 24 | 0 | 31.9 +- 2.4 | 39.1 +- 3.6 | 31.1 +- 6.5* | **40.2 +- 1.4** | 33.2 +- 7.3 | 29.4 +- 4.8* | 11.5 +- 13.9* |
|  |  |  |  |  | 32.7 +- 0.0 | 29.6 +- 6.3* | 0.1 +- 0.6* | **39.0 +- 0.0** | 29.7 +- 1.0* | 1.4 +- 0.4* | -0.2 +- 0.0* |
|  |  |  |  |  | 30.0 +- 1.9 | **38.6 +- 0.0** | 1.4 +- 1.6* | 33.1 +- 0.0 | 30.3 +- 11.1 | 2.4 +- 1.2* | -0.3 +- 0.1* |
| Walker2d-M-$10^2$ | 100 | 49 | 43 | 0 | 50.2 +- 4.0 | 42.0 +- 1.0* | 51.6 +- 1.7 | **53.2 +- 2.5** | 39.5 +- 4.8* | 37.6 +- 26.5* | 20.1 +- 15.5* |
|  |  |  |  |  | **50.2 +- 4.0** | 8.7 +- 0.6* | 26.2 +- 18.1* | 37.2 +- 9.3* | 27.1 +- 4.2* | 15.3 +- 11.2* | 8.9 +- 7.1* |
|  |  |  |  |  | 47.4 +- 0.1 | 39.4 +- 2.9* | **53.7 +- 0.0** | 47.9 +- 3.1 | 33.5 +- 7.6* | 14.1 +- 20.1* | 0.5 +- 1.2* |
| Walker2d-M-$10^3$ | 100 | 49 | 43 | 0 | 48.7 +- 1.9 | **61.7 +- 0.5** | 34.6 +- 13.2* | 57.3 +- 1.0 | 44.7 +- 6.9* | 37.5 +- 16.6* | 39.9 +- 2.0* |
|  |  |  |  |  | 47.6 +- 2.1 | 47.3 +- 10.4* | -0.3 +- 0.0* | 45.8 +- 0.6* | 34.2 +- 3.7* | 3.0 +- 0.4* | 12.2 +- 7.1* |
|  |  |  |  |  | 48.7 +- 1.9 | **52.7 +- 9.8** | -0.2 +- 0.1* | 48.6 +- 7.8* | 10.1 +- 11.4* | 24.6 +- 14.1* | -0.1 +- 0.0* |
| Walker2d-M-$10^4$ | 100 | 49 | 43 | 0 | 54.4 +- 3.5 | **60.2 +- 1.4** | 47.5 +- 1.5* | 58.6 +- 1.2 | 54.8 +- 2.5 | 41.5 +- 2.3* | 31.9 +- 20.3* |
|  |  |  |  |  | **56.1 +- 1.5** | 51.6 +- 11.3* | 0.2 +- 0.3* | 10.8 +- 6.4* | 38.6 +- 2.6* | 10.0 +- 1.5* | 5.4 +- 7.7* |
|  |  |  |  |  | 55.1 +- 0.0 | **58.6 +- 4.6** | 1.0 +- 1.7* | 46.9 +- 3.9* | 39.8 +- 1.1* | 19.0 +- 15.5* | 18.4 +- 23.1* |
| Walker2d-H-$10^2$ | 100 | 69 | 57 | 0 | 64.1 +- 4.9 | 47.6 +- 4.5* | 65.6 +- 0.6 | **74.3 +- 0.3** | 14.8 +- 6.1* | 24.3 +- 31.9* | 23.2 +- 3.6* |
|  |  |  |  |  | 67.0 +- 5.4 | 19.5 +- 15.8* | 4.7 +- 4.5* | **73.4 +- 1.1** | 11.6 +- 4.2* | 3.4 +- 0.0* | 14.8 +- 9.7* |
|  |  |  |  |  | 61.2 +- 1.3 | 38.3 +- 12.6* | **65.8 +- 0.5** | 59.6 +- 3.9* | 12.1 +- 3.8* | 6.5 +- 4.4* | 11.6 +- 2.8* |
| Walker2d-H-$10^3$ | 100 | 69 | 57 | 0 | 72.6 +- 4.2 | **76.6 +- 2.8** | 57.0 +- 9.4* | 75.3 +- 1.9 | 67.1 +- 9.6* | 48.0 +- 20.6* | 18.0 +- 3.0* |
|  |  |  |  |  | **74.4 +- 0.0** | 69.7 +- 7.2* | -0.3 +- 0.0* | 33.1 +- 12.4* | 57.9 +- 11.3* | 18.2 +- 9.2* | -0.2 +- 0.0* |
|  |  |  |  |  | 71.9 +- 3.5 | **72.8 +- 1.7** | 21.4 +- 30.8* | 60.9 +- 10.1* | 62.0 +- 11.0* | 32.7 +- 16.4* | -0.2 +- 0.0* |
| Walker2d-H-$10^4$ | 100 | 69 | 57 | 0 | 58.3 +- 8.4 | **77.9 +- 1.4** | 36.3 +- 4.5* | | 71.7 +- 7.0 | 48.0 +- 9.5* | 17.7 +- 0.8* |
|  |  |  |  |  | **60.1 +- 9.3** | 51.6 +- 16.3* | 1.5 +- 2.5* | 43.1 +- 20.0* | 14.9 +- 21.2* | 4.6 +- 3.8* | 1.3 +- 3.3* |
|  |  |  |  |  | 66.7 +- 0.0 | **79.3 +- 0.1** | 1.8 +- 0.0* | 74.0 +- 1.3 | 73.8 +- 8.0 | 1.9 +- 0.0* | 7.8 +- 7.5* |

Table 27: Normalized score for IB tasks. For each task, three lines indicate the results of online evaluation, FQE evaluation, WIS evaluation respectively. Bold numbers indicate the best result for each task, while numbers marked by * indicate results worse than BC. The task name is composed of the specific task, the quality of dataset, and the number of trajectories. L, M, and H stands for low, medium, high respectively. Det. is abbreviation of deterministic.

| Task Name | Expert Policy | Det. Policy | Behavior Policy | Random | BC | BCQ | PLAS | CQL | CRR | BREMEN | MOPO |
|---|---|---|---|---|---|---|---|---|---|---|---|
| IB-L-$10^2$ | 100 | -19 | -19 | 0 | -19.8 +- 1.6 | -287.5 +- 155.5* | -34.9 +- 23.6* | **2.5 +- 3.2** | -5.3 +- 14.4 | -34.5 +- 24.7* | -181.0 +- 162.7* |
|  |  |  |  |  | -19.2 +- 1.8 | -68.0 +- 0.0* | -68.0 +- 0.0* | -65.2 +- 0.0* | **-17.9 +- 3.9** | -1598.8 +- 0.0* | -1703.9 +- 0.5* |
|  |  |  |  |  | **-19.2 +- 1.8** | -411.2 +- 0.1* | -182.8 +- 162.5* | -150.2 +- 160.9* | -153.6 +- 181.3* | -97.0 +- 44.6* | -240.8 +- 140.2* |
| IB-L-$10^3$ | 100 | -19 | -19 | 0 | -16.2 +- 2.7 | -177.2 +- 155.1* | -30.5 +- 26.6* | **-0.4 +- 4.5** | -5.3 +- 17.1 | -37.3 +- 21.6* | -163.4 +- 177.7* |
|  |  |  |  |  | **-14.4 +- 0.2** | -68.0 +- 0.3* | -68.1 +- 0.0* | -53.8 +- 27.7* | -21.7 +- 10.2* | -19.3 +- 6.2* | -1704.1 +- 0.3* |
|  |  |  |  |  | -20.0 +- 0.0 | -68.2 +- 0.3* | -68.1 +- 0.0* | -179.0 +- 139.9* | **-14.6 +- 18.2** | -283.9 +- 0.0* | -1158.9 +- 771.6* |
| IB-L-$10^4$ | 100 | -19 | -19 | 0 | -18.6 +- 3.0 | -177.6 +- 155.4* | -146.5 +- 187.9* | -6.2 +- 13.9 | **-1.5 +- 11.9** | -122.6 +- 46.3* | -171.7 +- 171.1* |
|  |  |  |  |  | **-22.6 +- 0.0** | -68.2 +- 0.1* | -68.1 +- 0.2* | -135.4 +- 99.5* | -47.9 +- 21.6* | -101.3 +- 23.2* | -1158.4 +- 771.2* |
|  |  |  |  |  | -17.9 +- 3.3 | -67.9 +- 0.2* | -182.5 +- 162.4* | **3.7 +- 2.8** | -17.4 +- 0.0 | -68.6 +- 0.0* | -2.0 +- 0.0 |
| IB-M-$10^2$ | 100 | 25 | 25 | 0 | -9.2 +- 46.9 | -177.7 +- 155.2* | -291.7 +- 160.4* | 24.4 +- 4.9 | **25.6 +- 3.8** | -97.8 +- 104.1* | -59.8 +- 5.7* |
|  |  |  |  |  | 18.2 +- 0.0 | -67.9 +- 0.1* | -182.6 +- 162.6* | -27.9 +- 21.5* | **21.1 +- 0.0** | -349.4 +- 0.0* | -224.6 +- 140.9* |
|  |  |  |  |  | **25.7 +- 5.3** | -182.6 +- 162.2* | -297.8 +- 162.5* | -125.6 +- 187.0* | -3.1 +- 41.7* | -214.7 +- 314.9* | -612.6 +- 771.3* |
| IB-M-$10^3$ | 100 | 25 | 25 | 0 | 27.1 +- 0.4 | -181.6 +- 161.0* | -182.7 +- 161.9* | 25.2 +- 1.6* | **28.9 +- 2.9** | -16.0 +- 32.6* | -119.2 +- 85.7* |
|  |  |  |  |  | **26.6 +- 0.0** | -67.7 +- 0.2* | -67.6 +- 0.0* | -237.2 +- 0.0* | 8.7 +- 0.0* | -207.3 +- 117.1* | -67.3 +- 0.0* |
|  |  |  |  |  | **27.3 +- 0.2** | -297.3 +- 162.1* | -412.0 +- 0.3* | -1679.1 +- 0.0* | 27.2 +- 1.2* | 4.1 +- 1.5* | -123.1 +- 80.2* |
| IB-M-$10^4$ | 100 | 25 | 25 | 0 | 27.7 +- 2.7 | -181.5 +- 163.1* | -182.6 +- 162.3* | 26.9 +- 6.4* | **30.4 +- 0.4** | 1.6 +- 18.4* | -48.8 +- 26.2* |
|  |  |  |  |  | 29.4 +- 0.7 | -67.8 +- 0.1* | -67.7 +- 0.0* | -56.1 +- 13.0* | **31.3 +- 0.0** | 27.8 +- 0.3* | -1704.2 +- 0.0* |
|  |  |  |  |  | 27.3 +- 2.3 | -67.7 +- 0.3* | -67.7 +- 0.0* | -14.8 +- 32.2* | **28.5 +- 2.8** | -395.9 +- 0.0* | -1704.2 +- 0.0* |
| IB-H-$10^2$ | 100 | 70 | 70 | 0 | 57.8 +- 30.5 | -288.6 +- 77.0* | -178.5 +- 78.5* | 32.9 +- 27.0* | **73.2 +- 0.1** | -89.1 +- 108.5* | -77.0 +- 70.0* |
|  |  |  |  |  | **72.0 +- 0.0** | -842.3 +- 609.1* | -183.1 +- 162.3* | -140.5 +- 192.0* | -493.7 +- 801.7* | -1055.0 +- 457.8* | -269.4 +- 146.2* |
|  |  |  |  |  | 62.4 +- 33.2 | -241.3 +- 0.2* | -594.5 +- 744.6* | -25.1 +- 67.9* | **72.8 +- 0.1** | -811.5 +- 695.3* | -116.1 +- 77.7* |
| IB-H-$10^3$ | 100 | 70 | 70 | 0 | 9.4 +- 88.0 | -297.9 +- 80.7* | -171.4 +- 146.4* | 15.5 +- 48.9 | **69.7 +- 0.1** | -31.5 +- 113.4* | -97.5 +- 89.6* |
|  |  |  |  |  | -31.6 +- 77.9 | -398.4 +- 0.0* | -68.1 +- 0.2* | -165.7 +- 86.4* | **69.6 +- 0.0** | -114.8 +- 85.0* | -158.6 +- 114.9* |
|  |  |  |  |  | -94.1 +- 29.7 | -241.5 +- 0.0* | -297.7 +- 80.3* | -575.7 +- 331.5* | **72.2 +- 0.5** | -1145.9 +- 687.5* | -239.1 +- 3.2* |
| IB-H-$10^4$ | 100 | 70 | 70 | 0 | -34.2 +- 111.3 | -183.1 +- 81.4* | -184.7 +- 82.1* | 34.2 +- 34.1 | **61.7 +- 15.6** | -5.6 +- 11.5 | -127.2 +- 84.8* |
|  |  |  |  |  | -9.4 +- 124.6 | -412.0 +- 0.0* | -412.1 +- 0.1* | -130.6 +- 89.0* | **50.4 +- 15.2** | -12.1 +- 4.6* | -309.0 +- 135.5* |
|  |  |  |  |  | -185.6 +- 0.0 | -241.2 +- 0.7* | -240.2 +- 0.0* | -1220.8 +- 681.9* | **39.6 +- 0.0** | 0.4 +- 0.0 | -241.0 +- 0.0* |

Table 28: Normalized score for FinRL tasks. For each task, three lines indicate the results of online evaluation, FQE evaluation, WIS evaluation respectively. Bold numbers indicate the best result for each task, while numbers marked by * indicate results worse than BC. The task name is composed of the specific task, the quality of dataset, and the number of trajectories. L, M, and H stands for low, medium, high respectively. Det. is abbreviation of deterministic.

| Task Name | Expert Policy | Det. Policy | Behavior Policy | Random | BC | BCQ | PLAS | CQL | CRR | BREMEN | MOPO |
|---|---|---|---|---|---|---|---|---|---|---|---|
| FinRL-L-$10^2$ | 100 | -13 | -12 | 0 | 34.8 +- 60.2 | 23.2 +- 7.9* | 24.2 +- 4.2* | **48.3 +- 3.5** | 7.5 +- 1.0* | 34.8 +- 60.3 | 18.6 +- 5.7* |
| | | | | | 37.5 +- 58.2 | 32.9 +- 1.4* | 24.2 +- 4.2* | 30.4 +- 17.1* | 6.7 +- 0.0* | **119.9 +- 0.1** | 18.6 +- 5.7* |
| | | | | | 34.8 +- 60.2 | 26.8 +- 9.0* | 24.4 +- 4.9* | 49.4 +- 4.9 | -0.6 +- 10.4* | 78.8 +- 58.3 | 18.8 +- 5.5* |
| FinRL-L-$10^3$ | 100 | -13 | -12 | 0 | 18.9 +- 10.0 | 30.4 +- 6.8 | 62.6 +- 20.1 | **66.2 +- 2.3** | 24.7 +- 12.3 | 51.7 +- 49.5 | 17.6 +- 6.5* |
| | | | | | 11.6 +- 9.8 | 37.0 +- 2.0 | **73.6 +- 25.3** | 55.5 +- 15.0 | 16.1 +- 1.0 | 9.6 +- 1.4* | 16.4 +- 7.8 |
| | | | | | 26.1 +- 0.5 | 34.7 +- 4.6 | **56.7 +- 24.3** | 11.9 +- 9.5* | 26.3 +- 10.5 | 21.7 +- 3.4* | 13.9 +- 1.8* |
| FinRL-M-$10^2$ | 100 | 22 | 35 | 0 | 77.3 +- 74.5 | 21.3 +- 1.8* | 33.1 +- 19.6* | **84.2 +- 27.8** | 37.2 +- 9.5* | 77.3 +- 74.5 | 21.1 +- 6.2* |
| | | | | | 64.1 +- 82.1 | 20.1 +- 1.4* | 24.2 +- 22.4* | **71.8 +- 37.1** | 24.6 +- 10.9* | 6.0 +- 0.0* | 25.2 +- 5.9* |
| | | | | | 77.3 +- 74.5 | 20.5 +- 2.2* | 29.6 +- 20.3* | 36.2 +- 17.0* | 19.6 +- 12.6* | **102.1 +- 72.2** | 22.9 +- 4.5* |
| FinRL-M-$10^3$ | 100 | 22 | 35 | 0 | 6.4 +- 9.7 | 29.1 +- 14.5 | 50.9 +- 12.6 | 56.9 +- 25.7 | 33.0 +- 9.3 | **150.3 +- 100.0** | 20.5 +- 5.9 |
| | | | | | 1.4 +- 4.0 | 36.2 +- 10.4 | 11.8 +- 15.1 | 31.7 +- 23.3 | 19.4 +- 22.0 | **87.5 +- 68.8** | 24.8 +- 5.7 |
| | | | | | 14.2 +- 7.1 | 31.6 +- 12.0 | 42.4 +- 18.1 | 38.9 +- 29.1 | 22.2 +- 22.2 | **148.5 +- 102.4** | 18.1 +- 8.0 |
| FinRL-H-$10^2$ | 100 | 55 | 50 | 0 | 48.5 +- 26.2 | 16.6 +- 19.7* | 42.0 +- 27.6* | 57.6 +- 27.0 | 43.2 +- 20.3* | **70.6 +- 63.9** | 19.8 +- 6.0* |
| | | | | | **69.9 +- 14.6** | 0.4 +- 9.1* | 41.8 +- 15.7* | 48.8 +- 14.0* | 39.0 +- 3.3* | 5.6 +- 7.5* | 16.8 +- 1.8* |
| | | | | | 37.4 +- 30.3 | 9.7 +- 18.0* | 22.9 +- 29.1* | 29.8 +- 3.7* | 12.7 +- 1.4* | **58.9 +- 30.2** | 16.8 +- 1.8* |
| FinRL-H-$10^3$ | 100 | 55 | 50 | 0 | 14.2 +- 26.7 | 22.3 +- 19.8 | 52.9 +- 16.1 | 51.4 +- 20.4 | 35.9 +- 23.0 | **69.8 +- 65.9** | 19.2 +- 6.9 |
| | | | | | 27.6 +- 30.5 | 20.0 +- 17.8* | **46.2 +- 2.0** | 45.0 +- 1.3 | 34.4 +- 19.7 | 6.0 +- 30.6* | 15.9 +- 2.9* |
| | | | | | 0.8 +- 11.5 | 19.4 +- 18.2 | **47.5 +- 11.4** | 26.3 +- 5.1 | 27.4 +- 17.2 | 32.8 +- 33.8 | 25.1 +- 4.4 |

Table 29: Normalized score for CL tasks. For each task, three lines indicate the results of online evaluation, FQE evaluation, WIS evaluation respectively. Bold numbers indicate the best result for each task, while numbers marked by * indicate results worse than BC. The task name is composed of the specific task, the quality of dataset, and the number of trajectories. L, M, and H stands for low, medium, high respectively. Det. is abbreviation of deterministic.

| Task Name | Expert Policy | Det. Policy | Behavior Policy | Random | BC | BCQ | PLAS | CQL | CRR | BREMEN | MOPO |
|---|---|---|---|---|---|---|---|---|---|---|---|
| CL-L-$10^2$ | 100 | 35 | 38 | 0 | 30.3 +- 10.1 | 17.3 +- 3.6* | 35.1 +- 3.4 | 40.1 +- 1.4 | **44.7 +- 0.8** | 27.8 +- 9.5* | 10.8 +- 1.6* |
| | | | | | 16.9 +- 0.0 | 20.3 +- 1.0 | 12.3 +- 0.0* | 21.7 +- 0.0 | **42.3 +- 2.7** | 17.6 +- 0.4 | 9.4 +- 1.4* |
| | | | | | 25.1 +- 11.6 | 17.3 +- 3.6* | 30.5 +- 7.3 | 32.0 +- 7.8 | **39.7 +- 3.4** | 16.1 +- 1.5* | 10.2 +- 0.6* |
| CL-L-$10^3$ | 100 | 35 | 38 | 0 | 38.6 +- 1.8 | 25.0 +- 1.4* | 35.8 +- 2.5* | **46.9 +- 1.5** | 41.3 +- 2.0 | 40.1 +- 1.3 | 10.8 +- 1.7* |
| | | | | | 37.3 +- 0.1 | 22.6 +- 0.7* | 25.4 +- 0.0* | **39.3 +- 0.0** | 39.0 +- 3.3 | 36.5 +- 4.6* | 11.7 +- 0.3* |
| | | | | | 38.6 +- 1.8 | 22.9 +- 3.8* | 27.2 +- 2.6* | **38.8 +- 2.7** | 37.7 +- 0.6* | 33.9 +- 4.9* | 11.4 +- 0.5* |
| CL-L-$10^4$ | 100 | 35 | 38 | 0 | 38.3 +- 1.5 | 21.6 +- 0.8* | 36.9 +- 4.1* | **46.4 +- 1.7** | 42.8 +- 0.8 | 39.5 +- 1.0 | 10.9 +- 2.6* |
| | | | | | 38.3 +- 1.5 | 23.1 +- 0.4* | 22.1 +- 0.0* | **41.8 +- 0.5** | 37.9 +- 0.2* | 37.9 +- 0.8* | 9.7 +- 0.0* |
| | | | | | 38.3 +- 1.5 | 19.9 +- 2.6* | 28.5 +- 9.1* | **40.5 +- 0.0** | 39.9 +- 3.4 | 38.3 +- 0.2 | 7.9 +- 0.1* |
| CL-M-$10^2$ | 100 | 63 | 60 | 0 | 68.3 +- 5.6 | 29.9 +- 10.5* | 56.9 +- 3.9* | 66.8 +- 5.1* | **82.8 +- 0.9** | 63.6 +- 12.7* | 10.1 +- 2.6* |
| | | | | | 66.1 +- 3.4 | 19.1 +- 0.4* | 43.2 +- 22.1* | 45.2 +- 10.9* | **75.2 +- 6.0** | 24.8 +- 7.9* | 8.9 +- 2.3* |
| | | | | | 70.7 +- 3.1 | 24.2 +- 3.6* | 27.9 +- 17.1* | 33.1 +- 6.7* | **74.6 +- 8.0** | 46.0 +- 13.5* | 9.3 +- 0.4* |
| CL-M-$10^3$ | 100 | 63 | 60 | 0 | 63.3 +- 8.0 | 24.4 +- 3.5* | 58.5 +- 6.2* | 75.0 +- 0.6 | 74.2 +- 1.2 | **77.7 +- 12.5** | 10.8 +- 1.4* |
| | | | | | 57.4 +- 7.6 | 20.2 +- 0.7* | 35.5 +- 17.8* | 58.2 +- 0.7 | 74.9 +- 1.0 | **76.2 +- 13.6** | 9.0 +- 0.1* |
| | | | | | 68.2 +- 0.0 | 24.3 +- 0.9* | 61.4 +- 5.1* | **74.3 +- 3.3** | 70.2 +- 3.0 | 70.2 +- 2.3 | 9.0 +- 0.3* |
| CL-M-$10^4$ | 100 | 63 | 60 | 0 | 59.4 +- 15.8 | 22.1 +- 6.6* | 56.7 +- 2.9* | **77.1 +- 1.4** | 75.4 +- 0.6 | 58.7 +- 18.7* | 10.1 +- 1.4* |
| | | | | | 37.1 +- 0.0 | 13.1 +- 0.0* | 55.3 +- 7.1 | 60.9 +- 2.6 | **75.3 +- 0.7** | 15.5 +- 0.3* | 10.9 +- 0.8* |
| | | | | | 70.0 +- 0.0 | 13.0 +- 0.0* | 41.6 +- 21.7* | 67.4 +- 9.4* | **72.4 +- 1.9** | 45.7 +- 21.4* | 7.3 +- 0.7* |
| CL-H-$10^2$ | 100 | 94 | 95 | 0 | 110.5 +- 6.8 | 31.6 +- 13.8* | 88.2 +- 5.9* | 100.8 +- 2.4* | **119.3 +- 4.8** | 112.9 +- 6.4 | 11.9 +- 0.8* |
| | | | | | 105.1 +- 5.0 | 12.3 +- 0.1* | 24.5 +- 1.1* | 49.2 +- 7.6* | 109.5 +- 10.1 | **117.4 +- 0.8** | 10.2 +- 1.0* |
| | | | | | **110.5 +- 6.8** | 30.4 +- 3.7* | 66.8 +- 30.3* | 74.0 +- 27.1* | 109.7 +- 12.4* | 81.8 +- 44.1* | 9.9 +- 0.9* |
| CL-H-$10^3$ | 100 | 94 | 95 | 0 | 106.7 +- 1.6 | 37.5 +- 11.5* | 91.7 +- 9.3* | 104.7 +- 5.7* | **114.2 +- 2.2** | 110.7 +- 3.8 | 9.9 +- 0.7* |
| | | | | | **108.0 +- 0.0** | 26.0 +- 0.3* | 20.7 +- 0.1* | 84.6 +- 4.2* | 104.6 +- 3.9* | 75.6 +- 9.1* | 8.2 +- 0.3* |
| | | | | | 105.6 +- 1.7 | 27.7 +- 2.2* | 83.5 +- 23.1* | 102.6 +- 5.0* | **107.8 +- 2.7** | 106.3 +- 5.2 | 8.5 +- 0.4* |
| CL-H-$10^4$ | 100 | 94 | 95 | 0 | 98.5 +- 12.4 | 47.1 +- 18.2* | 92.5 +- 8.7* | 107.2 +- 7.3 | **113.0 +- 2.8** | 79.9 +- 44.8* | 10.8 +- 2.7* |
| | | | | | 89.9 +- 12.5 | 46.7 +- 1.2* | 18.5 +- 0.0* | 68.9 +- 0.5* | **111.5 +- 0.0** | 19.1 +- 0.7* | 9.9 +- 0.0* |
| | | | | | 107.5 +- 0.0 | 68.1 +- 1.6* | 97.4 +- 0.5* | 100.6 +- 4.7* | **113.3 +- 4.0** | 94.5 +- 25.6* | 8.8 +- 0.4* |

Table 30: Raw score for HalfCheetah tasks. For each task, three lines indicate the results of online evaluation, FQE evaluation, WIS evaluation respectively. Bold numbers indicate the best result for each task, while numbers marked by * indicate results worse than BC. The task name is composed of the specific task, the quality of dataset, and the number of trajectories. L, M, and H stands for low, medium, high respectively. Det. is abbreviation of deterministic.

| Task Name | Expert Policy | Det. Policy | Behavior Policy | Random | BC | BCQ | PLAS | CQL | CRR | BREMEN | MOPO |
|---|---|---|---|---|---|---|---|---|---|---|---|
| HalfCheetah-L-$10^2$ | 12284 | 3195 | 2871 | -298 | 3364.8 +- 43.1 | 3499.4 +- 42.6 | 3330.7 +- 51.8* | 3801.4 +- 33.8 | 3357.0 +- 20.4* | 4433.0 +- 222.9 | **4980.7 +- 231.3** |
| | | | | | 3334.4 +- 43.3 | 3427.5 +- 0.0 | 2915.9 +- 439.6* | 3667.0 +- 104.7 | 3347.2 +- 0.0 | **4298.5 +- 44.0** | 4265.4 +- 242.8 |
| | | | | | 3395.7 +- 0.0 | 3048.0 +- 198.8* | 3221.3 +- 23.0* | 3506.6 +- 201.7 | 3221.9 +- 82.1* | 4012.2 +- 424.0 | **4141.1 +- 546.5** |
| HalfCheetah-L-$10^3$ | 12284 | 3195 | 2871 | -298 | 3363.8 +- 27.0 | 3993.0 +- 49.9 | 3548.5 +- 3.8 | 4512.4 +- 65.5 | 3372.6 +- 24.4 | 4681.1 +- 230.9 | **4741.1 +- 117.2** |
| | | | | | 3350.9 +- 0.0 | 4024.3 +- 0.0 | 3541.5 +- 3.8 | **4312.7 +- 83.5** | 3302.4 +- 71.3* | 2671.5 +- 0.0* | 2833.0 +- 2432.8* |
| | | | | | 3359.9 +- 29.4 | 3696.0 +- 19.9 | 3384.5 +- 0.0 | 3139.3 +- 439.3* | 3355.9 +- 36.1* | 2492.1 +- 43.9* | **3916.6 +- 248.0** |
| HalfCheetah-L-$10^4$ | 12284 | 3195 | 2871 | -298 | 3332.9 +- 17.5 | 4320.2 +- 87.6 | 3549.7 +- 30.1 | **4715.6 +- 177.4** | 3393.8 +- 65.3 | 4621.9 +- 36.1 | 4442.4 +- 39.9 |
| | | | | | 3347.2 +- 0.0 | 4189.4 +- 144.2 | 3489.1 +- 57.6 | **4606.1 +- 145.3** | 3341.2 +- 26.3* | 4598.1 +- 0.0 | 2716.5 +- 2347.8* |
| | | | | | 3319.9 +- 16.5 | **3737.2 +- 88.1** | 3446.6 +- 88.3 | 3017.1 +- 543.9* | 3374.6 +- 36.8 | 2144.8 +- 433.4* | -599.4 +- 0.0* |
| HalfCheetah-M-$10^2$ | 12284 | 6027 | 5568 | -298 | 5857.0 +- 99.2 | 5134.3 +- 183.9* | 5573.8 +- 127.2* | 6190.7 +- 50.5 | 3126.8 +- 73.8* | 6285.0 +- 626.1 | **7635.4 +- 65.6** |
| | | | | | 5776.8 +- 0.0 | 1209.8 +- 990.1* | 4005.1 +- 0.0* | 2820.9 +- 455.6* | 1864.1 +- 160.4* | 5627.0 +- 0.0* | **6259.0 +- 51.1** |
| | | | | | 5930.3 +- 94.0 | 5103.6 +- 154.7* | 5468.6 +- 275.8* | 4759.8 +- 1433.4* | 2811.5 +- 589.7* | 5208.7 +- 1408.9* | **6227.9 +- 166.5** |
| HalfCheetah-M-$10^3$ | 12284 | 6027 | 5568 | -298 | 5866.4 +- 73.6 | 6062.8 +- 12.1 | 6092.3 +- 47.9 | 6576.2 +- 39.1 | 5137.8 +- 328.7* | 6666.9 +- 382.6 | **7534.7 +- 134.7** |
| | | | | | 5929.3 +- 0.0 | 5032.2 +- 662.2* | 3273.8 +- 0.0* | 2139.0 +- 0.0* | 3123.3 +- 847.9* | **6867.6 +- 0.0** | 4346.6 +- 3498.3* |
| | | | | | 5859.5 +- 67.9 | 5373.6 +- 905.3* | 6112.1 +- 55.3 | 5829.2 +- 447.8* | 2372.4 +- 29.5* | 4075.6 +- 1906.1* | **6667.8 +- 214.9** |
| HalfCheetah-M-$10^4$ | 12284 | 6027 | 5568 | -298 | 5995.9 +- 51.3 | 5944.6 +- 111.2* | 6091.5 +- 20.3 | 6723.2 +- 115.0 | 5243.8 +- 220.1* | **6724.1 +- 402.2** | 5195.4 +- 108.5* |
| | | | | | 5975.5 +- 0.0 | 3603.8 +- 133.1* | 3938.6 +- 778.1* | **6647.5 +- 196.8** | 2906.2 +- 56.6* | 5489.9 +- 1224.0* | 5394.7 +- 106.4* |
| | | | | | 5945.8 +- 0.0 | 4914.9 +- 1077.1* | 6104.5 +- 3.5 | 5251.7 +- 402.2* | 4984.2 +- 0.0* | **6715.5 +- 0.0** | -592.1 +- 0.0* |
| HalfCheetah-H-$10^2$ | 12284 | 9020 | 7836 | -298 | 5645.0 +- 4006.5 | 6945.7 +- 386.8 | 7781.0 +- 87.9 | **9011.7 +- 194.0** | 2715.7 +- 201.8* | 3352.0 +- 2859.2* | 5715.2 +- 1026.3 |
| | | | | | **8459.3 +- 0.0** | 5727.7 +- 0.0* | 1760.3 +- 416.0* | -92.0 +- 54.9* | 824.7 +- 983.2* | **5640.9 +- 0.0*** | 229.5 +- 104.6* |
| | | | | | **8471.8 +- 17.6** | 3305.6 +- 2519.3* | 5220.5 +- 2700.7* | 2452.9 +- 3507.6* | 1763.7 +- 702.9* | 3034.8 +- 2353.1* | 2707.6 +- 2313.5* |
| HalfCheetah-H-$10^3$ | 12284 | 9020 | 7836 | -298 | 8676.5 +- 59.5 | 8811.9 +- 44.0 | 9019.3 +- 105.3 | **9440.0 +- 166.5** | 7569.2 +- 236.6* | 6591.6 +- 2151.4* | 7994.1 +- 1300.5* |
| | | | | | **8721.3 +- 8.4** | 1929.2 +- 0.0* | 3597.3 +- 241.8* | -270.8 +- 15.8* | 869.8 +- 0.0* | **7131.3 +- 1045.9*** | 137.5 +- 533.6* |
| | | | | | 8727.2 +- 0.0 | 8161.5 +- 430.1* | **9092.7 +- 106.8** | -5.6 +- 129.0* | 3223.0 +- 736.0* | 3402.1 +- 318.6* | 1086.3 +- 302.3* |
| HalfCheetah-H-$10^4$ | 12284 | 9020 | 7836 | -298 | 8099.6 +- 344.7 | 8918.9 +- 181.1 | 9192.9 +- 74.3 | **9413.0 +- 109.3** | 8459.1 +- 55.0 | 1671.5 +- 357.7* | 657.0 +- 797.0* |
| | | | | | **8378.9 +- 0.0** | 2780.4 +- 0.0* | 2061.8 +- 590.8* | -129.8 +- 0.0* | 2891.7 +- 68.6* | 3007.2 +- 0.0* | -148.9 +- 0.0* |
| | | | | | 8306.1 +- 0.0 | 6264.1 +- 2707.3* | **9101.3 +- 0.0** | 8526.8 +- 435.2 | 8448.9 +- 46.3 | 1178.6 +- 419.9* | -594.1 +- 0.0* |

Table 31: Raw score for Hopper tasks. For each task, three lines indicate the results of online evaluation, FQE evaluation, WIS evaluation respectively. Bold numbers indicate the best result for each task, while numbers marked by * indicate results worse than BC. The task name is composed of the specific task, the quality of dataset, and the number of trajectories. L, M, and H stands for low, medium, high respectively. Det. is abbreviation of deterministic.

| Task Name | Expert Policy | Det. Policy | Behavior Policy | Random | BC | BCQ | PLAS | CQL | CRR | BREMEN | MOPO |
|---|---|---|---|---|---|---|---|---|---|---|---|
| Hopper-L-$10^2$ | 3294 | 508 | 498 | 5 | 533.4 +- 21.1 / **533.4 +- 21.1** / **533.4 +- 21.1** | 509.6 +- 11.3* / 370.6 +- 73.6* / 403.6 +- 0.0* | 518.6 +- 8.4* / 234.0 +- 190.2* / 328.0 +- 94.6* | **548.9 +- 15.4** / 499.5 +- 9.0* / 528.4 +- 25.1* | 543.8 +- 42.9 / 520.9 +- 0.5* / 529.6 +- 7.4* | 511.5 +- 31.1* / 356.4 +- 206.4* / 40.8 +- 1.3* | 169.1 +- 200.8* / 50.3 +- 4.2* / 38.0 +- 22.9* |
| Hopper-L-$10^3$ | 3294 | 508 | 498 | 5 | 502.6 +- 23.1 / 502.6 +- 23.1 / 484.8 +- 20.1 | 601.1 +- 6.3 / 494.0 +- 124.6* / 578.9 +- 31.9 | 638.9 +- 52.3 / **602.0 +- 52.1** / 533.8 +- 85.0 | 530.7 +- 4.0 / 511.3 +- 6.5 / 518.5 +- 12.8 | 557.1 +- 20.6 / 566.3 +- 72.5 / **580.0 +- 63.0** | **708.8 +- 250.7** / 74.5 +- 14.6* / 47.4 +- 5.3* | 209.9 +- 101.1* / 22.1 +- 30.8* / 125.3 +- 0.0* |
| Hopper-L-$10^4$ | 3294 | 508 | 498 | 5 | 513.2 +- 11.3 / 519.7 +- 12.6 / 522.1 +- 9.2 | 618.8 +- 47.4 / 477.1 +- 94.6* / **575.4 +- 63.0** | 578.8 +- 48.5 / 498.4 +- 37.0* / 546.7 +- 0.0 | 521.3 +- 1.1 / 501.1 +- 21.7* / 472.9 +- 2.4* | **693.6 +- 142.3** / **580.9 +- 20.4** / 538.7 +- 50.2 | 507.5 +- 50.3* / 345.9 +- 275.6* / 276.0 +- 0.0* | 248.2 +- 75.5* / 15.6 +- 16.9* / 25.8 +- 19.2* |
| Hopper-M-$10^2$ | 3294 | 1530 | 1410 | 5 | 926.1 +- 375.9 / 949.9 +- 358.0 / 1203.0 +- 358.0 | 1350.2 +- 50.5 / 695.1 +- 513.2* / 963.2 +- 466.2* | 1648.8 +- 112.6 / 1010.4 +- 231.1 / 1010.4 +- 231.1* | **2084.6 +- 308.5** / **1420.4 +- 276.1** / **2302.2 +- 270.5** | 1370.3 +- 323.5 / 984.0 +- 36.6 / 1200.6 +- 64.5* | 942.3 +- 207.7 / 213.0 +- 136.5* / 199.2 +- 78.0* | 64.2 +- 86.1* / 38.9 +- 23.8* / 80.9 +- 77.1* |
| Hopper-M-$10^3$ | 3294 | 1530 | 1410 | 5 | 1692.0 +- 894.1 / **2344.1 +- 863.0** / 998.1 +- 0.0 | 1573.6 +- 364.4* / 1091.4 +- 454.6* / 1101.8 +- 252.9 | 2018.2 +- 848.8 / 1068.7 +- 222.5* / 926.4 +- 953.9* | **2124.8 +- 231.8** / 1889.7 +- 46.9* / **1766.0 +- 5.0** | 1576.2 +- 346.2* / 1282.6 +- 496.0* / 1391.0 +- 392.5 | 816.5 +- 181.7* / 704.1 +- 0.0* / 704.1 +- 0.0* | 39.0 +- 48.8* / 2.8 +- 1.8* / 4.0 +- 0.3* |
| Hopper-M-$10^4$ | 3294 | 1530 | 1410 | 5 | 1794.1 +- 486.3 / **1873.7 +- 0.0** / **2031.1 +- 222.6** | 1865.5 +- 257.5 / 985.6 +- 104.0* / 1021.4 +- 30.4* | 2075.1 +- 560.0 / 474.7 +- 0.0* / 246.7 +- 161.3* | **2690.3 +- 429.8** / 1443.7 +- 212.8* / 1348.1 +- 81.9* | 1620.2 +- 73.0* / 1159.9 +- 640.3* / 156.5 +- 0.0* | 1521.7 +- 463.0* / 499.5 +- 92.7* / 551.2 +- 128.3* | 40.6 +- 28.2* / 3.2 +- 0.0* / 0.1 +- 2.2* |
| Hopper-H-$10^2$ | 3294 | 2294 | 1551 | 5 | 1465.0 +- 406.9 / 1289.2 +- 469.6 / 957.2 +- 0.0 | 1179.0 +- 213.6* / 443.4 +- 476.6* / 289.2 +- 0.0* | 1892.9 +- 228.4 / 360.2 +- 90.8* / 476.8 +- 224.6* | **2298.4 +- 282.0** / **1456.5 +- 495.1** / **1518.6 +- 381.1** | 2162.6 +- 420.6 / 1399.3 +- 18.3 / 1266.6 +- 537.3 | 941.9 +- 381.5* / 5.1 +- 0.3* / 6.1 +- 0.0* | 253.7 +- 277.2* / 4.9 +- 2.1* / 21.0 +- 25.2* |
| Hopper-H-$10^3$ | 3294 | 2294 | 1551 | 5 | 1424.1 +- 271.8 / 1424.1 +- 271.8 / 1363.5 +- 304.3 | 1691.0 +- 336.1 / 821.9 +- 721.0* / 876.2 +- 19.2* | 2504.3 +- 149.0 / 862.0 +- 304.8* / 796.3 +- 53.6* | **2525.1 +- 44.0** / **1713.0 +- 762.2** / **2280.6 +- 147.0** | 1813.9 +- 65.8 / 458.4 +- 122.9* / 873.6 +- 493.5* | 1082.2 +- 475.8* / 565.8 +- 9.7* / 1040.3 +- 511.2* | 382.1 +- 191.3* / 5.1 +- 1.4* / 6.2 +- 1.1* |
| Hopper-H-$10^4$ | 3294 | 2294 | 1551 | 5 | 1633.0 +- 464.6 / 1660.2 +- 444.2 / 1660.2 +- 444.2 | 928.6 +- 172.7* / 437.7 +- 595.8* / 753.3 +- 0.0* | 2178.1 +- 330.4 / 894.7 +- 205.5* / 1339.5 +- 509.9* | **2690.3 +- 239.6** / **2449.9 +- 567.9** / **2886.6 +- 152.1** | 2058.1 +- 163.5 / 982.0 +- 1123.1* / 1263.4 +- 941.0* | 1560.6 +- 898.7* / 504.3 +- 0.0* / 430.5 +- 0.0* | 191.6 +- 258.1* / 3.5 +- 3.1* / 38.8 +- 0.0* |

Table 32: Raw score for Walker2d tasks. For each task, three lines indicate the results of online evaluation, FQE evaluation, WIS evaluation respectively. Bold numbers indicate the best result for each task, while numbers marked by * indicate results worse than BC. The task name is composed of the specific task, the quality of dataset, and the number of trajectories. L, M, and H stands for low, medium, high respectively. Det. is abbreviation of deterministic.

| Task Name | Expert Policy | Det. Policy | Behavior Policy | Random | BC | BCQ | PLAS | CQL | CRR | BREMEN | MOPO |
|---|---|---|---|---|---|---|---|---|---|---|---|
| Walker2d-L-10$^2$ | 5143 | 1572 | 1278 | 1 | 1495.4 +- 179.0 | 1144.9 +- 17.9* | 1696.7 +- 263.9 | 1558.6 +- 52.2 | **1870.9 +- 248.4** | 1121.7 +- 1069.9* | 502.2 +- 468.2* |
| | | | | | **1495.4 +- 179.0** | 1058.0 +- 27.5* | 549.8 +- 24.4* | 839.0 +- 656.1* | 1393.3 +- 214.7* | 176.5 +- 149.7* | 232.1 +- 189.9* |
| | | | | | **1470.1 +- 0.0** | 381.9 +- 8.9* | 539.6 +- 126.2* | 431.8 +- 629.3* | 1457.5 +- 372.9* | 373.3 +- 85.7* | 194.1 +- 75.8* |
| Walker2d-L-10$^3$ | 5143 | 1572 | 1278 | 1 | 1466.5 +- 99.8 | 1953.6 +- 231.6 | 2166.8 +- 531.1 | **2298.8 +- 139.1** | 1753.1 +- 90.4 | 1667.9 +- 449.1 | 599.4 +- 725.3* |
| | | | | | 1394.7 +- 3.5 | 1378.2 +- 251.3* | 855.5 +- 1137.9* | **2353.5 +- 83.3** | 999.1 +- 546.5* | 453.8 +- 255.4* | 35.9 +- 63.2* |
| | | | | | 1538.2 +- 98.0 | 1508.7 +- 511.2* | 230.4 +- 80.0* | **1624.2 +- 66.7** | 325.5 +- 313.7* | 639.6 +- 247.3* | 47.2 +- 37.2* |
| Walker2d-L-10$^4$ | 5143 | 1572 | 1278 | 1 | 1642.7 +- 125.8 | 2010.1 +- 186.7 | 1601.7 +- 333.4* | **2070.4 +- 70.2** | 1710.6 +- 373.4 | 1511.7 +- 246.3* | 594.8 +- 714.2* |
| | | | | | 1681.9 +- 0.0 | 1522.6 +- 326.1* | 4.9 +- 31.2* | **2008.5 +- 0.0** | 1526.6 +- 52.8* | 71.1 +- 22.6* | -7.5 +- 1.5* |
| | | | | | 1542.5 +- 98.6 | **1983.8 +- 0.0** | 71.6 +- 80.1* | 1703.4 +- 0.0 | 1561.0 +- 568.2 | 126.2 +- 61.3* | -14.6 +- 5.5* |
| Walker2d-M-10$^2$ | 5143 | 2547 | 2221 | 1 | 2582.2 +- 205.5 | 2159.3 +- 53.8* | 2652.5 +- 89.8 | **2734.1 +- 129.0** | 2033.3 +- 246.3* | 1936.4 +- 1365.2* | 1036.5 +- 796.8* |
| | | | | | **2582.2 +- 205.5** | 449.7 +- 30.4* | 1346.3 +- 928.2* | 1912.6 +- 477.4* | 1857.0 +- 359.9* | 789.8 +- 575.4* | 456.7 +- 363.8* |
| | | | | | 2438.2 +- 3.6 | 2028.1 +- 147.7* | **2761.9 +- 0.0** | 2461.5 +- 159.9 | 1724.7 +- 389.2* | 727.8 +- 1032.0* | 24.2 +- 62.0* |
| Walker2d-M-10$^3$ | 5143 | 2547 | 2221 | 1 | 2503.3 +- 97.9 | **3173.7 +- 27.8** | 1778.7 +- 679.5* | 2947.7 +- 49.7 | 2300.4 +- 354.3* | 1927.8 +- 852.2* | 2051.9 +- 104.6* |
| | | | | | **2448.2 +- 109.9** | 2435.4 +- 534.0* | -16.0 +- 1.3* | 2356.2 +- 30.7* | 1761.4 +- 192.3* | 157.3 +- 22.8* | 629.8 +- 364.7 |
| | | | | | 2503.3 +- 97.9 | **2708.8 +- 503.2** | -9.4 +- 6.8* | 2501.3 +- 402.4* | 520.6 +- 587.4* | 1264.4 +- 725.9* | -5.5 +- 0.0* |
| Walker2d-M-10$^4$ | 5143 | 2547 | 2221 | 1 | 2795.7 +- 178.4 | **3095.9 +- 73.7** | 2444.1 +- 77.7* | 3016.4 +- 62.3 | 2816.9 +- 126.5 | 2132.7 +- 116.2* | 1642.6 +- 1045.4* |
| | | | | | **2886.3 +- 75.7** | 2652.3 +- 583.5* | 9.7 +- 17.5* | 556.2 +- 327.1* | 1983.3 +- 131.4* | 516.8 +- 74.7* | 276.4 +- 396.9* |
| | | | | | 2832.7 +- 0.0 | **3011.8 +- 236.9** | 53.8 +- 87.0* | 2413.8 +- 200.8* | 2049.4 +- 56.6* | 977.0 +- 797.7* | 947.1 +- 1190.3* |
| Walker2d-H-10$^2$ | 5143 | 3550 | 2936 | 1 | 3299.0 +- 252.8 | 2448.0 +- 229.6* | 3376.7 +- 30.5 | **3822.1 +- 14.2** | 761.1 +- 311.6* | 1250.8 +- 1642.4* | 1195.7 +- 185.7* |
| | | | | | 3448.4 +- 280.0 | 1001.8 +- 811.0* | 245.0 +- 233.5* | **3776.1 +- 58.6** | 595.9 +- 213.9* | 173.9 +- 0.0* | 763.5 +- 496.5* |
| | | | | | 3149.5 +- 68.7 | 1970.6 +- 647.6* | **3382.6 +- 25.5** | 3065.8 +- 201.9* | 622.5 +- 195.9* | 337.0 +- 226.0* | 595.8 +- 146.1* |
| Walker2d-H-10$^3$ | 5143 | 3550 | 2936 | 1 | 3736.5 +- 213.8 | **3939.4 +- 146.1** | 2930.5 +- 480.9* | 3870.9 +- 95.7 | 3453.2 +- 495.2* | 2470.9 +- 1057.8* | 924.0 +- 153.6* |
| | | | | | **3826.0 +- 0.0** | 3583.9 +- 369.1* | -15.7 +- 0.5* | 1703.1 +- 637.6* | 2975.7 +- 581.1* | 935.4 +- 471.2* | -8.5 +- 1.9* |
| | | | | | 3697.9 +- 181.2 | **3743.3 +- 89.3** | 1101.2 +- 1582.8* | 3132.0 +- 520.0* | 3187.1 +- 566.9* | 1684.3 +- 842.9* | -10.6 +- 0.0* |
| Walker2d-H-10$^4$ | 5143 | 3550 | 2936 | 1 | 3000.9 +- 429.7 | **4007.1 +- 71.1** | 1866.3 +- 230.8* | 3850.6 +- 42.5 | 3685.9 +- 361.3 | 2470.3 +- 487.2* | 910.4 +- 43.5* |
| | | | | | **3093.5 +- 479.9** | 2655.0 +- 838.8* | 76.8 +- 130.9* | 2217.5 +- 1026.9* | 767.0 +- 1088.1* | 236.5 +- 195.2* | 65.6 +- 168.8* |
| | | | | | 3432.9 +- 0.0 | **4078.7 +- 3.0** | 93.3 +- 0.0* | 3806.9 +- 66.0 | 3796.7 +- 413.2 | 98.4 +- 0.0* | 401.2 +- 386.0* |

Table 33: Raw score for IB tasks. For each task, three lines indicate the results of online evaluation, FQE evaluation, WIS evaluation respectively. Bold numbers indicate the best result for each task, while numbers marked by * indicate results worse than BC. The task name is composed of the specific task, the quality of dataset, and the number of trajectories. L, M, and H stands for low, medium, high respectively. Det. is abbreviation of deterministic.

| Task Name | Expert Policy | Det. Policy | Behavior Policy | Random | BC | BCQ | PLAS | CQL | CRR | BREMEN | MOPO |
|---|---|---|---|---|---|---|---|---|---|---|---|
| IB-L-$10^2$ | -180240 | -344311 | -344311 | -317624 | -344761.7 + 2189.9
-344010.5 + 2518.4
**-344010.5 + 2518.4** | -712660.8 + 213594.5*
-410980.4 + 0.0*
-882595.2 + 159.0* | -365620.4 + 32376.3*
-411022.0 + 54.0*
-568695.2 + 223249.5* | **-314124.0 + 4338.2**
-407261.1 + 0.0*
-523939.1 + 221024.0* | -324878.5 + 19785.4
**-342228.8 + 5426.2**
-528695.9 + 249141.0* | -365081.2 + 33973.5*
-2514183.4 + 0.0*
-450899.9 + 61330.4* | -566237.9 + 223483.5*
-2658538.0 + 628.8*
-648459.8 + 192663.8* |
| IB-L-$10^3$ | -180240 | -344311 | -344311 | -317624 | -339937.4 + 3645.8
**-337435.8 +- 207.1**
-345087.1 + 0.0 | -561099.6 + 213072.1*
-411067.4 + 380.0*
-411336.1 + 380.0* | -359489.8 + 36580.4*
-411180.5 + 30.6*
-411180.5 + 30.6* | **-318117.5 + 6148.1**
-391534.8 +- 38114.8*
-563498.4 +- 192211.8* | -324936.1 + 23527.7
-347386.4 +- 13984.6*
**-337748.6 + 25050.1** | -368816.0 + 29642.4*
-344129.4 + 8507.8*
-707684.5 + 0.0* | -542129.0 + 244107.3*
-2658827.7 + 367.7*
-1909782.3 +- 1060017.9* |
| IB-L-$10^4$ | -180240 | -344311 | -344311 | -317624 | -343210.4 + 4055.4
**-348684.4 + 0.0**
-342222.6 + 4569.2 | -561638.9 + 213473.5*
-411269.4 + 108.5*
-410884.1 + 266.8* | -518946.1 + 258148.7*
-411215.8 + 331.2*
-568326.4 + 223146.3* | -326129.3 + 19145.8
-503697.6 + 136754.9*
**-312506.4 + 3855.2** | **-319686.8 + 16332.2**
-383450.3 + 29621.1*
-341559.8 + 0.0 | -486088.4 + 63556.2*
-456833.6 + 31807.7*
-411850.7 + 0.0* | -553469.5 + 235120.3*
-1909064.2 +- 1059572.7*
-320385.5 + 0.0 |
| IB-M-$10^2$ | -180240 | -283121 | -283121 | -317624 | -330246.8 + 64458.5
-292685.3 + 0.0
**-282292.2 + 7349.0** | -561785.5 + 213250.4*
-410883.0 + 134.2*
-568506.2 + 222892.9* | -718368.5 + 220349.4*
-568521.1 + 222562.0*
-726813.5 + 223210.8* | -284143.4 + 6669.0
-355898.0 + 29505.0*
-490209.9 +- 256942.6* | **-282481.3 + 5208.6**
-288604.9 + 0.0
-321847.6 + 57271.1* | -451975.5 + 143006.8*
-797642.7 + 0.0*
-612565.6 + 432657.5* | -399812.5 + 7797.9*
-626245.1 +- 193554.9*
-1159253.2 +- 1059689.4* |
| IB-M-$10^3$ | -180240 | -283121 | -283121 | -317624 | -280375.7 + 558.5
**-281093.4 + 0.0**
-280112.1 +- 269.3 | -567089.1 + 221151.5*
-410678.6 + 299.6*
-726046.4 + 227700.2* | -568661.4 + 222473.2*
-410490.8 +- 0.0*
-883615.3 + 466.0* | -283066.3 + 2155.5*
-643440.7 + 0.0*
-2624408.2 + 0.0* | **-277867.7 + 4022.3**
-305740.2 +- 0.0*
-280204.8 +- 1612.8* | -339595.9 + 44727.3*
-602386.8 +- 160945.3*
-312046.3 + 2035.2* | -481452.0 +- 117723.5*
-410133.7 +- 41.9*
-486741.5 +- 110201.3* |
| IB-M-$10^4$ | -180240 | -283121 | -283121 | -317624 | -279500.8 + 3683.9
-277271.0 + 895.8
-280134.2 + 3153.4 | -566925.8 + 224088.0*
-410798.5 + 151.7*
-410605.6 + 412.9* | -566501.1 + 222914.8*
-410619.4 + 0.0*
-410619.4 + 0.0* | -280729.5 + 8810.7*
-394629.5 + 17923.3*
-337979.7 + 44268.3* | **-275867.2 + 599.8**
**-274633.4 + 0.0**
**-278415.0 + 37997.9** | -315430.2 + 25323.7*
-279422.7 + 454.7*
-861522.7 + 0.0* | -384687.7 + 35961.9*
-2658889.2 + 0.0*
-2658889.2 + 0.0* |
| IB-H-$10^2$ | -180240 | -220156 | -220156 | -317624 | -238255.2 + 41897.6
**-218726.7 + 0.0**
-231870.5 + 45676.1 | -714139.4 + 105782.0*
-1474776.7 + 836850.3*
-649128.2+ 328.8* | -562798.2 + 107829.8*
-569125.3 + 222954.3*
-1134395.0+ 1023028.0* | -272439.9 + 37056.6*
-510704.8 + 263728.2*
-352046.2 + 93275.8* | **-217115.0 + 173.9**
-995874.5 + 1101348.1*
-217558.2 + 172.2 | -440008.5 + 149038.2*
-1766982.1 + 628919.3*
-1432446.8 + 955206.6* | -423373.3 + 96169.4*
-687799.3 + 200914.1*
-477191.2 +- 106725.4* |
| IB-H-$10^3$ | -180240 | -220156 | -220156 | -317624 | -304761.7 +- 120946.8
-361090.7 +- 107008.9
-446940.8 + 40816.6 | -726915.6 + 110848.7*
-864952.4 + 0.0*
-649403.4 + 0.0* | -553143.3 + 201107.2*
-411240.4 + 282.5*
-726668.3 + 110379.9* | -296295.2 + 67178.7
-545224.5 + 118751.8*
-1108589.6 + 455436.8* | **-221840.5 + 152.2**
-221950.0 + 116752.7*
-218444.2 + 652.3 | -360952.6 + 155810.4*
-475409.2 + 116752.7*
-1891941.4 + 944473.7* | -451568.8 + 123100.2*
-535490.3 + 157894.3*
-646042.5 + 4369.4* |
| IB-H-$10^4$ | -180240 | -220156 | -220156 | -317624 | -364645.1 +- 152919.1
-330487.1 +- 171238.7
-572655.2 + 0.0 | -569215.6 + 111876.4*
-883653.4 + 0.0*
-648956.2 + 912.7* | -571397.9 + 112758.7*
-883833.5 + 173.4*
-647685.7 + 0.0* | -270589.9 + 46831.4
-497089.6 + 122231.0*
-1994869.7 + 936859.5* | **-232892.3 + 21416.0**
**-248358.5 + 20928.1**
**-263156.9 + 0.0** | -325249.6 + 15788.0
-334292.3 + 6358.7*
-317105.0 + 0.0 | -492383.6 + 116498.6*
-742097.7 +- 186183.2*
-648666.8 + 0.0* |

Table 34: Raw score for FinRL tasks. For each task, three lines indicate the results of online evaluation, FQE evaluation, WIS evaluation respectively. Bold numbers indicate the best result for each task, while numbers marked by $*$ indicate results worse than BC. The task name is composed of the specific task, the quality of dataset, and the number of trajectories. L, M, and H stands for low, medium, high respectively. Det. is abbreviation of deterministic.

| Task Name | Expert Policy | Det. Policy | Behavior Policy | Random | BC | BCQ | PLAS | CQL | CRR | BREMEN | MOPO |
|---|---|---|---|---|---|---|---|---|---|---|---|
| FinRL-L-$10^2$ | 631 | 150 | 152 | 206 | 353.8 +- 256.0 | 304.8 +- 33.5* | 308.9 +- 17.7* | **411.3 +- 15.0** | 237.8 +- 4.4* | 354.0 +- 256.3 | 285.2 +- 24.2* |
| | | | | | 365.5 +- 247.4 | 346.0 +- 5.9* | 308.9 +- 17.7* | 335.1 +- 72.5* | 234.6 +- 0.1* | **715.6 +- 0.3** | 285.0 +- 24.3* |
| | | | | | 353.8 +- 256.0 | 320.0 +- 38.1* | 309.5 +- 20.8* | 415.8 +- 20.7 | 203.5 +- 44.2* | **540.8 +- 247.7** | 286.0 +- 23.2* |
| FinRL-L-$10^3$ | 631 | 150 | 152 | 206 | 286.2 +- 42.6 | 335.2 +- 28.7 | 471.9 +- 85.5 | **487.5 +- 9.8** | 310.8 +- 52.1 | 425.6 +- 210.5 | 280.8 +- 27.7* |
| | | | | | 255.4 +- 41.6 | 363.3 +- 8.7 | **518.8 +- 107.4** | 441.7 +- 63.7 | 274.3 +- 4.4 | 247.0 +- 6.0* | 275.7 +- 33.1 |
| | | | | | 317.1 +- 2.0 | 353.5 +- 19.6 | **446.8 +- 103.3** | 256.5 +- 40.4* | 317.6 +- 44.6 | 298.1 +- 14.4* | 264.9 +- 7.8* |
| FinRL-M-$10^2$ | 631 | 300 | 357 | 206 | 534.7 +- 316.5 | 296.7 +- 7.7* | 346.8 +- 83.2* | **563.8 +- 118.0** | 364.2 +- 40.3* | 534.7 +- 316.5 | 295.6 +- 26.2* |
| | | | | | 478.3 +- 348.8 | 291.5 +- 5.8* | 308.9 +- 95.0* | **511.2 +- 157.8** | 310.5 +- 46.4* | 231.6 +- 0.0* | 312.9 +- 25.2* |
| | | | | | 534.7 +- 316.5 | 293.1 +- 9.3* | 331.6 +- 86.3* | 359.7 +- 72.1* | 289.5 +- 53.5* | **639.8 +- 306.9** | 303.5 +- 19.3* |
| FinRL-M-$10^3$ | 631 | 300 | 357 | 206 | 233.2 +- 41.3 | 329.8 +- 61.7 | 422.2 +- 53.7 | 448.0 +- 109.3 | 346.4 +- 39.4 | **844.8 +- 425.0** | 293.0 +- 25.2 |
| | | | | | 211.8 +- 16.9 | 359.7 +- 44.3 | 256.2 +- 64.3 | 340.9 +- 99.1 | 288.4 +- 93.4 | **577.7 +- 292.2** | 311.5 +- 24.2 |
| | | | | | 266.5 +- 30.2 | 340.2 +- 51.1 | 386.2 +- 77.0 | 371.5 +- 123.5 | 300.5 +- 94.2 | **837.2 +- 435.3** | 283.1 +- 33.9 |
| FinRL-H-$10^2$ | 631 | 441 | 419 | 206 | 412.1 +- 111.4 | 276.7 +- 83.6* | 384.5 +- 117.3* | 450.8 +- 114.8 | 389.7 +- 86.2* | **506.1 +- 271.7** | 290.3 +- 25.5* |
| | | | | | **503.1 +- 62.2** | 207.7 +- 38.6* | 383.7 +- 66.9* | 413.6 +- 59.5* | 271.3 +- 59.7* | 229.9 +- 31.9* | 277.2 +- 7.7* |
| | | | | | 365.1 +- 128.6 | 247.2 +- 76.4* | 303.5 +- 123.5* | 332.7 +- 15.7* | 259.8 +- 6.1* | **456.2 +- 128.5** | 277.2 +- 7.7* |
| FinRL-H-$10^3$ | 631 | 441 | 419 | 206 | 266.3 +- 113.3 | 300.7 +- 84.1 | 430.7 +- 68.4 | 424.3 +- 86.5 | 358.5 +- 97.9 | **502.8 +- 280.1** | 287.7 +- 29.2 |
| | | | | | 323.2 +- 129.5 | 290.9 +- 75.7* | **402.2 +- 8.3** | 397.3 +- 5.5 | 352.2 +- 83.8 | 231.6 +- 130.2* | 273.6 +- 12.3* |
| | | | | | 209.3 +- 48.9 | 288.3 +- 77.5 | **407.9 +- 48.5** | 317.7 +- 21.6 | 322.6 +- 73.3 | 345.5 +- 143.7 | 312.8 +- 18.5 |

Table 35: Raw score for CL tasks. For each task, three lines indicate the results of online evaluation, FQE evaluation, WIS evaluation respectively. Bold numbers indicate the best result for each task, while numbers marked by * indicate results worse than BC. The task name is composed of the specific task, the quality of dataset, and the number of trajectories. L, M, and H stands for low, medium, high respectively. Det. is abbreviation of deterministic.

| Task Name | Expert Policy | Det. Policy | Behavior Policy | Random | BC | BCQ | PLAS | CQL | CRR | BREMEN | MOPO |
|---|---|---|---|---|---|---|---|---|---|---|---|
| CL-L-$10^2$ | 50350 | 28500 | 29514 | 16280 | 26603.8 +- 3453.2 | 22179.4 +- 1237.0* | 28246.0 +- 1170.6 | 29928.2 +- 478.7 | **31525.3 +- 279.7** | 25749.4 +- 3234.2* | 19961.2 +- 550.7* |
| | | | | | 22052.9 +- 0.0 | 23207.1 +- 348.2 | 20456.1 +- 0.0* | 23673.5 +- 0.0 | **30684.6 +- 909.3** | 22287.6 +- 152.9 | 19488.7 +- 489.8* |
| | | | | | 24839.8 +- 3941.2 | 22179.4 +- 1237.0* | 26659.8 +- 2475.5 | 27187.2 +- 2653.9 | **29799.5 +- 1142.3** | 21778.2 +- 512.5* | 19765.2 +- 193.8* |
| CL-L-$10^3$ | 50350 | 28500 | 29514 | 16280 | 29439.4 +- 614.6 | 24786.0 +- 484.9* | 28482.5 +- 863.4* | **32265.3 +- 500.3** | 30334.9 +- 670.7 | 29935.4 +- 449.6 | 19957.8 +- 562.9* |
| | | | | | 28997.7 +- 20.9 | 23983.1 +- 254.7* | 24920.0 +- 0.0* | **29672.8 +- 16.9** | 29557.3 +- 1126.9 | 28706.8 +- 1566.6* | 20261.9 +- 106.4* |
| | | | | | 29439.4 +- 614.6 | 24098.2 +- 1295.0* | 25537.9 +- 873.9* | **29488.9 +- 921.6** | 29112.3 +- 196.2* | 27842.5 +- 1684.8* | 20159.0 +- 167.8* |
| CL-L-$10^4$ | 50350 | 28500 | 29514 | 16280 | 29345.4 +- 522.6 | 23627.0 +- 266.5* | 28845.1 +- 1410.7* | **32079.7 +- 572.1** | 30869.8 +- 263.6 | 29747.4 +- 352.2 | 19980.0 +- 872.8* |
| | | | | | 29345.4 +- 522.6 | 24133.2 +- 150.9* | 23795.0 +- 0.0* | **30504.8 +- 174.5** | 29177.7 +- 80.3* | 29186.0 +- 269.5* | 19582.4 +- 0.0* |
| | | | | | 29345.4 +- 522.6 | 23060.6 +- 890.4* | 25988.5 +- 3102.1* | **30066.9 +- 0.0** | 29874.1 +- 1145.4 | 29328.3 +- 68.2* | 18983.9 +- 42.2* |
| CL-M-$10^2$ | 50350 | 37800 | 36900 | 16280 | 39546.5 +- 1922.6 | 26460.1 +- 3564.4* | 35682.4 +- 1315.7* | 39054.2 +- 1726.5* | **44500.7 +- 300.4** | 37947.5 +- 4318.3* | 19731.3 +- 881.1* |
| | | | | | 38794.9 +- 1156.5 | 22792.6 +- 137.6* | 30986.0 +- 7513.9* | 31677.2 +- 3720.2* | **41893.8 +- 2028.6** | 24715.6 +- 2688.8* | 19314.4 +- 786.7* |
| | | | | | 40364.2 +- 1062.9 | 24541.9 +- 1225.4* | 25772.6 +- 5832.2* | 27561.4 +- 2285.6* | **41684.7 +- 2741.3** | 31941.2 +- 4602.8* | 19460.4 +- 146.0* |
| CL-M-$10^3$ | 50350 | 37800 | 36900 | 16280 | 37863.2 +- 2738.8 | 24598.1 +- 1188.5* | 36195.6 +- 2107.8* | 41820.2 +- 216.7 | 41549.5 +- 399.4 | **42765.6 +- 4271.8** | 19945.1 +- 466.5* |
| | | | | | 35839.0 +- 2595.3 | 23173.8 +- 239.6* | 28364.7 +- 6063.1* | 36102.6 +- 234.3 | 41801.4 +- 323.7 | **42235.7 +- 4646.5** | 19357.5 +- 39.0* |
| | | | | | 39509.3 +- 0.0 | 24550.7 +- 320.2* | 37189.0 +- 1722.1* | **41586.1 +- 1138.1** | 40198.9 +- 1005.7 | 40182.1 +- 776.7 | 19357.6 +- 86.2* |
| CL-M-$10^4$ | 50350 | 37800 | 36900 | 16280 | 36507.0 +- 5368.8 | 23803.5 +- 2241.1* | 35587.0 +- 986.1* | **42531.8 +- 491.7** | 41974.6 +- 220.7 | 36273.5 +- 6371.8* | 19730.2 +- 483.8* |
| | | | | | 28916.8 +- 0.0 | 20727.4 +- 0.0* | 35124.8 +- 2424.5 | 37025.2 +- 898.6 | **41922.7 +- 248.1** | 21555.0 +- 116.2* | 20010.3 +- 286.8* |
| | | | | | 40136.7 +- 0.0 | 20707.9 +- 0.0* | 30467.8 +- 7399.4* | 39241.9 +- 3200.9* | **40959.7 +- 631.3** | 31854.5 +- 7296.1* | 18761.1 +- 247.4* |
| CL-H-$10^2$ | 50350 | 48600 | 48818 | 16280 | 53943.8 +- 2305.0 | 27062.8 +- 4698.8* | 46322.6 +- 1994.3* | 50611.7 +- 827.5* | **56935.0 +- 1635.5** | 54739.3 +- 2173.2 | 20322.5 +- 268.4* |
| | | | | | 52089.0 +- 1702.4 | 20468.1 +- 44.3* | 24629.8 +- 360.3* | 33029.6 +- 2574.9* | 53586.0 +- 3431.2 | **56282.5 +- 279.6** | 19742.5 +- 337.8* |
| | | | | | **53943.8 +- 2305.0** | 26627.1 +- 1246.0* | 39026.6 +- 10330.9* | 41502.8 +- 9248.0* | 53671.2 +- 4224.4* | 44156.9 +- 15029.4* | 19668.0 +- 299.3* |
| CL-H-$10^3$ | 50350 | 48600 | 48818 | 16280 | 52621.9 +- 555.1 | 29071.0 +- 3907.4* | 47516.9 +- 3168.9* | 51960.6 +- 1946.3* | **55180.4 +- 755.5** | 54011.1 +- 1307.0 | 19657.1 +- 249.5* |
| | | | | | **53086.4 +- 0.0** | 25137.9 +- 104.7* | 23338.1 +- 41.2* | 45099.0 +- 1422.0* | 51928.8 +- 1319.2* | 42022.7 +- 3099.3* | 19078.0 +- 97.6* |
| | | | | | 52256.5 +- 586.8 | 25720.6 +- 760.1* | 44729.5 +- 7872.6* | 51239.6 +- 1716.9* | **53003.4 +- 910.1** | 52480.2 +- 1764.7 | 19191.3 +- 153.2* |
| CL-H-$10^4$ | 50350 | 48600 | 48818 | 16280 | 49844.1 +- 4217.6 | 32323.4 +- 6191.3* | 47798.9 +- 2952.2* | 52814.9 +- 2485.0 | **54777.5 +- 968.6** | 43514.7 +- 15253.5* | 19954.6 +- 915.1* |
| | | | | | 46893.9 +- 4261.5 | 32175.5 +- 401.8* | 22587.6 +- 0.0* | 39742.9 +- 166.3* | **54268.7 +- 0.0** | 22785.9 +- 238.8* | 19658.0 +- 0.0* |
| | | | | | 52920.5 +- 0.0 | 39473.3 +- 544.8* | 49472.0 +- 168.6* | 50544.4 +- 1591.3* | **54896.1 +- 1371.4** | 48462.3 +- 8717.9* | 19276.6 +- 122.3* |

