# OpenReview forum: "NeoRL: A Near Real-World Benchmark for Offline Reinforcement Learning"
_NeurIPS.cc/2021/Track/Datasets_and_Benchmarks/Round1 — Submitted to NeurIPS 2021 Datasets and Benchmarks Track (Round 1)_

### Official Review · Reviewer_umUd · 2021-07-03
**Review of NeoRL: A Near Real-World Benchmark for Offline Reinforcement Learning**

**Rating:** 6
**Confidence:** 4

**Strengths:**

- Offline RL is an important research area with wide potential applications and benchmarkings can help better algorithmic development
- The evaluation experiments of state-of-the-art offline RL methods on the proposed datasets are pretty comprehensive, i.e., 7 methods on a diversity of environments and a diversity of datasets;
- This paper takes seriously the problem of off-policy evaluation for selecting the best policy which is important for really deploy offline RL method in the real world, and does a good experiment on comparing off-policy evaluation with online policy selection that demonstrates a disagreement between these two evaluation method, which can shed light on further algorithmic development of offline RL and offline policy evaluation.

**Weaknesses:**

- The proposed gym-mujoco environment has an overlap with the D4RL dataset, e.g., the 50% sac variant seems to be identical to the medium dataset in D4RL;
- Some of the key properties of the propsoed benchmarking, e.g., limited data, conservative policy, are not very well justified (see questions in the Correctness section below)
- The writing of the paper can be fairly improved.


**Additional Feedback:**

All my questions are listed above.

**Clarity:**

The paper can be further improved in terms of the writing. Below are some examples:
- line 166 - 168: "... to mimic these gaps ... " -> "to fill these gaps"?
- line 170: "... to mimic the above issues ... " -> "to address such issues"?
- line 181 - 185, the paper first states "In real scenarios, the historical datasets may not contain rewards..." on line 184 it then states that "while our default datasets contain original environment rewards for fair comparisons", which confuese me since it seems to be  contradictory to the previous statement.
- line 259, 'commonly' -> 'Commonly'
- line 291, 'Although model-free methods leverage in offline RL algorithms and are easy to use' -> 'Although model-free methods leveraged in offline RL algorithms are easy to use'
- line 365 - 366 "From the normalized scores three evaluations," -> "From the normalized scores over three evaluations,"
- Please explain how average rank is computed.
- In section 7.1.1 a converged expert sac is listed as the baseline, but its performance was not reported in table 2.

**Correctness:**

I have the following questions:
- Is the dataset of 10^3 or 10^4 trajectories really small? According to the appendix, all environments has a maximal trajectory length of 10^3 maginitude, so 10^3 trajectories already are 10^6 transitions, which is what D4RL's dataset size. So in this sense, only the 10^2 trajectory dataset size can be considered as "limited data". Please clarify on this.
- Can the authors add more justifications that why their way of collecting the dataset, i.e., using trained sac of 25%, 50%, 75% , 100% of the maximal performance is conservative/not overly exploratory? (especially the 25% one)
- How is WIS run since only a dataset but not the behaviour policy is given?



**Documentation:**

There is sufficient detail on dataset documentation.

**Ethics:**

No ethical concerns

**Relation To Prior Work:**

It is clearly discussed how this work differs from previous contributions.


**Summary And Contributions:**

This paper proposes a new benchmarking dataset for offline RL algorithms, with a focus to close the reality gap in the dataset. They point out the following reality gaps in previous offline RL benchmarkings: 1) The use of a too exploratory policy to collect the dataset, while real-world policies are usually conservative to ensure safety; 2) The size of the dataset is too large, while real-world applications only usually have very limited historical data in the magnitude of dozens; 3) Online evaluation to select the best policy, while directly deploying the policies in the real-world test environment could be dangerous and undesirable.

The proposed new benchmarking contains industrial control, financial marketing, and city energy management scenarios. It solves these gaps by 1) using three-level size of datasets of 10^2, 10^3 and 10^4 trajectoreis so there is limited data; 2) using fully or partially trained sac policies to collect the data so the dataset is not overly exploratory; 3) besides online selection, also using offline policy evaluation to select the best policy based on a leave-out test dataset.

The paper further dose a comprehensive evaluation of some of the state-of-art offline RL methods on the proposed benchmarkings, and gives a detailed analysis of the findings.

---

> ### Author Response · Authors · 2021-07-10
> **Response to reviewer umUd.**
>
> Thanks for your insightful comments.
>
> Response to all: ...
>
> Q1: The proposed gym-mujoco environment has an overlap with the D4RL dataset, e.g., the 50% sac variant seems to be identical to the medium dataset in D4RL;
>
> A1: We would like to emphasize that our data collection method is different from the previous work. Previous work collects the data by sampling from the policy output distribution, which collects more EXPLORATIVE data. In NeoRL, with only 20% probability an action is sampled from the distribution, while with 80% probability an action is deterministic (mean of the Gaussian, please see Section 5.2 and datasheets for data collection details). Therefore, our data is much LESS EXPLORATIVE. In the updated paper, we add a comparison of data distribution on the D4RL medium dataset and NeoRL medium in Figure 2, showing that the D4RL dataset covers a wider range. The same RL algorithm for learning the policy does not weaken the difference. The 3 selected Gym-MuJoCo domains are widely used in existing benchmarks, so we introduce the conservative and limited data properties into these tasks to investigate the impact on previous benchmarking results.
>
> Q2: Is the dataset of 10^3 or 10^4 trajectories really small?
>
> A2: No, the limited data in NeoRL is 10^2​. The extra two sizes are provided to verify the impact of different amounts of data on different algorithms (mentioned in Section 5.2 Line 198-200).
>
> Q3: Can the authors add more justifications that why their way of collecting the dataset.... is conservative/not overly exploratory? (especially the 25% one)
>
> A3: Our data collection method is different, as in the answer to the Q1, which leads to conservative and less explorative datasets. Please note that the "conservative" refers to the data distribution, but has no connection to the performance of the behavior policy. We plot the $(s,a)$ distribution in the updated paper (Figure 2) and it shows even the distribution of the seemingly overlapped medium tasks is narrower in NeoRL.
>
> Q4: How is WIS run since only a dataset but not the behaviour policy is given?
>
> A4: We use a BC to recover the behavior policy with TanhGaussian distribution from the logged dataset. The implementation detail can be found in Appendix E.
>
>
>
> Q5: Line 181 - 185, the paper first states "In real scenarios..." on line 184 it then states that "while our default ..."
>
> A5: For real-world applications, we provide an interface to define reward function if necessary, while for benchmarking, we still use the original environment reward. We have revised the paper to make it clear (see Line 176-180).
>
>
>
> Q6: Please explain how average rank is computed.
>
> A6: The rank of compared baselines and algorithms is determined by the score on each task, and the final average rank is computed over the 51 tasks (see Line 341-343).
>
>
>
> Q7: A converged expert sac is listed as the baseline...
>
> A7: The expert policy is used as a reference for a good policy, however, we may not have access to it in real world. The raw score of the expert policy can be found in Appendix E. This score is used for normalization (the normalized score is always 100). The raw score will be the same on the same domain and it only loses the first place on a very small portion of tasks. So we just omit it in the ranking.

---

> > ### Comment · Reviewer_umUd · 2021-07-20
> > **Response**
> >
> > I thank the authors for the clarification. The response has addressed some of my concerns and I have updated my score accordingly.

---

### Official Review · Reviewer_eqVf · 2021-07-04
**More justification on the need of new tasks for benchmarking batch RL is needed**

**Rating:** 5
**Confidence:** 4
**Correctness:** See the above section.

**Strengths:**

The paper provides a set of new tasks and show extensive benchmarking results of existing algorithms. Additionally, the paper shows clearly that with offline policy selection, the performance of most batch RL algorithms degrades significantly.

**Weaknesses:**

1. The overall novelty of the work is limited. While a set of new tasks are proposed, given the existing benchmarks such as D4RL and RL Unplugged, it is not clear what are the main properties of the newly proposed tasks that set them apart. The differences in data properties and policy selection as described in Table 1 could also be done for the existing benchmark. The only difference to previous tasks that the authors claim is its high stochasticity and non-stationarity. But the importance of these properties are not discussed or empirically shown in the paper. Additionally, offline policy selection has also been proposed in previous work (RL Unplugged).

2. The paper claims that one difference of NeoRL dataset from previous benchmark is the conservative data collection. However, the paper still trains an RL policy in the original environment and use it as the behaviour policy. How is the conservative data collection implemented in these tasks? How much does this factor affect the relative performance of different algorithms?

**Additional Feedback:**

No additional feedback

**Clarity:**

The introduction could be improved by explaining more clearly the difference to previous benchmarks and the motivation for new tasks. There are numerous grammar and language issues in the paper and I would suggest the authors get editing help from someone with full professional proficiency in English.

**Documentation:**

There are not enough details in the documentation on how the dataset are collected or how to reproduce the data collection process.

**Relation To Prior Work:**

See the weakness section.

**Summary And Contributions:**

The paper proposes a new benchmark dataset for batch reinforcement learning, including different simulated tasks such as industrial control, financial trading, etc. The paper show results of existing algorithms under more constrained settings such as conservative actions, limited data and offline policy selection and show that many existing batch RL algorithms does not outperform behaviour cloning or the deterministic behaviour policy.

---

> ### Author Response · Authors · 2021-07-10
> **Response to reviewer eqVf.**
>
> Response to all: ...
>
>
> Q1: The differences in data properties and policy selection as described in Table 1 could also be done for the existing benchmark. The only difference to previous tasks that the authors claim is its high stochasticity and non-stationarity.
>
> A1: This work mainly focuses on correcting the data properties, not only proposing new tasks. You are right that the properties COULD also be done for the existing tasks, which unfortunately had not been done yet, and previous datasets may mislead the advance of the field due to the unrealistic data properties. A common benchmark of datasets, not just tasks, is quite necessary for comparing offline-RL algorithms, which is the motivation and contribution of this work.
>
>
>
> Q2: Additionally, offline policy selection has also been proposed in previous work (RL Unplugged).
>
> A2: Offline policy selection has been NOTICED in RL Unplugged, while has not realized yet. The RL Unplugged paper states *"Evaluation by offline policy selection (see Figure 2 (right)) has been less popular, but is important as it is indicative of robustness to imperfect policy selection, which more closely reflects the current state of offline RL for real-world problems. However it has downsides too, namely that there are many design choices including what data to use for offline policy selection, whether to use value functions trained via offline RL or OPE algorithms, which OPE algorithm to choose, and the meta question of how to tune OPE hyperparameters. Since this topic is still under-explored, we prefer not to specify any of these choices. Instead, we INVITE the community to innovate to find which offline policy selection method works best.".* Besides, the naive approach for offline policy selection in RL Unplugged was unrealistic, i.e., choosing the set of hyper-parameters that performs best overall on the online policy selection tasks from the same domain. On the other hand, D4RL paired a similar domain for each task to conduct online policy selection (hyperparameter tuning). In practice there is commonly no chance or a simulator from similar domain to perform online policy selection, thus NeoRL prepares the validation datasets for policy validation, which is not only feasible in practice but also a challenging to use validation dataset for policy selection.
>
>
>
> Q3: The paper claims ... still trains an RL policy in the original environment and use it as the behaviour policy. How is the conservative data collection implemented in these tasks?
>
>
> A3: We do NOT include the training data as the benchmark dataset, but only after the training, employ the trained policy as the fixed behavior policy to collect the dataset, which provides conservative and less explorative data collection. We have revised the paper to make this point clearer (see Section 5.2).
>
>
>
> Q4: How much does this factor (conservative data) affect the relative performance of different algorithms?
>
> A4: One observation from the new benchmark experiments is that many algorithms fail to outperform BC and the deterministic behavior policy, which were not observed in previous benchmarks due to their explorative data collection. One outstanding challenge of offline-RL is that the algorithm cannot explore. However, in previous benchmarks, this challenge was much weakened due to the explorative data collection. We would like to re-emphasize the challenge. We add a comparison of different algorithms tested on D4RL Gym-MuJoCo medium tasks in Table 3 (please see Line 366 - 370 for details). The comparison results imply most algorithms may be overestimated.

---

> > ### Comment · Reviewer_eqVf · 2021-07-19
> > **The clarification and updates from the authors has made the paper's contribution more clear**
> >
> > I thank the authors for the clarification. The added comparison in table 3 alleviates my second concern. I have updated my score.
> >
> > However, the added experiments in Figure 2 are confusing to me: While it shows that data in NeoRL is more constrained, it seems that there is almost no overlapping between the data in NeoRL and D4RL. I hope the authors can clarify if this is due to additional differences between NeoRL and D4RL, other than the amount of exploration in the behavior policy.

---

### Official Review · Reviewer_1JnW · 2021-07-05
**The aim of building this benchmark is clear but there are several problems.**

**Rating:** 4
**Confidence:** 3
**Clarity:** I could follow, but further revision …

**Strengths:**

1.	Data size and quality are split into different levels, making it possible for a more detailed analysis of models.
2.	The construction of this benchmark reveals some problems of existing off-line RL methods, providing insights for future model design.


**Weaknesses:**

1.	Comparison with other works is not detailed enough.
2.	Motivation for constructing this dataset should be justified with more evidence.


**Additional Feedback:**


This benchmark could support 4 selected tasks and the authors claim that in the future they will investigate more tasks, so what are the underlying rules to choose included tasks? What are the rules?

**Correctness:**

The claims made in the paper and the supporting experiments/evaluation methods look correct to me.

**Documentation:**

The documentation is provided but not so detailed.

**Ethics:**


I could not spot any ethical issues currently.

**Relation To Prior Work:**

The difference from previous contributions should be further explained and clarified.

**Summary And Contributions:**

This works aims at dealing with the reality gap of offline RL benchmarks, containing datasets from different domains (i.e. robotics, industrial control, finance trading, and city management) with controlled data amount.
Evaluation of some representative methods is conducted based on the proposed dataset.

---

> ### Author Response · Authors · 2021-07-10
> **Response to reviewer 1JnW.**
>
> Response to all:
> ...
>
> Q1: Comparison with other works is not detailed enough.
>
> A1: Existing benchmarks mainly consider task properties, such as partial observability, the difficulty of exploration, etc. Meanwhile, the real-world offline data are often conservative and limited. This work thus focuses on data properties. Moreover, the step of offline policy evaluation and selection is missing in previous benchmarks. Consequently, current offline RL algorithms may have been overly estimated (e.g., many algorithms fail to outperform the deterministic behavior policy). This work tries to solve the above issues in the new benchmark.
>
>
> Q2: Motivation for constructing this dataset should be justified with more evidence.
>
> A2: As the above common response states, there was a significant gap between the previous benchmarks and the real-world scenarios, and thus some offline algorithms tested on previous benchmarks may have been overly estimated. The NeoRL is motivated by the reality gap, and aims at providing a near real-world benchmark. According to the experiments, we do find some overly optimistic results in the previous benchmarks.
>
>
>
> Q3: The authors claim that in the future they will investigate more tasks, so what are the underlying rules to choose included tasks? What are the rules?
>
> A3: We believe offline-RL general for decision-making and controlling tasks but currently has not yet been widely recognized. We will include more benchmark tasks that are representative and have positive impact on the social development, such as industrial controlling, scheduling and dispatching scenarios. These tasks are with high-dimensional action spaces and complex transition dynamics, and for the sake of safety, the actions are more conservative. Thus, datasets and tasks from these domains often provide unprecedented challenges for offline-RL algorithms.

---

### Author Response · Authors · 2021-07-15
**Response to all**


We would like to highlight our contribution, that is, not only to propose new benchmark domains, but more importantly, to correct the way how offline data should be collected and used in practical reinforcement learning scenarios. According to our several years of experience in landing reinforcement learning in various real-world domains, such as recommender systems, sales promotion, industrial control, logistics optimization, etc., real-world offline RL data are quite different from the previous benchmarks (e.g., RL Unplugged and D4RL) in several aspects, including

(1) behavior policies are conservative and less explorative,

(2) baselines are deterministic but non-explorative,

(3) policy validation before going online is crucial.

The NeoRL benchmark, consisting of data from 3 extra near realistic domains and 3 old locomotion controlling domains with conservative and limited datasets, reflects the real-world scenario properties more closely, and thus helps develop practical offline-RL algorithms. Besides, as a MUST step before deployment, the policy selection and validation with two OPE methods are included in NeoRL benchmarking results, while it has never been realized in RL Unplugged and D4RL.

We found some major misunderstandings in our work:

(1) Even though both using a trained SAC policy to collect data, the data collection process in this work is different from the previous work in that we shut down the exploration with 80% probability (see Section 5.2 Line 185-192 and Line 194-196).

(2) The conservativeness refers to less explorativeness but has nothing to do with the policy performance. In real-world tasks, the behavior policy can have different performances, but are mostly deterministic.

We thank the review comments and have updated the paper accordingly to include the revisions (we use magenta color to highlight the main revisions on texts.). Particularly we have added Figure 2 to show that our data distribution is much more narrow than previous datasets. We also added Table 3 to show that some offline algorithms are overestimated on previous benchmarks with explorative data.

---

### Decision · Program_Chairs · 2021-07-26

**Decision:**

Reject

**Comment:**

This paper tries close the reality gap of offline RL benchmarks. The benchmark has datasets from several simulated domains (i.e. robotics, industrial control, finance trading, and city management) with controlled data amount. They have evaluated some offline RL baselines on their datasets.

The paper is not very well-written it feels a bit rushed. It seems like the reviewers were confused about the main points of the paper as well. The comparisons with other works is not detailed enough. There are some missing citations such as RWRL [1], also claims related to the OPE are not clear to me, what actually is different here compared to DOPE or offline policy selection tasks in RL Unplugged.

Also the paper claims that the behavior policies in NeoRL are more conservative and less explorative. But I am not fully convinced with that, it is something the paper claims without any solid backings.

The reviewer's scores also point this paper towards rejection. I would recommend the author's to consider rewriting the paper with clarifications and addressing the concerns raised by the reviewers and resubmit it to a future venue.

[1] Dulac-Arnold, Gabriel, et al. "An empirical investigation of the challenges of real-world reinforcement learning." arXiv preprint arXiv:2003.11881 (2020).